# The lytic polysaccharide monooxygenase CbpD promotes *Pseudomonas aeruginosa* virulence in systemic infection

Fatemeh Askarian [1✉], Satoshi Uchiyama[2,14], Helen Masson[3,14], Henrik Vinther Sørensen [4], Ole Golten[1], Anne Cathrine Bunæs[1], Sophanit Mekasha[1], Åsmund Kjendseth Røhr[1], Eirik Kommedal [1], Judith Anita Ludviksen[5], Magnus Ø. Arntzen [1], Benjamin Schmidt[2], Raymond H. Zurich[2], Nina M. van Sorge [6,7,8], Vincent G. H. Eijsink[1], Ute Krengel [4], Tom Eirik Mollnes[5,9,10,11], Nathan E. Lewis [2,3,12], Victor Nizet [2,13✉] & Gustav Vaaje-Kolstad [1✉]

The recently discovered lytic polysaccharide monooxygenases (LPMOs), which cleave polysaccharides by oxidation, have been associated with bacterial virulence, but supporting functional data is scarce. Here we show that CbpD, the LPMO of *Pseudomonas aeruginosa*, is a chitin-oxidizing virulence factor that promotes survival of the bacterium in human blood. The catalytic activity of CbpD was promoted by azurin and pyocyanin, two redox-active virulence factors also secreted by *P. aeruginosa*. Homology modeling, molecular dynamics simulations, and small angle X-ray scattering indicated that CbpD is a monomeric tri-modular enzyme with flexible linkers. Deletion of *cbpD* rendered *P. aeruginosa* unable to establish a lethal systemic infection, associated with enhanced bacterial clearance in vivo. CbpD-dependent survival of the wild-type bacterium was not attributable to dampening of pro-inflammatory responses by CbpD ex vivo or in vivo. Rather, we found that CbpD attenuates the terminal complement cascade in human serum. Studies with an active site mutant of CbpD indicated that catalytic activity is crucial for virulence function. Finally, profiling of the bacterial and splenic proteomes showed that the lack of this single enzyme resulted in substantial re-organization of the bacterial and host proteomes. LPMOs similar to CbpD occur in other pathogens and may have similar immune evasive functions.

[1] Faculty of Chemistry, Biotechnology and Food Science, Norwegian University of Life Sciences (NMBU), Ås, Norway. [2] Division of Host-Microbe Systems & Therapeutics, Department of Pediatrics, UC San Diego, La Jolla, CA, USA. [3] Department of Pediatrics, University of California, San Diego, School of Medicine, La Jolla, CA, USA. [4] Department of Chemistry, University of Oslo, Oslo, Norway. [5] Research Laboratory, Nordland Hospital, Bodø, Norway. [6] Department of Medical Microbiology, University Medical Center Utrecht, Utrecht University, Utrecht, The Netherlands. [7] Department of Medical Microbiology and Infection Prevention, Amsterdam University Medical Center, University of Amsterdam, Amsterdam, The Netherlands. [8] Netherlands Reference Laboratory for Bacterial Meningitis, Amsterdam University Medical Center, Amsterdam, The Netherlands. [9] K.G. Jebsen TREC, Faculty of Health Sciences, UiT- The Arctic University of Norway, Tromsø, Norway. [10] Department of Immunology, Oslo University Hospital, and K.G. Jebsen IRC, University of Oslo, Oslo, Norway. [11] Center of Molecular Inflammation Research, Norwegian University of Science and Technology, Trondheim, Norway. [12] Novo Nordisk Foundation Center for Biosustainability at UC San Diego, University of California, San Diego, School of Medicine, La Jolla, CA, USA. [13] Skaggs School of Pharmacy and Pharmaceutical Sciences, UC San Diego, La Jolla, CA, USA. [14] These authors contributed equally: Satoshi Uchiyama, Helen Masson. ✉email: fatemeh.askarian@nmbu.no; vnizet@health.ucsd.edu; gustav.vaaje-kolstad@nmbu.no

Lytic polysaccharide monooxygenases (LPMOs) are copper-dependent enzymes that cleave glycosidic linkages by oxidation[1–4]. Most characterized LPMOs act on recalcitrant polysaccharides such as cellulose ($\beta$-1,4(Glc)$_n$) and chitin ($\beta$-1,4 (GlcNAc)$_n$)[5]. Hence, current LPMO research focuses mainly on biomass degradation from fundamental and applied perspectives. All LPMOs possess a conserved active site with two histidine residues coordinating a copper ion in a so-called "histidine brace"[3]. Catalysis entails the reduction of the copper followed by activation of an oxygen-containing co-substrate to oxidize the C1 or C4 carbon of the polysaccharide substrate (reviewed in ref. [5]). The co-substrate may be either dioxygen or hydrogen peroxide, the latter yielding catalytic rates several orders of magnitude higher than the former[6–8]. LPMOs target multiple soluble and insoluble substrates, demonstrating their catalytic versatility (reviewed in ref. [9]). Despite being primarily associated with biomass degradation machineries[9,10], genes encoding LPMOs are found in many pathogenic organisms and have been speculated to contribute to bacterial physiological processes or pathogenicity phenotypes[11–19]. However, scant data exist to substantiate LPMO function during bacterial infection.

One important pathogen that expresses a putative LPMO is *Pseudomonas aeruginosa* (PA), a frequently multidrug-resistant microorganism associated with nosocomial infections of surgical sites, urinary tract, lung, and blood[20]. PA accounts for ~20% of hospital-associated Gram-negative bacteremia cases, with high mortality (>30%), particularly in patients receiving inappropriate initial antimicrobial treatment[21,22]. PA is known to explicitly disarm several aspects of the innate host defense and erodes efficacy of first-line antibiotics through an array of immune evasion factors and antibiotic-resistant determinants (reviewed in refs. [23,24]).

The putative LPMO expressed by PA is called "chitin-binding protein D" (CbpD). Many publications refer to CbpD as a virulence factor (e.g., ref. [25] and references within), yet no direct functional evidence exists. Prevalent across PA clinical isolates and cystic fibrosis-associated clones[26,27], CbpD is expressed under quorum-sensing control[28–32], secreted through the Type II secretion machinery[33,34], and modified by several post-translational mechanisms[25,35,36]. The induction of *cbpD* transcription by human respiratory mucus in mucoidal PA[37] and its high abundance in the secretome of an acute transmissible cystic fibrosis-associated strain[38], are two observations that strongly suggest an important function of this LPMO in disease pathogenesis.

In this work, we combine studies of CbpD structure and enzymatic activity with analysis of its immune evasion properties ex vivo and in vivo to unravel the function of this protein in PA bloodstream infection. We show that this highly conserved and prevalent tri-modular LPMO is a chitin-oxidizing virulence factor that enhances resistance of the bacterium to whole blood-mediated killing and virulence during ex vivo and in vivo infection by attenuation of the terminal complement cascade.

## Results

### CbpD has a monomeric, tri-modular structure.
Analysis of the CbpD amino acid sequence (Uniprot ID Q9I589) revealed a multidomain architecture comprising an N-terminal signal peptide (residues 1–25) followed by an auxiliary activity family 10 LPMO (residues 26–207, AA10), a module of unknown function (residues 217–315, annotated as "GbpA2" domain in the Pfam classification, here referred to as module X or MX) and finally a C-terminal family 73 carbohydrate-binding module (CBM73; residues 330–389) (Fig. 1a). The protein is highly conserved and prevalent among PA clinical and non-clinical isolates (Supplementary Fig. 1

and Supplementary Results). Phylogenetic analysis of predominantly pathogenic Gram-negative and Gram-positive bacteria placed the CbpD AA10 sequence in a cluster containing homologs from *Legionella spp.* (Supplementary Fig. 2). Most of the closely related LPMOs possess a modular architecture similar to CbpD. To gain further insight into the CbpD structure, homology modeling, molecular dynamics simulations, and small-angle X-ray scattering (SAXS) analysis were performed. Combined, these approaches revealed that CbpD is a monomeric, elongated protein, whose conformation may be affected by post-translational modifications (Fig. 1a–c and Supplementary Figs. 3−5, Supplementary Tables 1−3, and Supplementary Results).

### CbpD enzymatic performance is influenced by other redox-active virulence factors.
The CbpD active site resembles those found in other LPMO10s, containing a copper ion coordinated by two histidine residues (H26 and H129) and a semi-conserved glutamic acid residue (E199) (Fig. 1b). The recombinant, copper-saturated full-length enzyme produced in *Escherichia coli* (rCbpD$_{EC}$) was active toward $\beta$-chitin, resulting in the release of chitooligosaccharide aldonic acids ranging from 2 to 8 in the degree of polymerization (Fig. 1d). The CbpD catalytic module (AA10) alone was also active toward $\beta$-chitin, whereas no breakdown products were generated by the MX or CBM73 modules (Fig. 1d). CbpD chitin oxidation activity required an intact active site, as the mutation of one of the active site histidines (H129: rCbpD$_{EC-H129A}$) to alanine (Ala) abolished CbpD activity (Supplementary Fig. 6a, b). When secreted by PA, CbpD is post-translationally modified (refs. [25,35,36] and Supplementary Table 1), which may influence enzymatic performance. Full-length CbpD variants produced and purified from heterologous or homologous expression in *E. coli* (rCbpD$_{EC}$) or wild-type (WT) PA14 (*P. aeruginosa* UCBPP-PA14) (rCbpD$_{PA}$), respectively, were equally active on the $\beta$-chitin model substrate (Supplementary Fig. 6c).

Experiments using full-length and truncated forms of CbpD demonstrated that all variants bound chitin, albeit with different binding kinetics (Supplementary Fig. 7a). To explore other potential CbpD substrates, the binding properties of rCbpD$_{EC}$ and rCbpD$_{PA}$ were screened using a mammalian glycan array that contains 585 glycan structures present in mammals, including a variety of structures found in mucins such as core 1, type 1, blood group, and Lewis type glycans[39]. Binding was not detected to any glycan for either CbpD variant at 5 and 50 $\mu$g ml$^{-1}$ (Supplementary Data 1), nor was binding to GlcNAc$_6$ (water-soluble chitooligosaccharide) observed (Supplementary Fig. 7b and Supplementary Data 1).

In addition to CbpD, PA secretes the redox-active virulence factors azurin (AZU), a copper protein that contributes to electron transfer reactions[40], and pyocyanin (PCN), a zwitterionic secondary metabolite capable of oxidizing and reducing other molecules, including reduction of O$_2$ to H$_2$O$_2$[41,42]. We investigated the influence of azurin and pyocyanin on CbpD enzymatic performance. Pyocyanin boosted CbpD activity towards $\beta$-chitin using ascorbate as an external electron donor (Fig. 1e), perhaps due to pyocyanin catalyzing the formation of H$_2$O$_2$, which is an LPMO co-substrate. Excessive pyocyanin concentrations (1 mM) led to rapid inactivation of CbpD, similar to what is commonly observed for LPMOs exposed to high H$_2$O$_2$ concentrations (Fig. 1e). Azurin, on the other hand, has been proposed to protect PA from H$_2$O$_2$[43], but the addition of this copper protein to a reaction containing pyocyanin, CbpD, and ascorbate, did not abolish CbpD activity (Fig. 1f). Rather, 1 $\mu$M azurin activated 1 $\mu$M copper-free CbpD, which otherwise was inactive (Fig. 1f), suggesting that CbpD may be able to obtain copper from azurin.

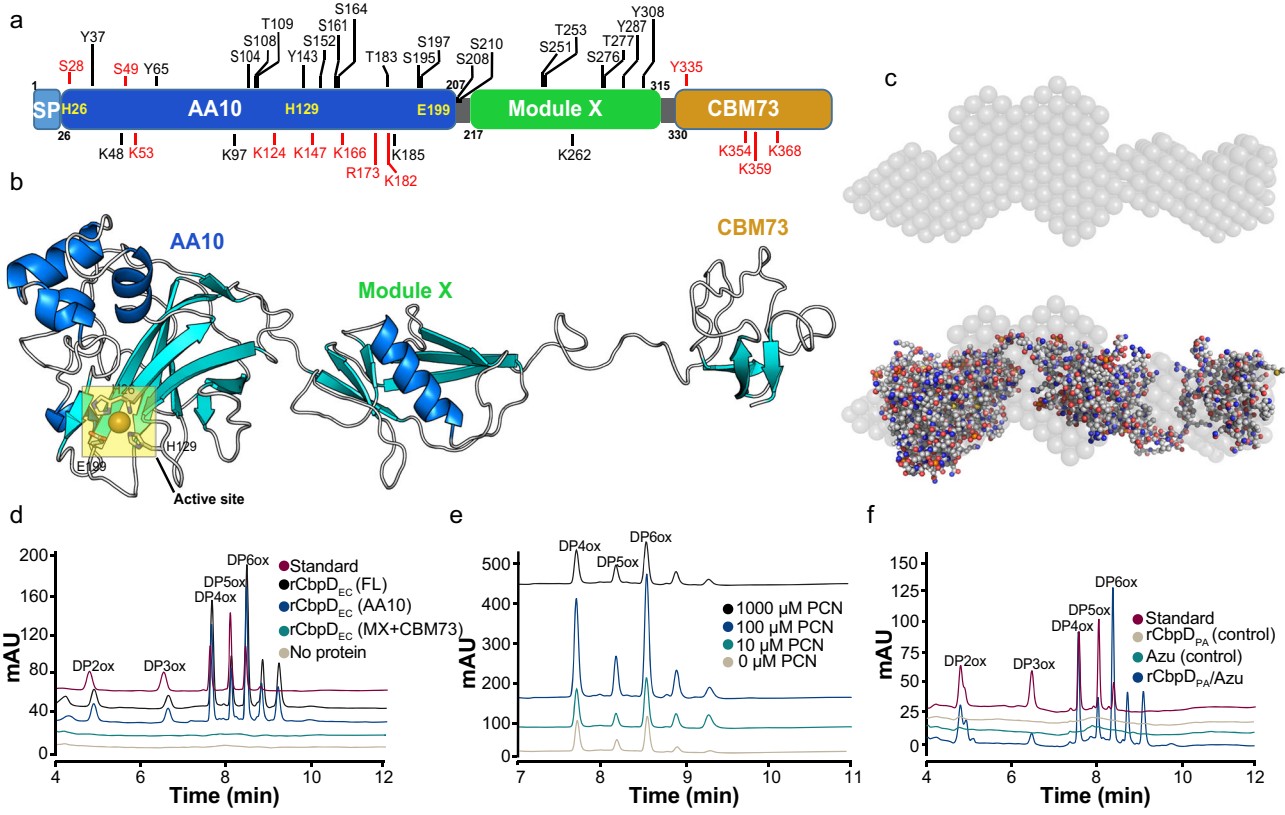

**Fig. 1 Multidomain structure of CbpD and evaluation of its activity on a model substrate. a** Schematic representation of the domain architecture of CbpD. The full-length protein contains a signal peptide (SP), an N-terminal AA10-type LPMO domain (AA10) followed by a module with unknown function (module X or MX) and a C-terminal CBM domain (CBM73). The residue boundaries for each domain, as well as potential post-translational modification (PTM) sites based on refs. [25,35,36] and Supplementary Table 1, are labeled; phosphorylation sites are indicated above and lysine/arginine modifications below the domain illustration. PTMs identified in this study (Supplementary Table 1) are colored red. **b** Homology model of CbpD generated with Raptor-X[92] showing flexible linkers in an extended conformation. The active site is indicated by a yellow partially transparent square. **c** SAXS model of monomeric CbpD ($\chi^2 = 2.35$; produced with Pepsi-SAXS[114]; SASBDB ID: SASDK42), superimposed onto the ab initio SAXS model "envelope" (produced with DAMMIF[111,112] based on an average of 20 calculated models). Panels **b** and **c** were generated using PyMol. **d** Product formation by 1 μM of rCbpD$_{EC}$ variants (FL: full-length; AA10: AA10 module only; MX + CBM73: the MX and CBM73 domains only) after a 2 h reaction at 37 °C with 10 mg ml$^{-1}$ β-chitin in 20 mM Tris-HCL pH 7.0, with 1 mM ascorbate as reducing agent, analyzed by HILIC. The degrees of polymerization (DP) of oxidized chitooligosaccharide aldonic acids in a standard sample are indicated. Control reactions without ascorbate did not show product formation. **e** HILIC analysis of reaction products emerging from a reaction of 1 μM rCbpD$_{EC}$ with 10 mg ml$^{-1}$ β-chitin, 250 μM ascorbate in 20 mM Tris-HCl pH 7.0, and serial dilutions of pyocyanin (PCN) for 2 h at 37 °C. The chromatograms have been offset on the *y* axis to enable visual interpretation of their quantitative magnitude. Chromatograms for reactions containing serial dilutions of PCN and with or without various reductants that were sampled at different time points are shown in Supplementary Fig. 7c. **f** HILIC analysis of reaction products emerging from a reaction of 1 μM of copper-free full-length rCbpD$_{PA}$ with 10 mg ml$^{-1}$ β-chitin in 20 mM Tris-HCL pH 7.0, 100 μM pyocyanin (PCN), 250 μM ascorbate, in the presence or absence of 1 μM azurin (Azu) after incubation for 2 h at 37 °C. Control reactions without added ascorbate did not show product formation (Supplementary Fig. 7d).

**Transcription of *cbpD* is subject to environmental variations.**
Absolute quantification of *cbpD* transcription in various conditions revealed several noticeable features. First, *cbpD* is transcribed during growth in Luria–Bertani medium (LB). When PA14 (WT) was grown in M9 minimal medium supplemented with glucose and casamino acids (M9$_{PA}$; a growth medium mimicking nutritional deprivation), *cbpD* abundance was ~1500 copies/100 ng RNA in mid-exponential phase (OD$_{600}$ = 0.6), approximately twice the levels seen in LB medium (Fig. 2a). The addition of normal human serum (NHS; 10% v/v, 30 min) to M9$_{PA}$ (M9$_{PA}$/NHS) at OD$_{600}$ nm = 0.6 did not change *cbpD* transcript abundance (Fig. 2a). Thus, under the conditions tested here, *cbpD* is constitutively transcribed, with expression levels being subject to environmental variations. Transcription of LPMO-encoding genes in two other pathogenic bacteria was also influenced by the growth condition (Supplementary Fig. 8a, b and Supplementary Results).

**Loss of *cbpD* distinctly influences the PA proteome under host-mimicking conditions.** Following verification of *cbpD* transcription, the functional impact of CbpD on PA pathophysiology was scrutinized. This was achieved by comparing the proteomic profiles of the WT parent strain PA and its isogenic mutant PA14Δ*cbpD* (hereinafter referred to as "ΔCbpD") grown to mid-logarithmic phase (OD$_{600}$ = 0.6) in LB medium or in a tissue culture medium (RPMI supplemented with 10% LB) that better mimics in vivo conditions, with or without supplemental normal human serum (NHS). Secretion of CbpD in LB and RPMI was verified by quantitative proteomic analysis of the secretome (Supplementary Fig. 8c).

Our analysis identified 2128 proteins, of which 1202 were shared for both strains under all three conditions (Fig. 2b and Supplementary Data 2). Principal component analysis (PCA) (Fig. 2c) and hierarchical clustering (Fig. 2e) showed strong coherence between biological replicates and a different protein

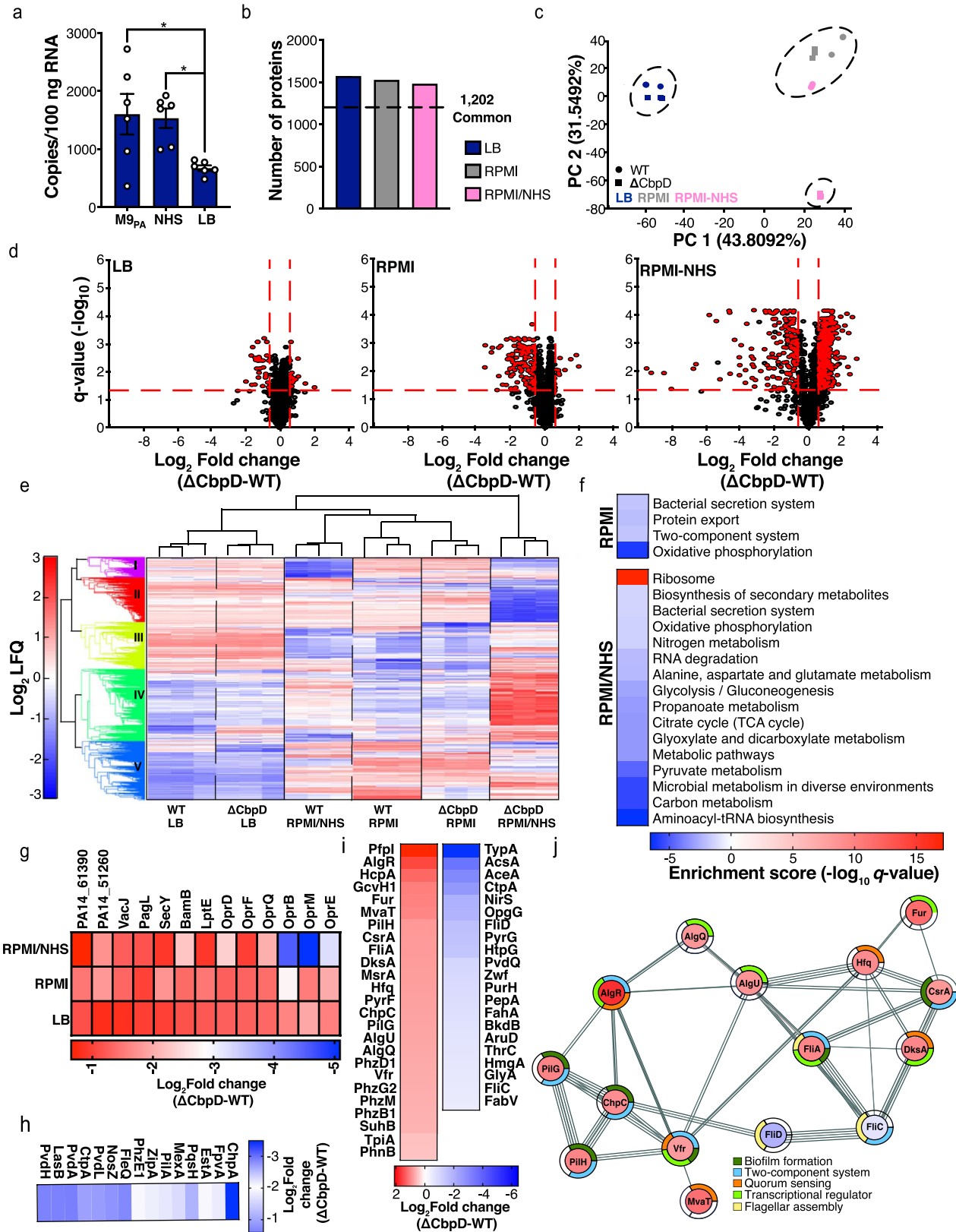

expression response of ΔCbpD compared to WT, particularly in the presence of human serum (RPMI/NHS). CbpD is part of cluster V (Fig. 2e, Supplementary Fig. 8e, and Supplementary Data 5), which is enriched in proteins related to transporter activity (Supplementary Table 4). Volcano plot analyses (Fig. 2d and Supplementary Data 3) showed that exposure to NHS

resulted in significant differential regulation of 589 proteins in ΔCbpD vs. WT (Fig. 2d, right panel), compared to a more modest number of differentially expressed proteins in LB ($n = 50$) (Fig. 2d, left panel) or RPMI ($n = 136$) (Fig. 2d, middle panel). KEGG Pathway enrichment analysis of the differentially regulated proteins (ΔCbpD vs. WT) revealed that loss of CbpD alters

**Fig. 2 CbpD expression and proteomics analysis. a** Expression of *cbpD* in wild-type (WT) PA14 was evaluated for the mid-exponential growth phase ($OD_{600} = 0.6$) in LB medium, and $M9_{PA}$ or $M9_{PA}$ supplemented with NHS (10%, 30 min incubation). The data are plotted as the mean ± SEM, representing three experiments performed in duplicate and analyzed by two-way ANOVA (Tukey's multiple comparisons test; $M9_{PA}$ vs LB $P = 0.0279$, $M9_{PA}$ vs NHS $P = 0.0410$). **b** Histogram showing the total number of the quantified proteins per condition in WT and its isogenic mutant, ΔCbpD. **c** Principal component analysis (PCA) performed on the entire proteome. For each individual replicate, the quantified proteins were plotted in two-dimensional principal component space by PC1 = 43.8092% and PC2 = 31.5492% and clustered according to growth condition and strain. **d** Volcano plots demonstrating differentially abundant proteins and *q* values of significance identified by comparing the ΔCbpD vs. WT proteomes. The red dotted line(s) crossing the *y* axis and *x* axis indicate significance cutoff at $q = 0.05$ ($log_{10} = 1.3$) and ($+/−$) 1.5-fold change ($log_2 = 0.58$) in protein abundance. Cutoff values for significance were set to fold change $\geq 1.5$ and $q \leq 0.05$ in a two-tailed paired *t* test. **e** Hierarchical clustering of proteins expressed by WT and ΔCbpD in LB, RPMI, and RPMI-NHS using agglomerative hierarchical clustering analysis. For visualization, rows (genes) have been standardized, so that the mean is 0 and the standard deviation is 1. **f** Heatmap showing KEGG pathways that were significantly enriched in ΔCbpD vs. WT upon growth in RPMI or RPMI/NHS. **g** Heatmaps showing the average fold change values ($log_2$) for significantly regulated proteins belong to shared proteome in the ΔCbpD compared to WT strain in all growth conditions. **h, i** Heatmaps showing the average fold change values ($log_2$) for selected regulators and virulence factors in the unique proteome that are significantly regulated in ΔCbpD relative to WT during growth in RPMI (**h**) or RPMI-NHS (**i**). The selected proteins are marked in bold in Supplementary Data 3. **j** Functional analysis of protein-protein interaction networks revealed some highly interconnected proteins listed in (**i**). The heatmap (**i**) was mapped to the nodes using a blue–white–red gradient that indicates fold change log2 LFQ (ΔCbpD/WT). Proteins without any interaction partners (singletons) or chains with no interaction with the main network have been omitted from the visualization. Source data are provided as a Source Data file (**a**) or Supplementary Data.

metabolism and protein synthesis (RPMI/NHS) and impairs the protein secretion apparatus (RPMI) of the ΔCbpD strain under host-mimicking conditions, hinting at the potential importance of the LPMO during infection (Fig. 2f). The most profound difference between WT and ΔCbpD in presence of NHS were reflected in clusters II and IV (Fig. 2e, Supplementary Fig. 8e, and Supplementary Data 5), which were also associated with decreased metabolic processes and increased translational activity, respectively (Fig. 2f and Supplementary Table 4). No significant pathway enrichments were observed when comparing the two strains grown in LB medium.

Several proteins associated with outer membrane homeostasis in PA[44–48] were significantly downregulated (Fig. 2g and Supplementary Data 3, 4, and 6) upon loss of CbpD or only detected in WT parent strain under all examined conditions. Channel proteins (OprM, OprD, OprQ, OprE, OprB, and OprF), LPS-modifying/assembly proteins (PagL, LptE, LptD), outer membrane protein assembly Bam complex (BamB, BamE, BamD) and VacJ (Fig. 2g and Supplementary Data 6) were among those proteins. In addition, several proteins associated with the type II secretion system (SecD, SecY, and XcpQ)[49] were commonly downregulated or not detected upon loss of CpbD (Supplementary Data 6). Despite the roles of these proteins in structural stability of the PA outer membrane, both strains displayed comparable detergent resistance upon growth in LB (Supplementary Fig. 9a). Given the function of LptE, LptD and PagL in lipopolysaccharide (LPS) biogenesis or cell surface localization, LPS was extracted from mid-log phase WT and the ΔCbpD mutant. Compositional and structural analysis revealed neither difference between the lipid-A structures (Supplementary Fig. 9d) nor in the O-antigen and outer core monosaccharide composition (Supplementary Fig. 9c). However, the inner core monosaccharide composition had a ~50% higher amount of 3-deoxy-D-*manno*-oct-2-ulosonic acid (Kdo) in the extracted LPS from ΔCbpD compared to WT (Supplementary Fig. 9b), indicating that CbpD has a direct or indirect influence on LPS composition.

When grown in the tissue culture medium RPMI, several virulence-related proteins (Supplementary Data 3; marked in bold), including proteases (e.g., LasB), pyoverdine-related biosynthesis (e.g., FpvA, PvdL, PvdH, PvdA), quorum-sensing-related regulators (e.g., PqsH), virulence-associated transcriptional factors (e.g., FleQ), and pilus synthesis (e.g., ChpC, PilA), were significantly repressed in the ΔCbpD strain compared to WT (Fig. 2h). Finally, upon exposure to NHS, a wider set of functionally interconnected regulators (e.g., Vfr, AlgU, MvaT,

AlgR, AlgQ, Fur, FliA) and proteins associated with virulence or environmental versatility of PA (Supplementary Data 3; marked in bold), were differentially up- or downregulated in the mutant lacking *cbpD* (Fig. 2i, j). Several super-regulators of quorum-sensing (QS) in PA (AlgR, DksA, MvaT, and Vfr)[50], or QS by-products such as phenazine synthesis (PhzM, PhzB1, and PhnB), T3SS regulator (Vfr), or virulence-associated transcriptional factors (AlgU, AlgR)[51] were upregulated (Fig. 2j, I) in ΔCbpD compared to WT in the presence of NHS. In addition, several proteins involved in the biosynthesis of antibiotics (BkdB, AcsA, Zwf, PurH, TpiA, GcvH, GlyA) or flagellar-associated proteins (FliD, FliC, FliA) were among the differentially (up/down) regulated proteins in ΔCbpD vs. WT (Fig. 2i, j). Proteome modulation could be a protective strategy to aid the pathogen survival within a hostile host environment. The different modulation strategies employed by WT and ΔCbpD PA strain suggest that CbpD action directly or indirectly affects several cellular processes associated with metabolism and pathogenicity upon exposure to host environmental conditions.

**CbpD promotes PA resistance to human whole blood killing ex vivo.** PA is an important cause of nosocomial bacteremia with high associated mortality[52], and as NHS resulted in distinct proteome modulation in ΔCbpD compared to WT, we next explored if the deletion of CbpD affected PA survival in freshly collected human blood. While the deletion of *cbpD* did not alter PA growth in standard media (Supplementary Fig. 10), it significantly reduced PA survival in human blood compared to the parent strain (Fig. 3a, left panel). Complementation of the ΔCbpD mutant by plasmid-mediated expression of *cbpD* (ΔCbpD:CbpD) restored CbpD expression (Supplementary Fig. 11a) and significantly enhanced human blood survival compared to the ΔCbpD mutant empty vector control (ΔCbpD:Mock) (Fig. 3a, right panel). Control experiments showed that the contribution of CbpD to PA survival in human blood was not attributable to differences in bacterial resistance to neutrophil phagocytosis, $H_2O_2$-mediated killing, the cytotoxicity of the recombinant full-length protein ($rCbpD_{PA}$) against immune cells (Supplementary Fig. 11b–d and Supplementary Results). CbpD expression did neither change in the presence of increasing concentrations of succinate (Supplementary Fig. 8f), an important LPS-controlled signaling metabolite produced by phagocytes (reviewed in ref. [53]).

While CbpD did not interact with 40 receptors associated with innate immune cell function (Supplementary Fig. 11e),

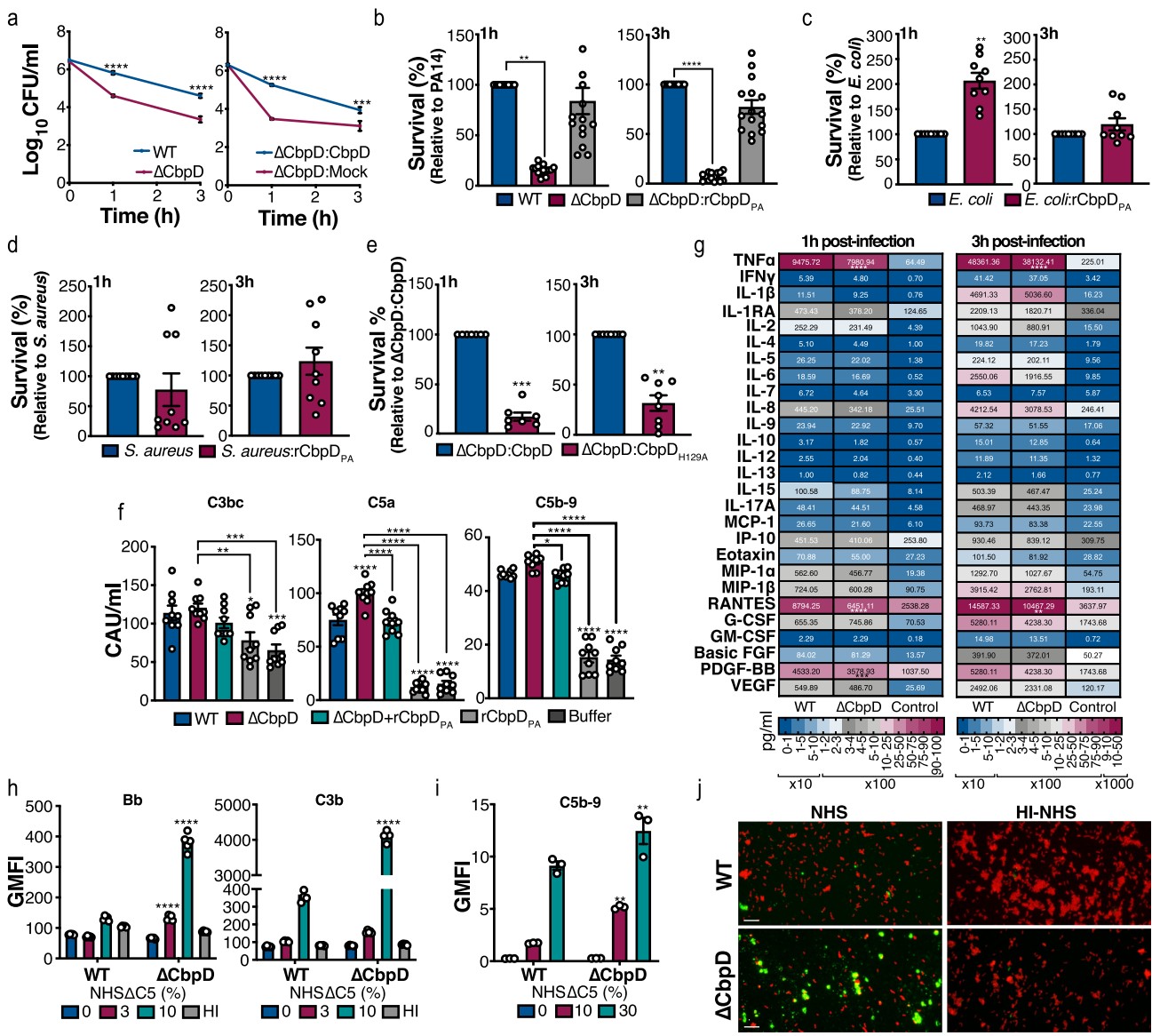

exogenous administration of rCbpD$_{PA}$ restored survival of the ΔCbpD mutant in human blood (Fig. 3b), suggesting that the secreted form of the protein may protect PA from immune clearance. To pursue this point further, we hypothesized that CbpD might also promote blood survival of other bacterial strains that lack LPMOs in their genome. According to the CAZy database[54], neither Gram-negative *E. coli* nor Gram-positive *Staphylococcus aureus* harbor LPMO-encoding genes in their genomes. We found that exogenous rCbpD$_{PA}$ promoted the survival of *E. coli* temporarily (1 h post infection) but not *S. aureus* in human blood (Fig. 3c, d), suggesting that the protective function of CpbD is neither restricted to the producing organism nor universal.

Lastly, to investigate whether the catalytic activity of CbpD is required for its role in promoting blood survival, we developed a construct encoding a catalytically inactive CbpD variant for in trans complementation of ΔCbpD (ΔCbpD:CbpD$_{H129A}$). The active site mutant ΔCbpD:CbpD$_{H129A}$ complemented strain had reduced whole blood survival compared to the WT protein ΔCbpD:CbpD complemented strain, indicating that CbpD catalytic activity is important for its protective role in blood survival (Fig. 3e).

### CbpD contributes to PA resistance to lysis by the terminal complement pathway.

The complement system plays a crucial role in killing Gram-negative bacteria through the assembly and subsequent insertion of the terminal membrane attack complex (MAC; C5b-9) in the bacterial cell envelope[55]. The importance of the complement system and the roles of C5 and MAC in defense against PA are well documented[56,57]. Complement activation is commonly measured by quantification of the activation products C3bc (proximal complement pathway), C5a, and C5b-9 (terminal complement pathway). Thus, the effect of CbpD on PA-induced complement activation was investigated by quantification of these products 30 min after the addition of WT PA (PA14) or ΔCbpD to human blood. Both strains significantly induced complement activation in whole blood as measured by high levels of C5a and fluid-phase C3bc compared to the buffer control (Fig. 3f). The addition of rCbpD$_{PA}$ to whole blood showed that the pseudomonal LPMO did not by itself trigger complement activation (Fig. 3f). CbpD did not inhibit the WT/ΔCbpD-induced proximal complement pathway activation in whole blood since C3bc concentrations were similar for both PA variants (Fig. 3f, left panel). The addition of WT PA to human blood resulted in lower levels of C5a compared to ΔCbpD and supplementation of

**Fig. 3 Ex vivo and in vitro analysis of the effect of CbpD deletion on PA virulence. a** Survival of PA WT, ΔCbpD, and ΔCbpD trans-complemented with a plasmid expressing CbpD or empty plasmid upon incubation in human blood (hirudin used as anti-coagulant). The data are plotted as the mean ± SEM, representing six (left panel)/three (right panel) experiments performed in triplicate. Data are analyzed by two-way ANOVA (Sidak's multiple comparisons). Left panel: 1 h $P = 2.1E-14$, 3 h $P = 5E-15$. Right panel: 1 h $P = 2.185E-12$, 3 h $P = 1.292E-4$. **b–d** Survival of PA WT (**b**), E. coli (ESBL) (**c**), and S. aureus (MRSA USA 300) (**d**) in human whole blood in the absence or presence of added rCbpD$_{PA}$ (20 μg ml$^{-1}$). Survival was calculated relative to inoculum. Untreated PA WT- (**b**), ESBL- (**c**), and MRSA- (**d**) infected blood was arbitrarily set equal to 100%, and bacterial survival in blood treated with rCbpD$_{PA}$ is represented as survival in percentage (%). The data are plotted as the mean ± SEM, representing five (panel **b**)/three (panels **c** and **d**) experiments performed in triplicate. Data are analyzed by two-way ANOVA (Tukey's multiple comparisons). **b** 1 h: WT vs ΔCbpD $P = 0.0030$, ΔCbpD: rCbpD$_{PA}$ vs ΔCbpD $P = 0.0022$; 3 h: WT vs ΔCbpD $P = 2.882E-7$, ΔCbpD:rCbpD$_{PA}$ vs ΔCbpD $P = 3.201E-5$, **c** 1 h: $P = 0.0020$; 3 h: $P = 0.898$. **e** Survival of ΔCbpD complemented with catalytically inactive CbpD (ΔCbpD:CbpD$_{H129A}$) or active CbpD in blood. The data are plotted as the mean ± SEM, representing $n = 7$ (1 h)/$n = 8$ (3 h) samples that were examined over three experiments. Data were analyzed by a two-tailed $t$ test (1 h: $P = 0.0001$, 2 h: $P = 0.0062$). **f** Quantification of soluble complement factors C3bc, C5a, and C5b-9/MAC in hirudin-treated human blood 30 min post infection with WT or ΔCbpD. When indicated, rCbpD$_{PA}$ was added together with the bacteria to a final concentration of 20 μg ml$^{-1}$. The results are given in complement arbitrary units (CAU) per ml. The data are plotted as the mean ± SEM, representing three experiments performed in triplicate. Data were analyzed by two-way ANOVA (Tukey's multiple comparisons). Left panel (C3bc)**:** WT vs buffer $P = 0.0008$, WT vs rCbpD$_{PA}$ $P = 0.0195$, ΔCbpD vs buffer $P = 0.0001$, ΔCbpD vs rCbpD$_{PA}$ $P = 0.0039$; Middle panel (C5a): WT vs ΔCbpD $P = 2.84096083E-7$, WT vs buffer 9.6E-14, WT vs rCbpD$_{PA}$ 9.6E-14, ΔCbpD vs ΔCbpD+rCbpD$_{PA}$ $P = 7.5720483E-8$, ΔCbpD vs buffer $P = 9.6E-14$, ΔCbpD vs rCbpD$_{PA}$ $P = 9.6E-14$; right panel (C5b-9): WT vs ΔCbpD $P = 0.0583$, WT vs buffer $P = 9.6E-14$, WT vs rCbpD$_{PA}$ $P = 9.6E-14$, ΔCbpD vs ΔCbpD+rCbpD$_{PA}$ $P = 0.0204$, ΔCbpD vs buffer $P = 9.6E-14$, ΔCbpD vs rCbpD$_{PA}$ $P = 9.6E-14$. **g** Cytokine profiling of whole human blood. Plasma harvested from blood samples infected with WT or ΔCbpD. The categorical heatmap shows the concentration of cytokines, chemokines or growth factors at 1 h and 3 h post infection. The data are depicted as the geometric mean of the cytokine values in each cell, representing three experiments performed in triplicate. Individual values are plotted in Supplementary Fig. 11j and 11k, in which the mean ± SEM is depicted. Data were analyzed by two-way ANOVA (Tukey's multiple comparisons) and the significant difference between WT and ΔCbpD is indicated by asterisks. Left panel (1 h): TNF $P = 5.282E-8$, RANTES $P < 1E-15$, PDGF-BB $P = 0.0007$; Left panel (3 h): TNF $P < 1E-15$, RANTES $P = 0.0029$. **h, i** Relative quantification of complement factors C3b and Bb (**h**) and C9 (**i**) bound to WT or ΔCbpD by flow cytometry. Bacteria were incubated with buffer or different concentrations of NHSΔC5, NHS or heat-inactivated serum (HI; 10 %) 45 min to 1 h prior to analysis. C3b, C5b-9, and Bb were detected using antibodies against C5b-9/C3b (both Alexa-fluor 488 labeled) or Bb following subsequent incubation with a secondary Alexa Fluor 488-conjugated antibody. The anti-C5b-9 binds to the C9 neoantigen of the C5b-9/MAC complex. The data are plotted as the geometric mean of fluorescence intensity (GMFI) ± SEM, representing $n = 5$ samples that were examined over two experiments (**h**) or three biological replicates (**i**). The gating strategy is provided in Supplementary Fig. 16. Data were analyzed by two-way ANOVA (Sidak's multiple comparisons) and the significant difference between WT and ΔCbpD is indicated by asterisks. **h** Bb: NHSΔC5 (3%) $P = 1.484E-9$, NHSΔC5 (10%) $P < 1E-15$; C3b: NHSΔC5 (10%) $P < 1E-15$; **i** NHS (10%) $P = 0.0021$, NHS (10%) $P = 0.0021$, NHS (30%) $P = 0.0029$. **j** Representative fluorescence microscopy images of C5b-9 complex deposition on PA WT or ΔCbpD upon incubation with 10% NHS or HI-NHS. The complex was detected using anti-C5b-9 followed by incubation with a secondary Alexa Fluor 488-conjugated antibody. Bacterial membranes were stained red using FM5-95. Two separate channels were merged into a single image by Fiji. The scale bar represents 10 μm. The imaging was performed twice. The significance is indicated by asterisks (*): *$P ≤ 0.05$; **$P ≤ 0.01$; ***$P ≤ 0.001$; ****$P ≤ 0.0001$. Source data are provided as a Source Data file (**a–h**).

rCbpD$_{PA}$ to ΔCbpD restored the C5a level to that observed for WT PA (Fig. 3f, middle panel). CbpD did not inhibit the formation of the WT/ΔCbpD-induced fluid-phase C5b-9, the terminal pathway end product (Fig. 3f, right panel). Of note, rCbpD$_{PA}$ slightly reduced the classical and alternative complement pathway activities when screened by measurement of the surface-bound C5b-9 complex (Supplementary Fig. 11f and Supplementary Results). Since C5a is a potent inflammatory mediator, we also quantified 27 key cytokines and chemokines in blood with PA added. Indeed, PA (WT/ΔCbpD) addition strongly induced several inflammatory cytokines (e.g. TNF, IL-1β, RANTES, PDGF-BB, IL-8) compared to control. Some of these cytokines were less prominent in blood with ΔCbpD added compared to WT PA, such as TNF and RANTES (1 and 3 h) and PDGF-BB (1 h only) (Fig. 3g and Supplementary Fig. 11j, k).

Cleavage of the C5 molecule to the C5a anaphylatoxin and the C5b MAC subunit is mediated by the proteolytic C5 convertase complex, which assembles on the bacterial surface as a result of complement activation[58]. We asked if the increase in C5a levels associated with the loss of CbpD may be linked to C5 convertase assembly. Indeed, incubation of WT and ΔCbpD PA with a dilution series of C5-depleted serum (ΔC5), showed higher labeling of ΔCbpD cells with alternative complement pathway C5 convertases components, C3b and Bb (Fig. 3h). This difference was not observed when the bacteria were pre-incubated with heat-inactivated ΔC5 serum or buffer (RPMI/HSA), which lacks the potential to deposit C3b and to convert the C5 molecule (Fig. 3h).

To further scrutinize the impact of CbpD on C5 convertase assembly, we hypothesized that lower convertase activity could lead to less deposition of MAC (C5b-9) on the bacterial surface. This can be quantified by using antibodies recognizing the C9 neoantigen of the C5b-9 complex[59]. Indeed, experiments with NHS (10 and 30%) showed higher C9 deposition on the ΔCbpD surface compared to WT PA (Fig. 3i) and on the complemented mutant (ΔCbpD:CbpD) compared to mock (Supplementary Fig. 11g). Next, we evaluated MAC deposition on the PA surface after incubation of bacterial cells with 10% NHS using fluorescence microscopy. C5b-9/MAC deposition was detected on the surface of the majority of ΔCbpD cells, but only to a limited extent in WT cells (Fig. 3j). Furthermore, a membrane integrity assay was used to assess outer membrane damage caused by MAC in WT PA, ΔCbpD mutant, and in trans-complemented constructs. No significant difference was observed between WT and ΔCbpD in the absence of serum, but exposure to 10% NHS resulted in a significant increase in permeability of the ΔCbpD mutant compared to WT PA or the ΔCbpD:CbpD complemented strain compared to ΔCbpD:Mock (Supplementary Fig. 11h). Transmission electron microscopy of serum-exposed bacteria showed predominantly intact cells with well-preserved cell envelopes for the WT PA strain (Supplementary Fig. 11i), whereas images of serum-exposed ΔCbpD showed cellular debris (amorphous materials) and marked morphological changes (e.g., loss of shape) indicative of cell death (Supplementary Fig. 11i). Taken together, these results show the importance of CbpD in protecting PA against bacterial lysis by the terminal complement pathway.

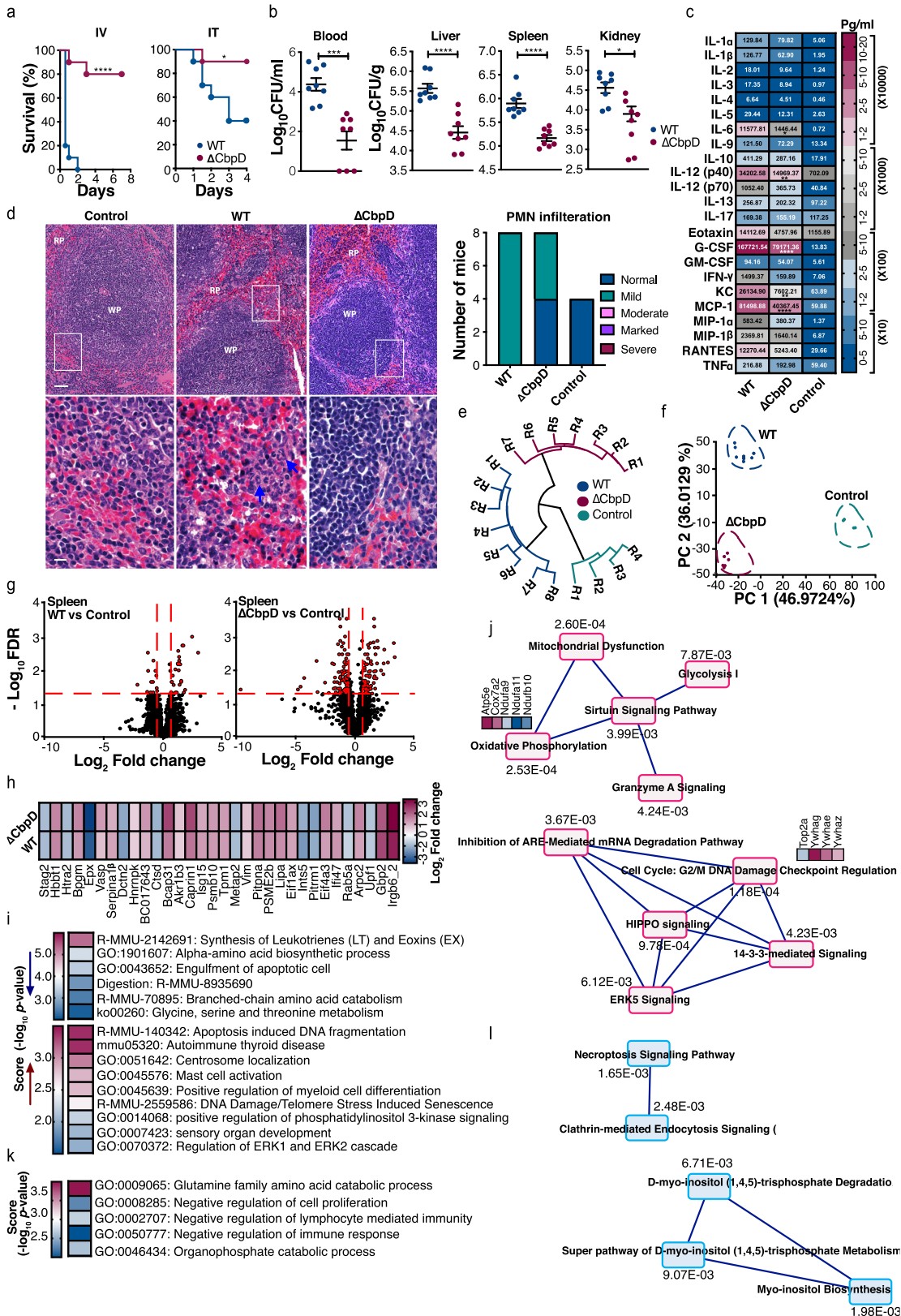

## CbpD contributes to PA virulence in murine systemic infection in vivo

Given the contribution of CbpD in resistance of PA to whole blood-mediated killing, we evaluated the impact of CbpD on PA systemic virulence in a murine intravenous (IV) challenge model. CD1 mice (8-week old) infected with $2 \times 10^7$ CFU/mouse WT PA experienced 100% mortality within 48 h post infection

(Supplementary Fig. 12b and Fig. 4a, left panel). In contrast to the universal mortality of mice challenged with the WT strain, 80% of isogenic mutant ΔCbpD-infected mice survived during the 48 h observation period (Fig. 4a, left panel). To investigate the effect of CbpD on bacterial clearance, mice were euthanized at 4 h post infection and the CFU in blood, liver, kidney, and spleen were

**Fig. 4 The effect of CbpD on PA pathogenesis in vivo. a** CD1 mice were inoculated intravenously (IV) with $2 \times 10^7$ CFU (left panel, 10 mice/group) and intratracheally (IT) with $5 \times 10^6$ CFU (right panel, 10 mice/group) PA WT or ΔCbpD per mouse. Survival is represented by Kaplan–Meier survival curves and analyzed by log-rank (Mantel–Cox) test (left panel: $P = 1.118E-5$, right panel: $P = 0.0250$). **b** Bacterial loads in the kidney, liver, spleen and blood (CFU/g for organs and CFU/ml for blood) of 8-week-old CD1 mice were enumerated 4 h post systemic infection (IV) with PA WT or ΔCbpD. The data are plotted as the mean ± SEM, representing one experiment performed with eight mice/group and were analyzed by two-tailed $t$ test (kidney $P = 0.0129$, blood $P = 0.0002$, liver $P = 5.827E-5$, spleen: $P = 1.910E-5$. Mock-infected mice ($n = 4$) were included as a control. **c** The categorical heatmap shows the concentration of cytokines, chemokines or growth factors in the serum of CD1 mice 4 h post infection (as described in **b**). The data are depicted as the geometric mean of the cytokine values, representing one experiment performed with eight mice/group. Mock-infected mice ($n = 3$) were included as a control. Individual values are plotted in Supplementary Fig. 12a, in which the mean ± SEM of each cytokine is depicted. The data were analyzed by two-way ANOVA (Dunnett's multiple comparisons test) and the significant difference between WT and ΔCbpD is indicated by asterisks (*). IL-6: $P = 0.0497$, IL-12 (p40): $P = 0.0018$, G-CSF $P < 0.0001$, KC: $P = 0.0063$, MCP-1: $P < 0.0001$. **d** Representative hematoxylin and eosin-stained sections of spleen tissues from infected (WT or ΔCbpD, $n = 8$ mice/group) and uninfected mice (control, $n = 4$ mice) collected 4 h post infection (as described in **b**), analyzed by light microscopy. White pulp (WP) and red pulp (RP) regions are marked. Polymorphonuclear (PMN) leukocytes are indicated with arrows. The scale bar represents 50 and 20 μm in the upper and lower panels, respectively. The histopathologic scoring analysis of PMN infiltration of the spleen tissue is shown as a histogram. **e** Hierarchical clustering of spleen proteome based on Spearman rank correlation coefficients. The infection status (as described in b) is indicated by color and R stands for replicate, (indicating number of mice/group). **f** Principal component analysis (PCA) of the identified proteins, showing segregation of the spleen proteomes into two different infected (WT or ΔCbpD) and one uninfected group. The quantified proteins are plotted in two-dimensional principal component space by PC1 = 46.9724% and PC2 = 36.0129%. **g** Volcano plot showing differentially abundant proteins in the spleens of WT- or ΔCbpD-infected mice relative to mock-infected mice. The red dotted line(s) through the y axis and x axis, indicate significance cutoff values at $q = 0.05$ ($\log_{10} = 1.3$) and ($+/-$) 1.5-fold change ($\log_2 = 0.58$) in protein abundance, respectively. Significance was determined using the two-tailed paired $t$ test. **h** Heatmap of $\log_2$ fold change values of the 32 significantly regulated proteins in infected (WT or ΔCbpD) vs. non-infected spleen. The proteins belong to the shared category. **i** Heatmap of enrichment score ($-\log_{10} P$ value, cutoff = 0.01) showing pathways/cellular processes that were enriched in the spleen proteome (Supplementary Data 10 and 11, unique category) of ΔCbpD-infected mice (down- or upregulated; arrows pointing down/up). Enrichment analysis was performed by Metascape and the $P$ value was calculated based on the cumulative hypergeometric distribution. **j** Depiction of the Ingenuity Pathway Analysis (IPA) of the unique proteome showing canonical pathway webs that are significantly altered in ΔCbpD-infected mice. The $P$ value was calculated by right-tailed fisher's exact test and is shown in the figure. Pathways without interaction with the web (singletons) are omitted from the visualization. The average fold change value of up- or downregulated proteins associated with some of the top-scored pathways are presented as heatmap and is visualized using a blue–white–red gradient. **k** Heatmap of enrichment score ($-\log P$ value, cutoff = 0.01) showing pathways/cellular processes that were enriched in the unique splenic proteomes (Supplementary Data 10 and 11) of WT-infected vs. uninfected mice Enrichment analysis was performed by Metascape and the $P$ value was calculated based on the cumulative hypergeometric distribution. **l** Depiction of the IPA of the unique proteome showing canonical pathway webs that are significantly altered in WT-infected mice. The $P$ value was calculated by right-tailed fisher's exact test and is shown in the figure. Pathways without interaction with the web (singletons) are omitted from the visualization. When applicable, the significance is indicated by asterisks (*): *$P \leq 0.05$; **$P \leq 0.01$; ***$P \leq 0.001$; ****$P \leq 0.0001$. Source data are provided as a Source Data file (**a–c**).

enumerated. Strikingly, the mice infected with ΔCbpD mutant had significantly reduced bacterial loads in blood and tissues compared to the mice infected with WT strain (Fig. 4b). Considering the significance of PA in respiratory infections, the effect of CbpD virulence was also studied in a murine intratracheal challenge model. WT PA infection of CD1 mice with $5 \times 10^6$ CFU/mouse experienced ~60% mortality within 4 days post infection while the ΔCbpD-infected mice had 10% mortality at the same time point (Fig. 4a, right panel). These data reveal an explicit requirement of CbpD for full PA virulence in these two in vivo models.

Serum cytokine/chemokine responses were profiled 4 h post bacterial systemic challenge. Significantly higher levels of IL-6, IL-12 (p40), G-CSF, keratinocyte chemoattractant (KC; a functional homolog of human IL-8/CXCL1), and MCP-1 (CCL2) were detected in mice challenged with WT PA compared to mice infected with the ΔCbpD mutant ($P < 0.05$) (Fig. 4c and supplementary Fig. 12a). This result suggests a heightened inflammatory response that tracks the increased bacterial burden seen in the fully virulent WT strain. Correspondingly, histopathological evaluation of splenic tissue samples revealed a slightly higher infiltration of neutrophils in WT-infected mice (Fig. 4d, left panel) compared to mock-infected (control) and ΔCbpD-infected mice (4 out of 8) (Fig. 4d, right panel).

Proteomics was used to gauge the effect of CbpD deletion on host-specific responses to PA systemic infection. The response of the spleens from CD1 mice infected with WT PA, ΔCbpD, or PBS (mock-infected) were analyzed. In total 2496, 2428, and 2271 proteins were quantified in WT-, ΔCbpD-, and mock-infected mice, respectively, and 2187 of these proteins were present

commonly across all treatments (Supplementary Data 7). Hierarchical clustering and principal component analysis revealed a distinct infectious agent-specific segregation of the splenic proteomes into WT, ΔCbpD, and mock-infected (control) clusters (Fig. 4e, f). The volcano plots of splenic proteomes showed 42 significantly regulated proteins in WT-infected vs. mock-infected mice (28 up- and 14 downregulated), whereas 160 differentially regulated proteins were detected in ΔCbpD-infected vs. mock-infected mice (77 up- and 83 downregulated; FC ≥ 1.5 and $q \leq 0.05$; Fig. 4g and Supplementary Data 8 and 10). Infected spleens shared 200 regulated proteins that had comparable relative expression irrespective of the infecting strain (Supplementary Data 8–11). The shared proteome consists of 32 significantly regulated proteins (FC ≥ 1.5, $q \leq 0.05$) (Fig. 4h and Supplementary Data 8 and 10) and 168 proteins that were only detected in the infected compared to control mice (no FC value, Supplementary Data 9 and 11). Functional analysis of this shared proteome (Supplementary Data 10 and 11) revealed high enrichment of several immune responses, including cellular response to IFNβ, RIG-I-like receptor signaling pathway, and Toll-like receptor signaling as a general response to PA infection (Supplementary Fig. 13a). Evaluation of the unique proteome signature associated with the deletion of CbpD revealed that 10 and 128 proteins were regulated (Supplementary Data 8 and 10) in response to WT PA or ΔCbpD infection, respectively (FC ≥ 1.5 and $q \leq 0.05$). In addition, 44 and 65 proteins were only detected in the WT and ΔCbpD-infected compared to control mice, respectively (Supplementary Data 9 and 11). The ΔCbpD associated unique proteome had a scattered expression with relative changes from +3.2 (Serpina3g) to −9.7 (Amy2)

compared to the control mice (Supplementary Data 8 and 10). String analysis revealed these proteins were highly interconnected (Supplementary Fig. 13b).

Functional exploration of the distinct markers (193 proteins, Supplementary Data 10 and 11) unique to the ΔCbpD-infected spleens showed enrichment of several cellular responses, including apoptosis-induced DNA fragmentation and autoimmune thyroid disease by the upregulated proteins, and synthesis of leukotrienes (LT) and eoxins (EX) and engulfment of apoptotic cells by the downregulated proteins (Fig. 4i). Inflammatory responses are well known to alter metabolism in host cells, which was corroborated by the observed enrichment of pathways associated with several terms coupled to metabolic processes such as "Glycine, serine and threonine metabolism", "Branched-chain amino acid catabolism", and "alpha-amino acid biosynthetic processes" in the downregulated proteins (Fig. 4i). Some of the proteins associated with the enriched cellular processes (e.g., isoforms of 14–3-3 proteins: Ywhae, Ywhag, Ywhaz) were localized in the central part of the STRING network analysis or clustered together (e.g. Hist1h1a, Hist1h1e, Hmgb2, Hmgb2) (Supplementary Fig. 13b). To further explore the interactions among regulated proteins (128) unique to the ΔCbpD-infected mice (Supplementary Data 10), an ingenuity pathway analysis (IPA)-based protein network was performed, and nine different disease-based and molecular networks were algorithmically generated (Supplementary Figs. 13c and 14). The "Inflammatory response, lipid metabolism, small-molecule biochemistry" was ranked (score 42) as the top enriched function- and disease-based protein network (Supplementary Fig. 13c). Several of the proteins associated with this network showed high connectivity to the extracellular signal-regulated protein kinases 1 and 2 (ERK 1/2) hub (Supplementary Fig. 14, network 1), which was also among the enriched pathways associated with upregulated proteins in ΔCbpD-infected mice (Supplementary Fig. 4i). On the single protein level, some of the upregulated proteins of this network are involved in immune responses e.g., C4b (representative of complement factor C4), Serpina3g, and H2-Ab1 (H-2 class II histocompatibility antigen). The other enriched networks with a score over 20 were "Cancer, Cell-To-Cell Signaling and Interaction, Cellular Movement", "Cancer, Cellular Assembly and Organization, Neurological Disease", "Cancer, Endocrine System Disorders, Protein Trafficking" and "Cancer, Molecular Transport, Organismal Functions" (Supplementary Figs. 13c and 14). Canonical pathway analysis of the unique proteome of the ΔCbpD-infected mice revealed significant enrichment of 15 different canonical pathways (Supplementary Fig. 13d), in which cell cycle: G2/M DNA damage checkpoint regulation (e.g., Top2a, Ywhae, Ywhag, Ywhaz), valine degradation I (e.g., Aldh1l1, Bcat2, Dbt), and oxidative phosphorylation (e.g., Atp5e, Cox72a, Ndufa11, Ndufa9, Ndufb10) were among the top three (Supplementary Fig. 13d and Fig. 4i). Several of these pathways were highly interconnected and formed two main canonical pathway webs that were associated with either metabolic processes or host immune responses (Fig. 4i).

Functional analysis of the regulated or detected markers (in total 54 proteins) (Supplementary Data 10 and 11) unique to the WT-infected spleens showed that these proteins were associated with negative regulation of immune responses (e.g., Arg1, Arg2, Ppp3cb, Clec2d, Cfh), negative regulation of cell proliferation (e.g., Cnn1, Hdac2, Pdcd4, Ddah1, Cers2), and several catabolic processes (Fig. 4i). IPA analysis of the regulated proteins unique to the WT-infected mice (Supplementary Data 10) revealed that 90% of the proteins were categorized under a network associated with "energy production, lipid metabolism, small-molecule biochemistry" (score 27) (Supplementary Fig. 13e). Several of these proteins (Cstb, Slc25a3, Clec2d, Pdcd4, Ppp3cb) were

directly or indirectly associated with the Myc hub, a master transcription factor of multiple proliferative genes (Supplementary Fig. 13e), which aligned with our enrichment analysis (Fig. 4i). Canonical pathway analysis of the unique proteome (Supplementary Data 10) showed necroptosis (e.g., Ppp3cb, SLC25A3) and myo-inositol biosynthesis (e.g., Impa1) as the profoundly enriched pathways in WT-infected mice (Fig. 4l).

Ultimately, analysis of the proteins involved in the complement cascade in the splenic proteome revealed the identification of several proteins associated with this system (e.g., *Cr2, Cfh, C4b (Slp), Cfb, C3, Vtn,* and *C1qbp*) (Supplementary Data 7). However, only the C4b molecule (representative of complement factor C4) (Supplementary Data 10) and complement factor H (*Cfh*) (Supplementary Data 11) were identified among the upregulated proteins unique to the ΔCbpD and WT-infected mice, respectively. Vitronectin (*Vtn*), which is involved in the regulation of the terminal complement cascade, was upregulated (Supplementary Data 11) in the shared proteome of infected vs. mock-infected mice.

Collectively, these data suggest that CbpD contributes to PA virulence in vivo and its loss leads to changes in the host proteome during systemic infection in vivo.

## Discussion

A predominant focus on the function and applications of LPMOs in biomass conversion has overshadowed the putative importance of these enzymes in bacterial virulence. The high prevalence of *lpmO* genes in pathogenic bacteria and their increased expression in multiple omics-type driven analyses in the context of host-pathogen interactions (reviewed in refs. [16,17]) suggest a potential biological function during infection. Our finding of the decline in bacterial survival in vivo in a murine model of PA systemic infection (Fig. 4), and in whole human blood ex vivo (Fig. 3) confirms that this LPMO can promote bacterial resistance to immune clearance, suggesting its function as a virulence factor. Aligned with our results, deletion of *lpmO* (lmo2467) in Listeria monocytogenes attenuated bacterial loads in the spleen and liver of mice[15].

The major impact of CbpD was reflected in our proteome studies that showed significant effects caused by *cbpD* deletion on both the bacterial and the host proteome (Figs. 2 and 4). The specific changes in the ΔCbpD proteome in response to a low concentration of NHS (which contains complement components), included downregulation of several metabolic processes and differential regulation of virulence factors, including an upregulation in the expression of several key regulators or transcriptional factors (e.g., AlgR, DksA, MvaT, AlgU, and Vfr). These changes reflect the struggle of the isogenic mutant to achieve physiological adaptation to the host environment and to overcome the complement-mediated stress (Fig. 2). Alteration in carbon metabolism and virulence has been suggested as an adaptive mechanism for PA to promote viability in human blood[60]. Despite the presence of strong crosstalk among virulence-associated regulators in PA[51], the proteome modulation strategy employed by the mutant was not sufficient to produce a functional impact in ΔCbpD defense during sepsis. This underscores the importance of CbpD in PA pathogenesis and adaptability during infection ranging from acute bacteremia to respiratory tract infection (Figs. 3 and 4).

To cope with sepsis, the systemic activation of the innate immune system triggers a variety of interconnected signaling pathways for a successful host response. Although WT- and ΔCbpD-infected mice showed similar trends in the enrichment of several pro-inflammatory pathways in their shared splenic proteomes (e.g., Toll-like, Rig-1-like, IFNβ), comparison of the

unique proteomes revealed that ΔCbpD and WT elicited different host immune responses in the infected mice. The differences in the splenic unique proteomes were reflected by the ΔCbpD-infected mice showing modulation of a larger number of proteins and enrichment of distinct cellular processes or pathways such as apoptosis, cell cycle, and ERK 1/2 pathways (Fig. 4). Indeed, the role of apoptosis, one of the top enriched pathways, as a means to aid the host during infection is well established (reviewed in ref. [61]) and is suggested to operate by removing the intracellular niches for selected pathogens[61]. Moreover, the cellular debris resulting from apoptosis can trigger activation of innate immune responses (reviewed in ref. [61]) such as the complement system, which subsequently enhances clearance of the infection (reviewed in ref. [62]). The different host immune response elicited by the ΔCbpD strain compared to the WT may reflect the hypovirulent nature of the ΔCbpD strain and underpins the contribution of CbpD significantly to the extent and consequence of PA-associated infection. From the perspective of understanding PA systemic infections in general, one important host marker unique to the WT-infected mice was the complement-negative regulator factor H (Supplementary Data 11). This observation indicates that the enrichment of cellular processes such as "negative regulation of immune responses" in the WT-infected mice may be partly responsible for the development of intense bacteremia observed for this strain. Importantly, increased expression of factor H during bacterial infection is associated with prolonged hospitalization of the patients[63].

Although PA strains show different levels of resistance to complement-mediated lysis[64], the importance of the complement system, particularly the membrane attack complex (MAC; C5b-C9), in the eradication of PA is well documented[65,66]. In addition, complement-deficient mice show an exacerbated inflammatory response[56] and high susceptibility to PA infection[56,57]. The role of MAC in killing/lysis of Gram-negative bacteria has been acknowledged/discussed in several ex vivo or in vivo studies, which have shown that this lytic macromolecular complex can clear the majority of the bacteria within minutes to hours, depending on the experimental setting and the bacterial strain (e.g., refs. [67–69] and reviewed in refs. [55,70]). Recently, it has been shown that assembly of C5 convertase on the bacterial surface[58] and direct attachment of C5b-7[71], a MAC precursor, to the bacterial outer membrane, is crucial for MAC-mediated bacterial lysis. While rCbpD did not entirely block complement activation, our data show that CbpD decreases C5a generation, assembly of the C5 convertase, and deposition of C9 and C5b-9 (MAC), which could explain the observed protective effect of the LPMO against complement-mediated killing in blood. Interestingly, rCbpD$_{PA}$ could also protect naturally LPMO-deficient Gram-negative pathogen E. coli from being killed in blood temporarily but did not affect the survival of S. aureus, which like other Gram-positive bacteria, is resistant to MAC lysis. The high conservation of the CbpD sequence, its prevalence among PA strains (Supplementary Fig. 1 and Supplementary Results), and the common occurrence of CbpD-like LPMOs in the genomes of several Gram-negative bacteria (Supplementary Fig. 2) suggest it may be fruitful to explore this family of enzymes as a determinant of resistance towards complement-mediated bacterial clearance.

While the association between CbpD and complement-mediated resistance is evident, we cannot exclude the effects of CbpD on outer membrane homeostasis in PA under environmental stress. Several proteins are involved in outer membrane homeostasis and LPS biogenesis/transport[44–48] in PA, and our proteomics studies showed that the deletion of cbpD led to decreased expression of several of these proteins (Fig. 2). Interestingly, the LPS inner glycan core composition of ΔCbpD showed a higher abundance of Kdo, possibly resulting from the altered expression of proteins involved in LPS transport and biosynthesis. Such structural changes may contribute to the higher susceptibility of the ΔCbpD outer membrane to antimicrobial host components, e.g., MAC. (as was indeed observed; Supplementary Fig. 10), and thus enhance MAC-mediated clearance. In support of this possible role of CbpD, it has been shown that several Gram-negative bacteria can resist MAC-mediated lysis by altering their surface properties (ref. [72] and references within).

The effect of CbpD on survival in blood ex vivo depends on its catalytic activity, as the catalytically inactive H129A variant could not complement cbpD deficiency (Fig. 3). Similar to all other biochemically characterized LPMOs from pathogenic bacterial species, CbpD degrades chitin by oxidation. Using chitin as a model substrate we noted that CbpD enzymatic performance was affected by other redox-related virulence factors that are secreted by PA, namely pyocyanin, possibly generating the LPMO co-substrate $H_2O_2$, and azurin, which could supply the LPMO with copper (Fig. 1). Considering the significance of copper in host nutritional immunity and bacterial physiology[73], pathogens have developed multiple strategies to overcome this nutritional challenge[73]. Azurin is an important component of a $Cu^{2+}$-scavenging pathway in PA, more specifically a type VI secretion system (T6SS)-mediated metal transport pathway, which is involved in sequestering of $Cu^{2+}$ from the environment[74]. Intriguingly, several existing observations connect CbpD with pyocyanin and azurin. First, the secretion of azurin and CbpD are regulated through the same post-transcriptional regulator, called Crc[75]. Second, pyocyanin, azurin, and CbpD are all regulated through quorum sensing[76]. Thus, given the CbpD-compatible concentrations of PCN measured in human blood (up to 130 μM)[77], and the presence of ascorbate in the blood[78], it is conceivable that these secreted compounds act in concert during PA systemic infection. Alternatively, $H_2O_2$ can also be provided through the oxidative respiratory burst produced by phagocytes during infection, an important mechanism of antimicrobial host defense[79]. Copper could also be acquired from the host by CbpD itself, as shown in a recent study of the fungal pathogen Cryptococcus neoformans, which utilizes an LPMO-like protein for copper acquisition during infection[19].

Since CbpD can degrade chitin, a polysaccharide not found in humans, it is pertinent to ask whether the enzyme can also be involved in nutrient acquisition outside the host. The genome of WT PA encodes a GH18 and a GH19 chitinase, which gives support to the latter hypothesis. Conversely, a study investigating chitin utilization by PA concluded that the bacterium could neither utilize chitin nor GlcNAc$_2$ the latter being the major degradation product of chitin hydrolysis[80]. On the other hand, several other important pathogens are also known to contain chitin-active LPMOs and chitinases, and some of these, for example, Vibrio cholerae[81], can utilize chitin as a nutrient source[82], which may be taken to suggest a role of the LPMO in nutrient acquisition. Of note, the genome of V. cholerae encodes two LPMOs. Based on functional studies, e.g., with the use of knock out strains in infection models, the somewhat enigmatic four-domain LPMO called GbpA, is considered a bacterial colonization factor related to mucosal colonization[11,13,83]. In light of our current work and considering the higher survival of ΔCbpD-infected mice in an intratracheal infection model in vivo, it could be that GbpA has dual virulence properties and contributes to immune evasion in a different context of infection, e.g., systemic disease, similar to that of CbpD. Intriguingly, some PA strains contain a second LPMO-encoding gene. For example, PA7 encodes genes for two LPMOs, cbpD and cbpE, being homologs of a PA14 CbpD and V. cholerae GbpA, respectively. Notably, the PA7 strain is suggested to have adapted alternative virulence

mechanisms compared to other PA strains[84]. In this context, it is noteworthy that the crystal structure of GbpA shows structural similarity to macroglobulin-like (MG) domains 2, 3, and 4 of the C5b molecule present in the C5b-C6 complex (Supplementary Fig. 5d). Such a similarity is not apparent for the CbpD model, which exhibits an elongated shape in solution, as also has been observed for GbpA by SAXS experiments[13]. The flexibility of these proteins, as e.g. observed in our SAXS, MD, and modeling data (e.g., Fig. 1) may allow the adoption of a U-shaped conformation like the GbpA crystal structure (Supplementary Fig. 5e). The similarity of virulence-related LPMOs to complement factors may indicate a role of these proteins related to molecular mimicry.

In conclusion, we show that deletion of the PA LPMO-encoding gene, *cbpD*, reduces bacterial survival during systemic infection by increased MAC-mediated killing of the bacteria. Comparative proteomic analyses revealed how the deletion of this single virulence factor modulates both bacterial and host responses upon infection, with effects extending to multiple proteins and pathways. Future research is needed to unravel the molecular mechanisms underlying CbpD-mediated immune evasion by PA. Our work also raises the possibility that LPMOs function as moonlighting enzymes with biological functions not restricted to polysaccharide degradation and significantly advances our understanding of this fascinating enzyme family.

## Methods

**Bacterial strains and cell lines**. *Pseudomonas aeruginosa* (PA) strains utilized in the study included wild-type (WT) PA14 (UCBPP-PA14), a human clinical isolate[85], and its isogenic mutant PA14Δ*cbpD* (ΔCbpD), provided by Prof. Hogan (Geisel School of Medicine at Dartmouth, USA). Clean deletion of ΔCbpD was confirmed by Illumina next-generation sequencing (MicrobesNG, http://www.microbesng.uk) at a minimum coverage of >50× and using 2 × 250 bp paired-end reads, and by comparing against the PA14 reference sequence (NC_008463), using the "Map to Reference" option in Geneious Prime 2019.0.4 (www.geneious.com). SNPs and deletions in the sequenced genomes compared to the reference genome were found using the Geneious Prime Find Variations/SNPs function. *E. coli* strains were routinely grown in Luria–Bertani broth (LB, BD Difco) at 37 °C. PA strains were grown at 37 °C in brain heart infusion (BHI, Oxoid) enriched medium or in LB at 200 rpm unless otherwise indicated. When needed, ampicillin (Sigma), kanamycin (Sigma), carbenicillin or (Alpha Aesar) were added at 100, 50, or 300 μg ml⁻¹, respectively, unless otherwise indicated. *Vibrio anguillarum* NB10 (serotype O1) (provided by Prof. Wolf Watz, Umeå University, Sweden) was isolated at the Umeå Marine Research Centre, Norrbyn, Sweden during a natural outbreak of vibriosis[86] and was grown in trypticase soy broth (TSB, Oxoid) at room temperature (RT). *Enterococcus faecalis* V583, a vancomycin-resistant clinical isolate[87,88], was grown in BHI at 37 °C. *S. aureus* USA300-MRSA (TCH1516, ATCC BAA-1717) and ESBL-producing uropathogenic *Escherichia coli*, recovered from urine, were grown in BHI and LB at 37 °C, unless otherwise stated.

Neutrophils were purified from the heparinized venous blood of healthy volunteers using 1-Step polymorphprep (Fresenius Kabi Norge AS) through gradient centrifugation (ethical approval information is provided in "human serum/blood"). Monocytes (THP-1) and HL-60 cell lines were purchased from ATCC. HL-60 were maintained in Iscove's Modified Dulbecco's Medium (IMDM) (Sigma), supplemented with 20% (v/v) fetal bovine serum (FBS) (Invitrogen Life Technologies), penicillin (100 units ml⁻¹), and 100 μg ml⁻¹ streptomycin (Biowest) in a CO₂ incubator (5% CO₂) at 37 °C. THP-1 cells were maintained in RPMI 1640 (Gibco) medium supplemented with 2 mM L-glutamine (Gibco), 10% (v/v) FBS, 4.5 g L⁻¹ glucose (Gibco), 10 mM HEPES (Gibco), 1.0 mM sodium pyruvate (Gibco) and 0.05 mM 2-mercaptoethanol (Sigma). Cells were maintained at density 4 × 10⁵–1 × 10⁶ ml⁻¹ and passaged every 4–5 days.

**Construction of complementation vectors**. Plasmids and primers used in this study are shown in Supplementary Table 5. The complementation constructs were cloned in the *E. coli–PA* shuttle vector pGM931[89] (provided by Assoc. Prof. Federica Briani, University of Milan, Italy). All pGM931 derivatives were replicated in and isolated from *E. coli* using the GeneJET plasmid isolation kit (Thermo Scientific) and transformed into ΔCbpD. To construct plasmids pGM-CbpD and pGM-CbpD-His₆ (containing a poly-histidine tag (His₆ tag) at the C terminus of the protein) the *cbpD* gene with its native promoter was PCR-amplified from PA14 (WT) genomic DNA with primers CbpD-KpnI-FW and CbpD-SbfI-RV/CbpD-SbfI-His₆-RV and cloned into pGM931 digested by KpnI and SbfI restriction enzymes (NEB), using either the In-Fusion HD cloning kit (Clontech) or double digestion following ligation approach. In addition to the primers used for

amplification, CbpD-Out-FW, CbpD-Int-RV and pGM-FW were used for plasmid verification purposes. The pGM-CbpD complementation vector containing the H129A mutation in the CbpD sequence was obtained with the absence (pGM-CbpD_H129A) or presence (pGM-CbpD_H129A-His) of a C-terminal hexahistidine tag, using a mutagenesis and cloning service provided by GeneScript.

**Cloning of CbpD in *E. coli***. Expression constructs for production of recombinant CbpD (rCbpD_EC) were created as follows: A codon-optimized gene for *E. coli* expression (GenScript) encoding *cbpD* (residue 1–389, UniProt ID; Q02I11), was amplified (Supplementary Table 5) for subsequent cloning into the pNIC-CH (Addgene) expression vector by ligation-independent cloning[90], which adds a hexa-histidine tag to the C terminus of the protein. For protein expression, the construct was transformed into One Shot® BL21 Star™ (DE3) *E. coli* cells (Invitrogen). Genes-encoding truncated versions of rCbpD_EC, AA10 (residues 1–210) and MX + CBM73 (residues 211–389) were generated by PCR using primer pairs rCbpD-pNIC-FW/rCbpD-AA10-pNIC-RV and rCbpD-MX/CBM73-pNIC-FW/rCbpD-pNIC-RV, respectively, and cloned into the pNIC vector as described above. The CbpD gene containing the H129A mutation was obtained using the GenScript mutagenesis service, from which the mutated gene (*cbpD*_H129A) was obtained. The mutated CbpD gene was cloned into the pNIC expression vector, as described above, yielding the rCbpD_EC-H129A and rCbpD_EC-pNIC vectors. For expression in *E. coli*, the vector was transformed into BL21 Star™ (DE3).

**Expression of recombinant CbpD variants in *E. coli* (rCbpD_EC) and PA (rCbpD_PA)**. Expression of all recombinant CbpD variants (rCbpD_EC, rCbpD_EC-H129A, AA10, and CBM73) was performed by the cultivation of *E. coli* BL21 Star (DE3) containing the relevant expression plasmid in Terrific Broth (TB) medium supplemented with 50 μg ml⁻¹ kanamycin in 1-L flasks at 22–25 °C using a LEX-24 Bioreactor (Harbinger Biotechnology). IPTG was added to a final concentration of 0.1–0.2 mM when the culture reached OD₆₀₀ = 0.5–0.6 (before transferring to a LEX-24 Bioreactor), and the induction lasted ~16 h. The expression of all recombinant CbpD his-tagged variants in PA ΔCbpD (rCbpD_PA and rCbpD_PA-H129A) was performed in LB medium supplemented with 300 μg ml⁻¹ carbenicillin in 1 L flasks at 22–25 °C using a LEX-24 Bioreactor (Harbinger Biotechnology). For induction of the $P_{BAD}$ promoter in the pGM931-based vectors, arabinose (Sigma) was added to the bacterial culture to a final concentration of 6.6 mM, followed by incubation for ~16 h.

**Purification of rCbpD and truncated variants**. All rCbpD_EC and rCbpD_PA variants were purified using a periplasmic purification approach, except rCbpD_EC (MX + CBM73; see below for details). For periplasmic purification, cells were harvested by centrifugation for 10 min at 8000 × g. The periplasmic fraction containing the mature protein was extracted from the harvested cells by osmotic shock[91]. The resulting extracts were centrifuged for 15 min at 22,000 × g and passed through a 0.22 μm filter prior to protein purification. Periplasmic extracts were loaded onto a 5 ml HisTrap™ High-Performance column (GE Healthcare) connected to an ÄKTA purifier FPLC system (GE Healthcare), and purification was performed based on the manufacturer's instructions.

To purify rCbpD_EC (MX + CBM73), the culture was harvested by centrifugation at 8000 × g and resuspended in lysis/binding buffer (20 mM Tris-HCl pH 8.0) followed by cell disruption by sonication using a Vibra cell Ultrasonic Processor (Sonics). Cell debris was removed by centrifugation at 8000 × g and the crude extract was passed through a 0.2-μm filter. Cytoplasmic extracts were loaded onto a HisTrap™ High-Performance column (GE Healthcare) connected to an ÄKTA Purifier FPLC system (GE Healthcare), and purification was performed based on the manufacturer's instructions.

Impurities were removed by gel filtration using an Äkta purifier chromatography system operating a HiLoad 16/6016/600 Superdex 75 size-exclusion column (GE Healthcare). The flow rate was set to 1 ml/min and the eluent used was 15 mM Tris-HCl/150 mM NaCl pH 7.5. Fractions containing pure full-length or truncated variants of rCbpD (evaluated by SDS-PAGE) were pooled, concentrated, and buffer exchanged to 15 mM Tris-HCl/150 mM NaCl pH 7.5, using a Vivaspin 20 (10-kDa molecular weight cutoff) centrifugal concentrators (Sartorius Stedim Biotech GmbH). Protein purity was estimated to be over 95% for all the enzymes using SDS-PAGE. Protein concentrations were either determined using the Bradford assay (Bio-Rad) or by absorbance at A280, using the theoretical extinction coefficient (http://web.expasy.org/protparam/). The concentrations of purified proteins were always controlled by the latter method before employment of the protein in any assay used in this study.

**Computational details of molecular dynamics (MD) simulations**. A 3D homology model of CbpD was built from amino acid residues 26–389, using the RaptorX server[92,93]. The most likely protonation state of each titratable amino acid side chain in the RaptorX model of CbpD was assessed using the H + + server[94] and the N-terminal His26 and His129, which constitute the histidine brace, was assigned the HIE and HID protonation states, respectively. The cysteine pairs Cys39/Cys52, Cys89/Cys204, Cys160/Cys176, and Cys358/Cys367 appeared to form disulfide bridges and were renamed to CYX. Finally, a copper ion was added to the histidine brace, and this model will be referred to as no-PTM. To generate models of post-translationally modified CbpD, amino acid residues Ser, Thr, and

Tyr, which had experimentally been identified[25,35] as phosphorylation sites (Fig. 1a, b), were substituted with the modified residues. In the model named PTM-1, these residues were replaced by S1P, T1P, and Y1P (–HPO$_3^-$), and in PTM-2 by SEP, TPO, and PTR (–PO$_3^2$).

The protein force field ff19SB[95] and a histidine brace-Cu(I) force field[96] were applied to all three models when preparing the models for MD simulations. To describe the phosphorylated residues, the phosaa10[97] force field was applied to the PTM-1 and PTM-2 models. Sodium ions were added to the models to obtain charge neutrality, and 38 additional sodium and chloride ions were added to the no-PTM model to compensate for the higher number of charges in the phosphorylated models. All three models were solvated in a truncated octahedron of OPC water[98] with a minimum distance of 12 Å to the boundary, resulting in models consisting of about 275,000 atoms.

These models were subjected to the first round of 3000 steps of energy minimization with 10 kcal mol$^{-1}$ Å$^{-2}$ positional restraints on all but the hydrogen atoms before continuing the second round of 3000 steps with 10 kcal mol$^{-1}$ Å$^{-2}$ positional restraints on Cα-atoms only using sander in the AmberTools19[99]. Then all systems were heated to 300 K in the NVT ensemble over a period of 40 ps. Density equilibrations were performed at 300 K for 0.5 ns at a constant pressure of 1 atm using the Berendsen barostat with a pressure relaxation time of 1 ps. In the final 100 ns equilibration step, performed in the NVT ensemble using the weak coupling algorithm and a time constant of 10 ps to regulate the temperature, the center of mass (COM) distance of AA10 and CBM73 modules was restrained to 100 Å using a force constant of 5 kcal mol$^{-1}$ Å$^{-2}$. Following the equilibration of the no-PTM, PTM-1, and PTM-2 models, the simulations were continued for another 525 ns without any restraints, accumulating 262500 snapshots in each trajectory. In all simulations, we used a time step of 2 fs, periodic boundary conditions with a 10 Å cutoff for nonbonded interactions, and particle mesh Ewald (PME) treatment of long-range electrostatics[100]. Hydrogen atoms were constrained by the SHAKE algorithm. Simulations were performed using the CUDA version of PEMEMD included in AMBER18[101]. Analysis of production trajectories was performed using the *cpptraj* module included in AmberTools19[99] and in-house Python scripts.

**Small-angle X-ray scattering**. Small-Angle X-ray Scattering analysis was performed at beamline BM29[102] at the European Synchrotron Radiation Facility (ESRF) in Grenoble, France (wavelength, $\lambda = 0.992$ Å). Scattering intensities were recorded as a function of the scattering vector $q = (4\pi/\lambda) \sin \theta$, where $2\theta$ is the scattering angle and $\lambda$ the wavelength. Data were collected in ten frames of 1 s at 20 °C in the q-range 0.00449–0.51787 Å$^{-1}$ for a concentration series of 2.50, 1.25, 0.63, and 0.31 mg ml$^{-1}$ rCbpD$_{EC}$ in a buffer containing 15 mM Tris-HCl pH 7.5 and 150 mM NaCl. Likewise, the buffer itself was measured for 10 s under the same conditions. Complementary experiments at 24 °C and 37 °C were performed in-house at the University of Oslo on a Bruker NanoStar instrument (RECX). The scattering intensities were corrected for electronic noise, empty cell scattering, and detector sensitivity. Intensities were calibrated to absolute units with H$_2$O scattering as standard, and scattering contribution from the buffer was subtracted using the beamline software BsxCuBE[102]. The data were then extrapolated to zero solute concentration using the *ALMERGE* software[103], and the extrapolated data were used for the subsequent data analysis. A summary of the data collection statistics and analysis is given in Supplementary Table 2 (prepared according to the guidelines suggested in ref. [104]) and Supplementary Fig. 4. The BioSAXS data were deposited in the SASBDB with accession code SASDK42, and the home-source SAXS data and statistics are accessible via SASBDB codes SASDJQ5 (24 °C) and SASDJR5 (37 °C). Pair-distance distribution function (from inverse Fourier-transformation[105]), the radius of gyration (from Guinier analysis[106]), and Porod volume (from the Porod Debye-function[107,108])) were calculated using tools in *PRIMUS*[109] from the *ATSAS* package[110]. 20 low-resolution models were calculated using ab initio shape determination using the *DAMMIF* software[111] and an average model was created with the *DAMAVER* software[112], and refined with *DAMMIN*[113]. *DAMMIF*, *DAMAVER* and *DAMMIN* are all part of the *ATSAS* package[110]. The RaptorX model (Fig. 1b) was fitted to the SAXS data using *Pepsi-SAXS*[114], with the three domains modeled as rigid bodies. The resulting atomistic model was subsequently superimposed with the ab initio *DAMAVER* model using *SUPCOMB*[115] (Fig. 1c).

**CbpD activity assay**. The activity of CbpD on β-chitin was evaluated as described previously with modifications[81], using metal-free water in all steps and preparations of buffers. Purified rCbpD was saturated incubation with a threefold to fivefold molar excess of copper (Cu(II)SO$_4$) in sodium phosphate buffer (50 mM, pH 6.0) for 30 min at RT. Excess copper was removed by passing the enzyme solution through a PD MidiTrap G-25 column (GE Healthcare), and the first 1 ml (out of 1.5 ml) of the protein fraction was harvested to ensure no copper contamination. Thereafter, the protein was buffer exchanged to 15 mM Tris-HCl/150 mM NaCl pH 7.5. Solutions containing copper-saturated rCbpD were stored for up to 7 days at 4 °C. CbpD activity was assessed by incubating 10 mg ml$^{-1}$ β-chitin (France Chitin) with 1 μM of the protein in 200 μl reactions buffered by 20 mM Tris-HCL pH 7.0 in the presence of 1 mM ascorbate (Sigma) as an external electron donor unless otherwise indicated. Additional electron donors including NADH (Sigma; 1 mM end concentration) and pyocyanin (Sigma; 10, 100, 1000 μM end concentration), the latter combined with ascorbate (250 μM end

concentration) in some reactions when indicated. In reactions set up to probe putative copper transfer from azurin to CbpD devoid of copper, the activity of CbpD was evaluated by reactions where 10 mg ml$^{-1}$ β-chitin was incubated with 1 μM of apo-CbpD (CbpD devoid of copper) in 200 μl reactions buffered by 20 mM Tris-HCL pH 7.0. This reaction was supplemented with pyocyanin (10, 100, and 1000 μM), 250 μM ascorbate, in the presence or absence of 1 μM purified azurin from PA (Sigma, dissolved in metal-free water). Generation of apo-CbpD was performed by incubating the LPMO (10 μM) with 2 mM EDTA in 15 mM Tris-HCl/150 mM NaCl pH 7.5 overnight, followed by addition of 5 mM MgCl$_2$ (Sigma), 15 min of incubation at room temperature, and finally passing the protein solution through a PD MidiTrap G-25 column, where the first two thirds (i.e., 1 ml) of the protein eluate was collected. The solution containing apo-CbpD was stored at 4 °C until use. All enzyme reactions were incubated at 37 °C in an Eppendorf Comfort Thermo-mixer at 800 rpm. Samples were harvested at the indicated time point (e.g., 0.5 or 2 h), centrifuged at 10,000 × g, and the supernatant was immediately filtered using a 0.22-μm filter. Reaction products were analyzed by UPLC (Agilent) using a 150-mm long hydrophilic interaction chromatography (HILIC) column (Acquity UPLC BEH amide, 1.7 μm). The sample injection volume was 5 μl, and the flow rate 0.4 ml min$^{-1}$. The gradient was as follows: 74% ACN (A), 26% 15 mM Tris-HCl pH 8.0 (B) held for 5 min, followed by a 3 min gradient to 62% A, 38% B. Column reconditioning was obtained by a 2 min gradient back to initial conditions (74% A, 26% B) and subsequently running at these conditions for 2 min. For ex vivo analysis, rCbpD$_{PA}$ was used "as is" without copper saturation. All HPLC data were collected and analyzed using the Chromeleon software (version 7.2.9; Thermo Scientific).

**CbpD-binding assay**. The binding assay was performed by incubating 10 μM rCbpD$_{PA}$, rCbpD$_{PA-H129A}$, AA10 (AA10 module only), and MX + CBM73 (the Module X and CBM73 domains only) with 10 mg·ml$^{-1}$ ß-chitin in 20 mM Tris-HCL pH 7.0. Reactions were incubated at 22 °C in an Eppendorf Comfort Thermo-mixer at 800 rpm. Samples were harvested at various time points, centrifuged at 16,000 × g, and 2 μl of the sample was used for determination of the protein concentration using Nanodrop (A280). The amount of protein in the samples was presented as percent relative to the control sample, the latter arbitrarily set to represent 100% unbound.

**Glycan microarray analysis**. Recombinant CbpD (rCbpD$_{PA}$ and rCbpD$_{EC}$), were screened for glycan binding against a library of 585 natural and synthetic mammalian glycans (version 5.2) in 15 mM Tris-HCl pH 7.5, 150 mM NaCl at final concentrations of 5 or 50 μg/ml by Core H (Consortium of Functional Glycomics (http://www.functionalglycomics.org), as described previously[116]. The CbpD variant purified from PA (rCbpD$_{PA}$) was used without the addition of copper (the activity of the protein was verified before shipping), whereas rCbpD$_{EC}$ was screened in the presence of 25 μM Cu(II) SO$_4$. The glycans, carrying an amino linker, were printed onto chemically modified (*N*-hydroxysuccinimide-activated) glass microscope slides. The results are presented as average relative fluorescence units (RFU) (Supplementary Data 1).

**Microscale thermophoresis (MST)**. MST analysis was performed using a Monolith NT.115 system (NanoTemper Technologies). Thus, 50 nM of His-tagged rCbpD$_{PA}$ was labeled using the Monolith NT His-Tag Labeling Kit RED-tris-NTA, according to the supplier's protocol (NanoTemper Technologies). All measurements were conducted in standard capillaries (NanoTemper Technologies) at 25 °C using 60% LED power and 20%, 40%, or 60% MST power. Laser on and off times was set at 30 s and 5 s, respectively. Hexa-*N*-acetyl-chitohexaose (NAG$_6$) (Megazyme) was used at a final concentration of 1 mM.

**Bacterial growth curves**. PA14 and ΔCbpD were grown overnight in 5 ml LB. The next day, bacteria were washed in PBS, adjusted to OD$_{600}$ 0.4. Thereafter, the bacteria diluted 1:100 in BHI, LB, RPMI (without phenol red) supplemented with 10% heat-inactivated FBS. When indicated, WT PA and ΔCbpD were grown in M9 (supplemented with 2 mM MgSO$_4$ and 0.1 mM CaCl$_2$), where succinate (Millipore) replaced glucose as the sole carbon source at final concentrations of 5 μM and 40 mM. The bacteria were transferred to a 96-well microtiter plate (Corning Inc.) in a total volume of 150 μl. Growth was monitored at 37 °C by measuring OD$_{600}$ every 15 min using a Synergy H1 Hybrid Reader (BioTek) or a Varioskan$^{TM}$ LUX multimode microplate reader (Thermo Scientific). Shaking was performed 15 s before each reading. The medium control was included as background.

**Hydrogen peroxide killing assay**. WT PA (PA14) and ΔCbpD were grown overnight in LB. The next day, bacteria were diluted 1:100 and re-grown in LB to the mid-exponential growth phase. Bacteria were washed in PBS and incubated with different concentrations (0, 1, 10, 100, and 1000 mM) of H$_2$O$_2$ in RPMI supplemented with human serum albumin (HSA) at 37 °C for 10 min in a 96-well microplate at a final volume of 200 μl. Bacterial survival was evaluated by serial dilution with subsequent plating on LB/BHI agar plates, followed by overnight incubation at 37 °C and counting of colonies.

**Normal human serum (NHS) preparation and reagents**. To prepare pooled NHS, blood was drawn from several healthy volunteers (see "Ethical approval") and allowed to clot for 10–15 min at RT. After centrifugation (10 min, 2000 × g), serum was immediately collected, pooled, and stored at −70 °C. Heat-inactivated (HI-NHS) serum was obtained by incubating serum for 30 min at 56 °C in a water bath. Normal human serum and sera deficient for complement components were also purchased from Complement Technology. Purified recombinant *Ornithodoros moubata* complement inhibitor (OmCI)[117] was provided by Prof. Rooijakkers (University Medical Center Utrecht, the Netherlands).

***cbpD* expression profile**. The expression of *cbpD* in PA14 (WT) was assessed using droplet digital PCR™ (ddPCR™, Bio-Rad). The expression of *cbpD* was assessed during growth in LB and M9$_{PA}$, which is M9 medium (Gibco) supplemented with 0.5% (w/v) Casamino Acids (Difco), 0.2% (w/v) glucose (Sigma), 2 mM MgSO$_4$ (Sigma), and 0.1 mM CaCl$_2$ (Sigma). An overnight culture of WT was diluted 1:100 in M9$_{PA}$ and incubated at 37 °C under shaking conditions. Upon reaching OD$_{600}$ = 0.6, normal human serum (NHS, 10% v/v) was added to the bacterial cultures (M9$_{PA}$) and incubation was continued for 30 min. To gain further insight into the effect of serum on the expression of *lpmO* genes in other pathogens, overnight cultures of *V. anguillarum and E. faecalis*, grown in TSB and BHI, were diluted in M9$_{VA}$ (M9 medium with 1% NaCl (w/v, Sigma), 0.2% (w/v) Casamino Acids, and 0.5% (w/v) glucose) and LM17$_{EF}$ (1.25 g l$^{-1}$ Matrix Fish Peptone (Sigma), 1.25 g l$^{-1}$ Bacto yeast extract (BD Difco), 0.25 g l$^{-1}$ ascorbic acid (Sigma), 0.125 g l$^{-1}$ MgSO$_4$ (Sigma), 9.5 g l$^{-1}$ disodium glycerolphosphate (Sigma), 0.025 g l$^{-1}$ MnSO$_4$ (Sigma), respectively. Upon reaching OD$_{600}$ = 0.6, the bacterial cultures were supplemented with 10% (v/v) pooled salmon serum (SS), and the cultures were incubated for another 30 min. Salmon serum was obtained from 5 *Salmo salar* (~1–1.5 kg in size), which were kept at the fish laboratory belonging to the Norwegian University of Life Sciences. The fish were killed by a sharp blow to the head before blood sample collection in non-heparinized tubes. Cells were harvested, immediately resuspended in RNA protect solution (Qiagen), and further processed for gene expression analysis. RNA extraction was performed using the RNeasy mini kit (Qiagen) with an initial lysis step in TE-buffer containing lysozyme (Sigma). The samples were further treated with HL-dsDNase digestion (ArcticZymes) according to the manufacturer's instructions. RNA integrity and quantity were determined by Nanodrop. Reverse transcription of the total RNA was performed on 100 ng of RNA using the iScript™ Reverse Transcription (RT) Supermix (Bio-Rad) according to the manufacturer's recommendations. Droplet digital PCR analysis was performed using EvaGreen ddPCR™(Bio-Rad) according to the manufacturer's recommendations. The analysis was conducted using *cbpD*, *lpmO$_{VA}$*, and *lpmO$_{EF}$* detection primers (Supplementary Table 5). In each assay reactions containing either no template or no reverse transcriptase, controls were included as negative controls.

**Analysis of the bacterial proteome**. Overnight cultures of WT and ΔCbpD in LB were diluted in LB, RPMI/HSA (0.05%) and RPMI/HSA (0.05%) supplemented with 1% human serum, in triplicate, followed by incubation at 37 °C with shaking. To support bacterial growth, 10% LB was added to the RPMI cultures. Samples were harvested at the exponential phase. Thereafter, 1 × PhosSTOP™ (Roche), 1 mM phenylmethylsulfonyl fluoride (PMSF, Sigma), and 1× Complete Mini EDTA-free protease inhibitors (Roche) were immediately added to the samples. Cell pellets and supernatants were separated by centrifugation (4500 × g, 15 min, 4 °C). The supernatant was concentrated using 3000 kDa column (Amicon) (20x). The bacterial cell pellet was resuspended in 20 mM Tris-Cl (pH 7.5), 0.1 M NaCl, 1 mM EDTA, 1× Complete Mini EDTA-free protease inhibitors, and lysozyme (0.5 mg·ml$^{-1}$) to a concentration of ~2 × 10$^{10}$ colony-forming unit (CFU)/ml (through OD600 measurement) and cells were disrupted by sonication (20×, 5" off-5" on, 27% amp) under ice-cold conditions, the cellular debris was removed by centrifugation (4500 × g, 30 min, 4 °C).

Proteins were run on SDS-PAGE (10% Mini-PROTEAN gel, Bio-Rad Laboratories) for 1 cm and then stained with Coomassie brilliant blue R250. The gel was cut into several slices, proteins were reduced, alkylated, and in-gel digested as described previously[118]. The peptides were solubilized in 0.1% (v/v) trifluoroacetic acid (TFA) and desalted using C$_{18}$ ZipTips (Merck Millipore), according to the manufacturer's instructions. The peptide mixtures were analyzed using a nano HPLC-MS/MS system, as described previously[118]. The system was composed of a Q-Exactive hybrid quadrupole–orbitrap mass spectrometer (Thermo Scientific) equipped with a nano-electrospray ion source. Mass spectral data were acquired using Xcalibur (v.2.2 SP1). MS raw files were processed with the MaxQuant software suite (version 1.6.3.3.) for label-free quantification (LFQ) and identification of proteins[119]. Trypsin was set as a proteolytic enzyme, and two missed cleavages were permitted. The "match between runs" feature of MaxQuant was applied with default parameters. Data were filtered to a 1% peptide and protein level false discovery rate (FDRs) using a target-decoy reverse database approach[120].

To determine significantly changing proteins between isogenic mutant, ΔCbpD, and WT (PA14) strains, a pairwise *t* test was performed between each pair of proteins (Supplementary Data 2). The variance was assessed by an *F* test to ensure the correct statistical assumptions were used. To account for multiple comparisons, Benjamini–Hochberg correction was applied with a false discovery rate of 5%. Differentially abundant proteins were defined by having q values ≤ 0.05 and fold

change ≥1.5 (log$_2$ = 0.58) (Supplementary Data 3). In addition, proteins that were identified in either the ΔCbpD or WT strain, but not in the other, were also deemed significant. These proteins do not have an associated fold change or q value, rather they are annotated as either "up" or "down" regulated in the ΔCbpD strain compared to WT (Supplementary Data 4). Using the list of significantly up- and downregulated proteins (Supplementary Data 3 and 4), hypergeometric enrichment was used to calculate KEGG pathways that are enriched in the ΔCbpD and WT strains. The P values from the hypergeometric calculation were subjected to FDR correction (Benjamini–Hochberg), and pathways with q values ≤ 0.05 were defined as enriched. An enrichment score was calculated for each of these enriched pathways. Principal component analysis (PCA) was used to reduce the dimensionality of the multivariate data to two principal components so that the data clusters could be visualized graphically. To do this, K-nearest neighbors (kNN) imputation was implemented to fill in missing values when possible. If values for all replicates of a sample were missing, the values were replaced with the minimum detected value for that replicate. To further investigate strain and protein grouping, agglomerative hierarchical clustering was performed (Supplementary Data 5). MATLAB R2018b was used for all plots and analyses.

The effect of succinate (Millipore) on CbpD expression levels was assessed using label-free quantitative proteomics. An overnight culture of WT (PA14) was diluted and re-grown in M9 supplemented with 0.2% (w/v) glucose (Sigma), 2 mM MgSO$_4$ (Sigma), 0.1 mM CaCl$_2$ (Sigma). Upon reaching OD$_{600}$ = 0.4, succinate was added to the samples at the final concentration of 0, 0.5 μM, 5 μM, 5 mM, and 50 mM. The samples were incubated at 37 °C with shaking and harvested upon reaching OD$_{600}$ = 0.7–0.8. Thereafter, 1 mM PMSF and 1× Complete Mini EDTA-free protease inhibitors were immediately added to the samples. Cell pellets and supernatants were separated by centrifugation (4500 × g, 15 min, 4 °C). The bacterial pellet was processed and analyzed by proteomics methods, as described above.

For the analysis of PTMs on CbpD, purified rCbpD$_{PA}$ was stored in 20 mM Tris-HCl pH 8, reduced with DTT (Sigma; 10 mM), and subsequently alkylated with iodoacetamide (Sigma; 15 mM) in the dark for 30 min at room temperature. Digestion was performed with an enzyme/protein ratio 1:40 using trypsin or chymotrypsin overnight at 37 °C. The reaction was terminated with TFA (Fluka) to a final concentration of 1%. The peptide was desalted using C18 ZipTips (Merck) according to the Manufacturer's recommendations and analyzed with mass spectrometry as described above. For the analysis of PTMs on CbpD, the PEAKS software[121] was used, using both trypsin and chymotrypsin as enzymes and with semi-specific cleavage allowing for three missed cleavages. Carbamidomethylation was set as a fixed modification while phosphorylation (STY), acetylation (K), and methylation (KR) were allowed as variable modifications. In addition, we included oxidation of methionines and deamidation of asparagines and glutamines; however, these were only included for peptide coverage and not for PTM analysis. Results were filtered to 0.1% FDR ($-_{10}$log P > 32.9) and only modified peptides with AScore > 20 were considered valid.

**Bacterial survival assays in blood**. The viability of PA in whole human blood was assessed as described previously[122] with modifications. The bacteria were grown in LB washed twice with 1× PBS and resuspended in RPMI/HSA. Blood from healthy donors was collected in tubes containing hirudin (SARSTEDT or Roche) and was mixed (160 μl) with PA (~2–4 × 10$^6$ CFU ml$^{-1}$), MRSA USA300 (~3 × 10$^6$ CFU ml$^{-1}$) or *E. coli* ESBL (~3 × 10$^6$ CFU ml$^{-1}$) in RPMI 1640 (Gibco, Life Technologies, UK) containing 0.05% human serum albumin (HSA) (Sanquin/provided by van Sorge) (RPMI/HSA). When indicated, rCbpD$_{PA}$ at a final concentration of 20 μg ml$^{-1}$ was added to the samples. Infected samples (final volume of 200 μl) were incubated for 1–3 h at 37 °C on a rotator. Subsequently, the blood cells were lysed by adding 800 μl of H$_2$O (ice-cold) supplemented with 0.3% (w/v) saponin (Sigma), vortex vigorously, and immediately plated. For MRSA a 10 min water bath sonication step after lysing was included. Bacterial survival was evaluated by serial dilution with subsequent plating on LB/BHI agar plates, followed by overnight incubation at 37 °C and counting of colonies.

**Whole-blood phagocytosis**. For fluorescent labeling, bacteria were grown in 5 ml LB, washed twice with 1× PBS followed by resuspension in 5 ml 0.1 N sodium carbonate (Sigma) containing 0.5 mg ml$^{-1}$ fluorescein isothiocyanate (FITC; Sigma) for 1 h at room temperature and protected from light. Bacteria were washed extensively in PBS (three times), resuspended in RPMI/HSA to an OD$_{600}$ of 1.0, and re-diluted five times in RPMI/HSA. After the labeling reaction, 20 μl FITC-labeled WT or ΔCbpD were incubated for 20 min at 37 °C with freshly isolated human blood (160 μl) containing hirudin to prevent coagulation. When indicated, rCbpD$_{PA}$ at a final concentration of 20 μg ml$^{-1}$ was added to the samples. The phagocytosis reaction was stopped using a fluorescence-activated cell sorter lysing solution (BD Biosciences). The samples were washed twice with RPMI/HSA, resuspended in the same solution supplemented with 4% paraformaldehyde (PFA, Alfa Aesar), and analyzed by a flow cytometer using CellStream (Luminex). Gating of cells was carried out on the basis of forward and side scatter. The fluorescence intensity (FL) of 10000 gated neutrophils was measured for each sample, and phagocytosis was identified when neutrophils expressed fluorescence. The geometric mean of the fluorescence intensity (GMFI) was calculated using CellStream software.

**Cytokine measurement in whole human blood**. Freshly drawn hirudin-treated whole human blood was incubated with WT and ΔCbpD as described in the "Bacterial survival assays in blood", for 1 and 3 h. Plasma samples were immediately collected by centrifugation (10 min, 2000 × g) at 4 °C, stored −70 °C. Cytokine profiling was performed using a Bio-Plex Pro™ Human Cytokine 27-plex Assay (Bio-Rad) according to the manufacturer's instructions.

**Measurement of complement activation product in whole human blood**. Freshly drawn hirudin-treated whole human blood was incubated with WT and ΔCbpD for 30 min under shaking as described in the "Bacterial survival assays in blood". When indicated, $rCbpD_{PA}$ at a final concentration of 20 µg ml$^{-1}$ was added to the samples. After incubation, the samples were immediately treated with 1 µM EDTA and plasma was collected by centrifugation (10 min, 2000 × g) at 4 °C. The concentrations of products resulting from complement activation, C3bc, and TCC, were measured using in-house ELISA assays as described in detail previously[123]. Activation of the final common pathway (reflected in C3bc levels) was quantified using the mAb bH6 (Siemens Healthcare) specific for a neoepitope exposed in C3b, iC3b, and C3c, as previously described. Activation of the terminal complement complex/membrane attack complex (TCC/ MAC) was measured using the mAb aE11 specific for a neoepitope on C9[59]. The level of C5a in plasma was detected using a C5a ELISA kit from Hycult.

**Immunoblot analysis of CbpD**. An overnight culture of bacteria in LB was diluted 1:100 and grown in LB supplemented with carbenicillin (300 µg ml$^{-1}$) and arabinose (6.6 mM) to the mid-exponential growth phase. To assess the release of CbpD, 1 mM PMSF and 1× Complete Mini EDTA-free protease inhibitors were immediately added to the bacterial culture, and the supernatant was collected by centrifugation (4500 × g, 15 min, 4 °C). The culture supernatant was passed through a 0.22 µm filter to remove remained bacterial debris and concentrated 20 times using an Amicon Ultra 10 K centrifugal filter device (Millipore). The presence of CbpD was evaluated by immunoblotting using a CbpD-specific antibody. The affinity-purified CbpD antibody was generated by Davids Biotechnologie via immunizing rabbits with peptides KDGYNPEKPLAWSDLEPA and DAQGRDAQRHSLTLAQGANGA. HRP-conjugated goat anti-rabbit IgG (Invitrogen, Cat no. 65–6120, dilution 1:5000) was used as the secondary antibody. The uncropped blot is provided as Supplementary Fig. 16.

**Cell viability measurement**. THP-1 and HL-60 cells were seeded in 96-well plates (Corning) containing RPMI and IMDM supplemented with FBS (see "Bacterial strains and cell lines"), respectively. The cells were left untreated or treated with $rCbpD_{PA}$ at a final concentration of 0.2, 20, 50, or 100 µg ml$^{-1}$. At various time points, plates were centrifuged, and the cell culture supernatants were collected and validated for viability using a cytotoxicity detection kit (Promega) according to the manufacturer's guideline.

**Competition for surface-expressed receptor binding**. To determine a putative receptor for CbpD in blood, the mix of neutrophils and PBMCs ($5 \times 10^6$ cells ml$^{-1}$) were incubated with 20 µg ml$^{-1}$ $rCbpD_{EC}$ for 15 min in RPMI/HSA on ice. Subsequently, FITC-, PE-, and APC-conjugated mAbs directed against a panel of surface-expressed receptors of leukocytes were added, followed by 30 min incubation on ice. Phycoerythrin (PE)–conjugated monoclonal antibodies (mAbs) directed against CD114 (LMM741,Cat no. 554538, dilution: 1:10), CD87 (VIM5, Cat no, 555768, dilution 1:3), CD54 (HA58, Cat no. 555511, dilution 1:7.5), CD58 (1C3, Cat no. 555921, dilution 1:10), CD35 (E11, Cat no. 559872, dilution 1:6), CD47 (B6H12, Cat no. 556046, dilution 1:5), CD119 (GIR-208, Cat no. 558934, dilution 1:15), CD49b (12F1, Cat no. 555669, dilution 1:15) and CD44 (515, Cat no. 550989, dilution 1:150) were purchased from BD Biosciences. Fluorescein isothiocyanate (FITC)-labeled mAbs directed against CD11a (HI111, Cat no.5553, dilution 1:4), CD15 (MMA, Cat no.332778, dilution 1:150), CD147 (HIM6, Cat no: 555962, dilution: 1:10), CD18 (3G8, Cat no. 347953, dilution 1:15), CD31 (WM54, Cat no. 555445, dilution 1:10), CD46 (E4.3, Cat no. 555949, dilution 1:15), CD9 (M-L13, Cat no. 555371, dilution 1:7.5), and CD66 (B1.1, Cat no: 551479, 1:15) were purchased from BD Biosciences. Allophycocyanin (APC)–labeled monoclonal antibodies against CD13 (WM15, Cat no. 557454, dilution 1:4), CD14 (M5E2, Cat no. 555399, dilution 1:4), CD16 (3G8, Cat no. 557710, dilution 1:300), CD11b (ICRF44, Cat no. 550019, dilution 1:4), CD11c (B-ly6, Cat no. 559877, dilution 1:4), CD29 (MAR4, Cat no. 559883, dilution 1:10), CD55 (IA10, Cat no. 555696, dilution 1:15) and CD45 (HI30, Cat no. 555485, dilution 1:75) were purchased from BD Biosciences. Anti-CD120a-FITCI (16803.161, Cat no. FAB225G, dilution 1:4), CD120b-FITC (22235.311, Cat no. FAB226F, dilution 1:5), CD181-PE (42705.111, Cat no. FAB330P, dilution 1:10), CD182-PE (CXCR2, Cat no. FAB331P, dilution 1:6), and Siglec-9-APC (191240, Cat no. FAB1139A, dilution 1:20) were from R&D Systems. Anti-CD63-PE (CLB-gran/12, Cat no. IM1914U, dilution 1:15), CD43-FITC (6D269, Cat no. sc-70684 FITC, dilution 1:15), BLTR-FITC (202/7B1, Cat no. MCA2108F, dilution 1:7.5) and CD89 (Mip8a, Cat no. MCA1824PE, dilution 1:7.5), CD32-PE (7.3, Cat no. NBP2-47830PE, dilution 1:150), CD282-PE (T2.5, Cat no.12-9024-82, dilution 1:30), were purchased from Beckman Coulter Immunotech, Santa Cruz, Abd-Serotec, and Ebioscience, respectively. CD10-APC (MEM-78, Cat no. CD1005, dilution: 1:10) were obtained

from Invitrogen. CD88-PE (S5/1, Cat no. 344304, dilution 1:50) were purchased from BioLegend. FITC-conjugated formyl-Nle-Leu-Phe-Nle-Tyr-Lys which is abbreviated as FITC-fMLP (Cat no. F1314, dilution 1:1000) were purchased from Invitrogen. After washing, fluorescence was measured by flow cytometry (FACS-Verse, Becton Dickinson). The data associated with PMNs are provided.

**Complement deposition on bacteria**. The complement deposition was assessed as described previously[58], with modifications. Bacteria were grown to mid-log phase ($OD_{600} \sim 0.5$–0.6) in LB, washed in RPMI/HSA to remove secreted proteins, and suspended in RPMI/HSA to an $OD_{660}$ of 1.0. The washed bacteria were then diluted 10 times in RPMI/HSA. Next, 50 µl of bacterial suspension was incubated with 50 µl of diluted NHS, HI-NHS and C5-depleted NHS (ΔC5) (at the final concentration of 0, 10, and 30% (v/v) unless indicated otherwise). The samples (total volume = 100 µl) were incubated at 37 °C under constant shaking for 45 min. RPMI/HSA was used as a serum diluent/buffer or washing solution. To measure C3b, C5b9, and Bb deposition, bacteria were incubated with mouse anti-C3b (previously described in ref. [58], in house Alexa Fluor 488-labeled, 3 µg ml$^{-1}$ or C5b9 (in house Alexa fluor-488-labeled, aE11, 1 µg ml$^{-1}$ (both were kindly provided by Prof. Rooijakkers, University Medical Center Utrecht, the Netherlands) and murine monoclonal anti-human Bb (1 µg ml$^{-1}$, Quidel) in 100 µl RPMI/HSA for 45–60 min at 4 °C. When needed, bacteria were further incubated with Alexa Fluor 488-conjugated goat anti-mouse IgG (1 µg ml$^{-1}$, Invitrogen). Bacteria were washed and analyzed by flow cytometer using the MACSQuant (Miltenyi biotech) or CellStream (Luminex) by analyzing 10000 events per condition. Bacteria were gated based on their forward and side scatter properties and the geometric mean of the bacterial population was determined. The data were analyzed in FlowJo or the CellStream software.

**Complement activity analysis**. To assess the effect of $rCbpD_{PA}$ on the functional capacity of the classical (CP) and alternative (AP) complement pathways, the protein at the final concentration of 0, 10, 20, 50, 100, and 200 µg ml$^{-1}$ was incubated with 90% (v/v) pooled NHS for 5 min at 37 °C. C5-depleted NHS (ΔC5) and the C5 inhibitor OmCI (20 µg ml$^{-1}$) were included as controls. The functionality of the classical (CP) and alternative (AP) complement activation pathways was assessed by the Wielisa®Complement system Screen (SVAR Life Science) using C5b-9 levels as readout[124]. The ELISA and % of activity calculation were performed according to the manufacturer's instructions.

**Immunofluorescence microscopy**. Immunofluorescence microscopy was performed as previously described[125], with modifications. Bacteria were prepared as described above under "Bacterial Survival Assays in Blood" and incubated with 10% (v/v) NHS or HI-NHS for 45 min at 4 °C under shaking. The samples were washed twice with PBS supplemented with 1% BSA (Sigma) (PBS/BSA). Next, the bacteria were fixed by PBS/BSA containing 4% paraformaldehyde (PFA, Alfa Aesar) for 20 min at RT. Bacteria were washed once with PBS/BSA, blocked in PBS/BSA for 1 h at RT, and incubated with mouse anti-C5b-9 (1 µg ml$^{-1}$, aE11, Santa Cruz) followed by incubation with Alexa Fluor 488-conjugated goat anti-mouse IgG (1 µg ml$^{-1}$, Life Technologies) for 45 min. Bacteria were washed once with PBS/BSA and then with Hanks Balanced Salt Solution (HBSS, Gibco). Ultimately, the pellet was resuspended in HBSS supplemented with the lipophilic dye FM5-95 (Invitrogen, 10 µg ml$^{-1}$) to label bacterial membranes. Labeled bacterial cells were mounted onto glass slides (covered with a thin layer of agarose gel) and secured with coverslips. Fluorescence microscopy was performed on a Zeiss AxioObserver equipped with an ORCA-Flash4.0 V2 Digital CMOS camera (Hamamatsu Photonics) and ZEN Blue software. Images were acquired through a ×100 phase-contrast objective and analyzed by ImageJ/Fiji (v 2.1.0/1.53c).

**Membrane integrity analysis**. Outer membrane permeability of WT, ΔCbpD, and the in trans-complemented strain upon exposure to human serum was evaluated, as previously described[45] with modifications. An overnight culture of bacteria was diluted 1:100 and grown in LB to the mid-exponential growth phase. Bacteria were washed with PBS and resuspended to $OD_{600} = 0.4$ in RPMI/HSA. NHS or HI-NHS was added at the final concentration of 10% (v/v) to a subset of the bacterial cultures. The cultures were shaken at 37 °C for 1 h, after which cells were collected by centrifugation (3000 × g, RT, 5 min) and resuspended in 10 mM Tris buffer pH 8.0. Next, N-phenylnaphthylamine (NPN) was added at a final concentration of 40 µM in a final volume of 200 µl per sample, in 96-well flat-bottom plates (Costar). Plates were immediately read in a Synergy H1 Hybrid Reader (BioTek) using excitation and emission wavelengths of 250 nm and 420 nm, respectively.

**Detergent sensitivity assay**. The sensitivity of WT and ΔCbpD to sodium dodecyl sulfate (SDS) was evaluated as previously described with minor modifications[47]. The bacterial culture was exposed to increasing concentrations of SDS (0, 1, 2, and 5%, w/v), incubated for 5 min at 37 °C, and then the $OD_{600}$ was read using a Varioskan™ LUX multimode microplate reader (Thermo Scientific).

**Isolation, purification, and analysis of LPS**. An overnight culture of WT and ΔCbpD was diluted 1:100 and grown in LB medium to the exponential growth

phase (OD$_{600}$: 0.8–0.9). Bacteria were washed with PBS, and the pellet was kept at −80 °C until LPS extraction. LPS was extracted from washed and packed cells by the Westphal and Jann (1965) extraction method[126]. Briefly, samples were suspended in water and an equal volume of pre-heated 90% phenol (Sigma), followed by stirring of the reaction mixture at 65 °C for 40 min[127]. Samples were then immediately cooled down to 10 °C and centrifuged (2968 × $g$, 45 min, 10 °C). The top phenol saturated aqueous layer containing LPS was carefully pipetted out and extensively dialyzed against water to remove phenol. The samples were lyophilized, ultra-centrifuged (120,000 × $g$, 4 h, 10 °C), and the precipitated LPS was subjected to additional Benzonase nuclease (EMD Millipore) and Proteinase K (Sigma) treatments, followed by the second round of ultra-centrifugation at (120,000 × $g$, 4 h, 10 °C). The precipitated LPS was used for monosaccharide and fatty acid composition analysis by GC-MS (Agilent Technologies) as a TMS derivative of methyl-glycoside or alditol acetate derivative. Analysis of keto-deoxy-D-*manno*-8-octanoic acid (Kdo) was done by ultra-performance liquid chromatography with fluorescence detection (RP-UPLC, Acquity, Waters) using a BEH-C18 column (Waters). Briefly, a known amount of LPS was hydrolyzed using 2 M HOAc (80 °C, 3 h), followed by removal of acid, and tagging Kdo with 4, 5 methylenedioxy-1, 2-phenylenediamine dihydrochloride (DMB, Sigma) prior to UPLC analysis. For lipid-A analysis, a known amount of purified LPS was hydrolyzed using 2% HOAc (100 °C, 2 h). Next, the lipid-A was precipitated by centrifugation (5000 × $g$, 5 min), dried by lyophilization, and dissolved in a 3:1 chloroform: methanol (v/v) mixture containing 1 mM ammonium-formate (Sigma). The samples were analyzed using an LTQ-XL Ion-Trap (Thermo Scientific) mass spectrometer in negative mode.

**Transmission electron microscopy (TEM)**. Overnight cultures of WT and ΔCbpD in LB were diluted 1:100 and grown in LB to the mid-exponential growth phase. To visualize the effect of serum on bacterial lysis, the bacteria were washed in PBS and suspended in RPMI-HSA supplemented with 10 % (v/v) NHS or HI-NHS for 1 h. The cell pellets were then washed in 0.1 M sodium cacodylate buffer (Sigma) and fixed briefly in a mix of 2% paraformaldehyde (Sigma) and 2.5% glutaraldehyde (Sigma) in 0.1 M sodium cacodylate. Samples were treated with 1% osmium tetroxide (OsO$_4$; TAAB) before stepwise dehydration in increasing concentrations of ethanol, followed by incubation in acetone. Next, the samples were infiltrated with propylene oxide, embedded using the EMbed 812 resin kit (Electron Microscopy Sciences), and finally polymerized 48 h at 60 °C. Ultrathin sections were collected on copper grids and stained with 2% uranyl acetate and 0.5% lead citrate on an automated contrasting instrument Leica EM AC20 (Leica Microsystems). Finally, the grids were imaged using TEM (Philips CM10 microscope) equipped with a 600-W camera (Gatan) (80–90 kV).

**Murine model of intravenous infection and serum cytokine/chemokine profiles**. An established model of pseudomonal systemic infection was employed to identify the difference in virulence between WT and the isogenic mutant ΔCbpD. Mice were Kept in filter-top cages with access to food pellet and water under controlled ambient temperature (20–22 °C) and relative humidity (30–70%), 12-h light/12-h dark cycle. Eight-week-old CD1 mice (Charles River Laboratories) were infected intravenously with approximately 2 × 10$^7$ CFU of WT or ΔCbpD by tail vein injection ($n = 10$ for mortality estimation, $n = 8$ for pathophysiological analysis). In parallel, an additional set of mice ($n = 4$) were mock-infected with PBS and served as control. Observed mortality was recorded twice per day up to 14 days post infection. For evaluation of bacterial burden, mice were euthanized 4 h post infection, and organs were subsequently homogenized using a Mini BeadBeater (Biospec). Homogenized samples were serially diluted in PBS and plated on Luria–Bertani agar (LA) to enumerate CFU per gram of tissue (kidney, spleen, and liver) or per milliliter of blood. The sera obtained from infected and control mice (mock-infected with 1× PBS) were used for cytokine profiling, using Bio-Plex Pro™ Mouse Cytokine 23-plex Assay (Bio-Rad) according to the manufacturer's instructions.

**Murine model of lung infection**. Overnight culture of WT and ΔCbpD were re-grown in LB to the early-exponential growth phase (OD$_{600}$ = 0.40) at 37 °C. Bacteria were washed with PBS, and the pellet was diluted in PBS to yield a final concentration of 5 × 10$^6$ CFU/30 μL (the inoculation volume). Mice ($n = 10$/group) were anesthetized with 100 mg kg$^{−1}$ ketamine and 10 mg kg$^{−1}$ xylazine. Once sedated, the vocal cords were visualized using an operating otoscope (Welch Allyn) and 30 μL of bacteria was instilled into the trachea during inspiration using a plastic gel loading pipette tip. Mice were placed on a warmed pad for recovery, observed every 12 h after infection, and lethal events were recorded accordingly.

**Histopathology analysis**. The spleen of the infected or uninfected animals ($n = 8$ for WT, $n = 8$ for ΔCbpD, and $n = 4$ for the control) was dissected and fixed by incubation in 10% paraformaldehyde (PFA) for 24 h, after which the samples were placed in 70% ethanol (long-term storage), until the tissues were embedded in paraffin wax. The spleen sections were deparaffinized according to the standard protocols through immersion in xylene and rehydrated in an ethanol/water gradient, followed by washing with deionized (DI) water. The rehydrated sections/slides were stained with hematoxylin and eosin and evaluated microscopically. Images were captured with an Olympus BX43 light microscope mounted with IDS

UI3260CP-C-HQ 2.3 MP camera (IDS Imaging Development Systems GmbH) using the whole slide scanning software, MicroVisioneer (MicroVisioneer Freising).

**Defining the murine splenic proteome in response to PA14 and ΔCbpD infection**. To characterize organ-specific responses to WT and ΔCbpD infection, mice were injected intravenously, and spleens were collected upon infection, as described in "Murine Model of Intravenous Infection". Uninfected mice (injected with 1× PBS) served as a mock control. The organ homogenates were processed as previously described[128] with modifications. The homogenates were lysed in 75 mM NaCl (Sigma), 3% sodium dodecyl sulfate (SDS, Fisher), 1 mM NaF (Sigma), 1 mM beta-glycerophosphate (Sigma), 1 mM sodium orthovanadate (Sigma), 10 mM sodium pyrophosphate (Sigma), 1 mM PMSF, 1× Complete Mini EDTA-free protease inhibitors, and 1× PhosSTOP™ in 50 Tris-HCL (Sigma), pH 8.5. Insoluble debris was then pelleted by centrifugation for 20 min at 4500 × $g$ and supernatants were transferred to new tubes. Proteins were separated by SDS-PAGE with a 10% Mini-PROTEAN gel and stained with Coomassie brilliant blue R250. The gel was cut, after which proteins were reduced, alkylated, and in-gel digested as described previously[118]. The rest of the analysis (trypsinization, LC-MS/MS) was performed as described in the section "Analysis of the bacterial proteome".

Statistical significance (Supplementary Data 7) was determined by $t$ test. The variance was assessed by an F-test to ensure the correct statistical assumptions were used. To account for multiple comparisons, Benjamini–Hochberg correction was applied with a false discovery rate of 5%. Differentially abundant proteins were defined by having $q$ values ≤ 0.05 and fold change ≥ 1.5 (log$_2$ = 0.58) (Supplementary Data 8). Principal component analysis (PCA) was used to reduce the dimensionality of the multivariate data to two principal components so that the data clusters could be visualized graphically. For this, $K$-nearest neighbors (kNN) imputation was implemented to fill in missing values when possible. If values for all replicates of a sample were missing, the values were replaced with the minimum detected value for that replicate. To further investigate the strain and protein grouping, agglomerative hierarchical clustering was performed and visualized as a polar dendrogram or a heatmap with dendrograms on the $x$ and $y$ axes. Enrichment analysis (Supplementary Data 10 and 11) was performed using Metascape (Stable release 3.5, March 1, 2019)[129] with minimum overlap, $P$ value cutoff, and minimum enrichment set at 3, 0.01, and 2, respectively. STRING-db[130] was utilized through the String App for Cytoscape (Version 3.7.1) to visualize connections between significantly changing spleen-specific proteins. Connections were limited to the default confidence of 0.4 with interactions restricted to the query only. MATLAB R2018b was used for all plots and analyses unless stated otherwise.

Proteins that were uniquely and differentially regulated in a WT/ΔCbpD-specific manner (WT: 10; ΔCbpD: 128; Supplementary Data 10) were imported into Ingenuity Pathway Analysis (IPA) for core analysis (Ingenuity Systems). IPA (QIAGEN Inc.) uses upstream regulator analysis (URA), downstream effects analysis (DEA), mechanistic networks (MN), and causal network analysis (CNA) prediction algorithms to assign functional annotations to proteins and perform regulatory network analysis[131,132]. IPA software was used to perform canonical pathway analysis to determine significantly affected pathways and measure their overlap based on common molecules. IPA's interaction network analysis was used to build networks identifying regulatory events connecting molecular signaling signatures to transcriptional effects.

**Ethics declarations**. Mice were utilized to perform pseudomonal systemic and intratracheal infections. These experiments complied with all ethical regulations for animal research and were conducted under the UC San Diego approved IRB protocol S00227M in accordance with the rules and regulations of the Institutional Animal Care and Use Committee. Blood was drawn from several healthy volunteers (male and female) in accordance with ethical principles of the *Helsinki* Declaration and under the protocols that have been approved by UMC Medical Ethics Committee and REK-Norway (2018/1586). All blood donors provided written informed consent.

**Statistical analysis**. All data, except proteomics, computational modeling, and enzymatic activity, were analyzed and plotted using GraphPad Prism 8.0. Data are presented as the means ± standard error of the mean (SEM) unless otherwise indicated in the figure or supplementary data legend. $T$ test, two-way ANOVA, or log-rank test (in vivo survival) in the GraphPad Prism software package (version 8.3) were used to identify statistical significance ($P < 0.05$). The statistical analysis associated with proteomic data or computational modeling is described in detail under relevant sub-sections.

**Reporting summary**. Further information on research design is available in the Nature Research Reporting Summary linked to this article.

## Data availability
The constructs generated in this study are available upon request. The mass spectrometry proteomics data have been deposited to the ProteomeXchange Consortium via the

PRIDE[133] partner repository with dataset identifiers PXD017971, PXD021888, and PDX018769. SAXS data have been deposited in the SASBDB (SASBDB IDs: SASDK42 (20 °C), SASDJQ5 (24 °C), and SASDJR5 (37 °C). The CbpD structure prediction was based on templates provided by publicly available protein structures with PDB identifiers 6IF7, 2WWX, 2I1S, 4TX8, 1ED7, 2D49, 5IN1, and 4XZJ. For analysis of CbpD structural similarity, the following publicly available structures were used: PDB identifiers 2XWX (for comparison with GbpA) and 4A5W (for comparison with the C5bC6-complex). All data required to evaluate the paper's conclusions are present in the paper, the Supplementary Information, and Source Data. Any additional data are available from the corresponding authors upon reasonable request. Source data are provided with this paper.

## Code availability

The GitHub repository has been used to store code associated with CbpD modeling[134] and the pipeline used for proteomic analysis[135].

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

## Acknowledgements

This work was supported by grants from the Research Council of Norway (grants no. 249865 to G.V.K, 272201 to U.K., 240967 to Å.K.R.), the Norwegian University of Life Sciences, and NIGMS (R35 GM119850 to N.E.L.). We are thankful from Prof. Hogan (providing the isogenic mutant), Prof. S.H.M. Rooijakkers (providing some of the reagents associated with complement analysis), and Assoc. Prof. Federica Briani (providing pGM931 plasmid). We acknowledge services provided by the Norwegian Centre for X-ray Diffraction, Scattering and Imaging (RECX), MicrobesNG, Saga Cluster (owned by the Norwegian Metacenter for High-Performance Computing, project NN1003K), Proteomics Core Facility at the Norwegian University of Life Sciences, Protein-Glycan Interaction Resource of the CFG (supporting grant R24-GM098791) and the National Center for Functional Glycomics (NCFG) at Beth Israel Deaconess Medical Center, Harvard Medical School (supporting grant P41-GM103694). We appreciate the help of Dr. Choudhury for LPS-associated analyses at GlycoAnalytics core, UC San Diego. SAXS experiments were performed on beamline BM29 at the European Synchrotron Radiation Facility (ESRF), Grenoble, France. We are grateful to Local Contact Mark Tully at the ESRF for providing assistance in using beamline BM29. We are also grateful for the great assistance of Assoc. Prof. M. Kjos, Dr. M. Junghare, Dr. G. Mathiesen, Dr. Elvis Chikwati, Dr. D.A.C. Heesterbeek, M. Skaugen, M. Ruyken, and E.C. Hasle Kokkim, and Dr. Sergei Grudinin.

## Author contributions

F.A., V.N., and G.V.K. designed the experiments and wrote the paper. F.A. performed the experiments and associated analysis. H.M. and N.L. assisted in data analysis on proteomics. S.U., B.S., and R.Z. assisted in in vivo analysis. S.M. and E.K. assisted in enzymatic characterization. J.A.L. assisted in cytokine profiling and complement analysis. Å.R., H.V.S., and U.K. performed structural analysis and modeling. A.C. and O.G. assisted in protein purification. O.G. assisted in TEM imaging and PTM analysis. M.Ø.A. assisted in proteome-related work. N.V.S. assisted in receptor binding analysis. N.V.S., V.E., U.K., N.L., and T.M. contributed to experimental design and intellectual input. All authors reviewed and approved the paper.

## Competing interests

The authors declare no competing interests.
