## [Peer Review File · Nature Communications]

REVIEWER COMMENTS

Reviewer #1 (Remarks to the Author):

To the authors - This is an interesting and comprehensive study of a chitinase produced by *P. aeruginosa* that interacts with complement. The data are fascinating, convincing and studies appear to be well done. A few questions remain that might add to the interpretation of your results.

1. The authors have described the role of CbpD during blood infection. As pointed out in the introduction, *P. aeruginosa* is a major respiratory and skin pathogen. Is CbpD promoting pulmonary or skin infection or is its activity primarily limited to the bloodstream where complement would be expected to play a more important role?
2. *P. aeruginosa* induces LPS-mediated inflammation to cause disease. LptE exposes LPS in the bacterial surface. LptE is reduced in the cbpD PA14 mutant. Is lack of surface-exposed LPS the reason why there is less inflammation and less recovered CFUs by the mutant?
3. As briefly noted in the discussion, CbpD production is controlled by Crc, which is the catabolite repressor control that controls carbohydrate consumption by *P. aeruginosa*, and is activated by succinate, which in turn. Succinate is a major LPS-controlled immunometabolite secreted by macrophages. Is succinate facilitating CbpD-mediated complement attack resistance in PA14? What in the host induces CbpD? Is it succinate?
4. Page 2, line 111, it makes references to Fig S6D. 6D panel is not shown in Figure S6.

Reviewer #2 (Remarks to the Author):

Reviewer Summary

The manuscript entitled "The lytic polysaccharide monooxygenase CbpD promotes *Pseudomonas aeruginosa* virulence in systemic infection" by Askarian et al. describes the structure and the role of the lytic polysaccharide monooxygenase, CbpD, during *P. aeruginosa* (PA) infection. The authors describe CbpD as a copper-dependent, tri-modular enzyme whose AA10 catalytic module facilitates the degradation of the model substrate, β -chitin and whose function is augmented in the presence of pyocyanin and azurin.

The authors then evaluated the expression of cbpD and the proteome of PA14 and Δ CbpD under different infection-mimicking conditions. The authors found that distinct proteomic profiles exist for PA14 and Δ CbpD under host physiological conditions, with an observed decrease in metabolic activity and increased demand for protein expression.

The authors then explored the effect of CbpD on virulence under physiological conditions. The authors attribute previously observed effects of CbpD on virulence to immune evasion via resistance to complement-mediated lysis. The authors, however, did not discount the possible effects of CbpD on membrane homeostasis. The authors also showed a differential *ex vivo* production of a small subset of cytokines in response to infection with either PA14 or Δ CbpD in whole blood samples.

Finally, the authors examine the requirement of CbpD during systemic infection in mice, finding that mice intravenously injected Δ CbpD survival rate after 48 hours is significantly higher than those infected with PA14 (80% vs 0%). The authors found that under these infection conditions, Δ CbpD is attenuated, provoking reduced neutrophil infiltration into infected organs and a reduction in serum cytokine levels compared to PA14-infected mice. The authors further probed the effect of CbpD through analysis of the proteome during systemic PA infection in mice infected with either PA14 or Δ CbpD. Finally, the authors found that the control, PA14-infected, and Δ CbpD-infected have distinct proteomic signatures, further suggesting the role of CbpD in the pathogenesis of PA.

Reviewer Comments

Overall, the reviewer finds that the research presented in this manuscript is sound, reproducible, and is relevant to the field as the authors are effective in demonstrating the effects and role of CbpD on the pathogenesis of *P. aeruginosa* during systemic infection. However, the reviewer does recommend that the authors revise, clarify, and elaborate on key aspects of their work discussed below prior to publication in Nature Communications.

Major concerns/Questions to Discuss or Address (to be edited/refined by us)

Line 133 page 3, LB media is called bacteriological media, please refer to it as Luria-Bertani media as this is a commonly used name, unless the authors refer to a different LB media.

Line 194 page 4. Did the authors mean a host for heterologous CbpD expression? Can the authors please clarify this wording in the text?

Figure 1

Figure 1, Can the authors please explain why panels E and F are not in the same format? Additionally, the figure legend for 1E says the conditions were in the presence or absence of 100 uM pyocyanin, but the left axis shows a serial dilution of PCN.

The authors propose that azurin may donate copper to CbpD during infection; however for the sake of the readership, this reviewer suggests that the authors elaborate on the relevance of copper during pathogenesis, especially the need for azurin in copper scavenging.

The authors propose that pyocyanin may donate H₂O₂ as a co-substrate but H₂O₂ could also be provided by neutrophil or macrophage respiratory burst. Why isn't this addressed?

Figure 2

In this reviewer's opinion the first panel of Figure 2A is not providing much information (RNA expression of CbpD in LB media), this figure is just presenting expression in different growth phases. Which seems more an experiment to set up conditions than an actual finding.

The authors emphasize that CbpD is secreted and performed proteomic analyses, but when cbpD expression is evaluated (by RNA), the authors do not examine protein-level production of CbpD or if it is actually secreted under "host-mimicking" conditions by PA14. Please clarify this point.

Line 163 has typo - processed should be processes

Can the authors further contextualize and interpret their proteomics results pertaining to CbpD's role in virulence or systemic infection? For instance, why might the loss of CbpD lead to transcriptional and metabolic changes during infection?

Figure 3

Did the rCbpDA that was added exogenously already contain copper? This would speak to the role of azurin (at least) or if it is acquiring copper from the blood.

Can the authors provide a reference that demonstrates complement activation and killing in the whole blood experiments is sufficient time for bacterial killing? For example, why might host cell apoptosis be affected by the loss of CbpD?

Figure 4

Does CbpD have an impact on virulence in canonical respiratory infections (not systemic)? In a normal route of infection does CbpD play a role in protecting against complement? Intranasal infections would also potentially address and elucidate the actual functional role of CbpD, perhaps on degrading mucins like other LPMOs. The immune context of the blood is in stark contrast to the mucosal immune context. This reviewer is curious about the choice of intravenous versus intranasal infection models. Can the authors please elaborate on this?

Is CbpD acting to evade complement in vivo? This question is not addressed.

I'm a bit surprised you don't see TNF α /IL-1/IL-6 in PA14, those are pretty stereotypical sepsis cytokine. Please explain.

Can the authors further contextualize and interpret their host proteomics results pertaining to CbpD's role in virulence or systemic infection?.

Supplemental

Line 111 refers to Fig. S6D which does not exist.

Nomenclature of glycans in supplemental dataset was not clear and mucins were not evaluated (similar to function of GbpA in *V. cholera*)

The Phylogenetic tree in figure S2 has no branch support, please add this information.

Reviewer #3 (Remarks to the Author):

I am largely commenting on the structural part of the Ms. The authors constructed a homology model of CbpD, performed MD simulations and validated/refined the model against the SAXS data.

The quality of the experimental SAXS data collected on a lab source is somewhat limited. The analysis is based on the measurements

of a single concentration only (in a normal practice, a concentration dependence is measured and extrapolation to zero solute content is made and the authors should explain why was this not done).

Overall, though, the structural model looks feasible and corroborates the simulations. What appears missing is the relation of the structural results to the subsequent studies (functional, molecular biological, ex vivo etc).

It would have been useful to explain how do the structural model and its observed flexibility help in the understanding of the functional features of the protein.

According to the publication guidelines of SAXS data

(Trehwella et al Acta Cryst. (2017), D73, 710),

a Table summarising the results should be presented

(see Table 5 in the guidelines).

Further comments:

The references to the programs used for Fig. 1B, C (Raptor-X, DAMMIF, Pepsi-SAXS) must be given in the main text. Similar holds for the deposition codes to SASBDB.

Chi2: give a proper greek letter

The reported value of the maximum size ($D_{max}=121$ Angstrom) is meaningless.

The accuracy of D_{max} is at best 5 Angstrom, for such a long particle rather 10 Angstrom.

Fig S4: range of q in plate A is incorrect.

Supplementary Ref 32 appears incomplete

Revision of “The lytic polysaccharide monoxygenase CbpD promotes *Pseudomonas aeruginosa* virulence in systemic infection” (NCOMMS-20-25008-T)

Throughout this response, comments from the Reviewers are colored red. Our responses are colored dark blue. In the marked version of the revised manuscript, new text has been highlighted in yellow, while text that has not been rewritten but rather moved within the manuscript is highlighted in grey. We also did some general rephrasing and corrected typing errors, which all are colored green in the revised manuscript. We have also changed “PA14 wild-type” to “WT”/ “WT PA” in the whole manuscript and figures for simplification.

Reviewer #1 (Remarks to the Author):

To the authors - This is an interesting and comprehensive study of a chitinase produced by *P. aeruginosa* that interacts with complement. The data are fascinating, convincing and studies appear to be well done. A few questions remain that might add to the interpretation of your results.

Response: We thank the Reviewer for the supportive comments on the quality and depth of our data and its interesting nature. We appreciate the valuable specific feedback as well.

1. The authors have described the role of CbpD during blood infection. As pointed out in the introduction, *P. aeruginosa* is a major respiratory and skin pathogen. Is CbpD promoting pulmonary or skin infection or is its activity primarily limited to the bloodstream where complement would be expected to play a more important role?

Response: As pointed by Reviewers #1 and #2, evaluating the role of CbpD in respiratory infections is highly relevant and interesting. There are indeed some indications of CbpD being important for respiratory infection, e.g. the induction of *cbpD* transcription upon exposure to human respiratory mucus in mucoidal PA (Cattoir *et al.* Mbio 3, doi:ARTN e00410-12) or its high abundance in the secretome of an acute transmissible cystic fibrosis-associated strain (Scott, *et al.* J. Proteome Res. 2013, 12: 5357-5369, doi:10.1021/pr4007365). In the original study we decided to pursue our research in the direction of blood stream infection since:

1. We obtained no binding of rCbpD_{EC} and rCbpD_{PA} in the mammalian glycan array screening, which covers a variety of structures found in mucins (for details, please see our reply to the question raised by Reviewer 2 on “Nomenclature of glycans in supplemental dataset was not clear and mucins were not evaluated (similar to function of GbpA in *V. cholera*)”.
2. Bloodstream infection causing by PA has a high clinical significance/relevance and we obtained a phenotype in a bacteremia model using *ex vivo* and *in vivo* analysis.
3. As stated in the original “DISCUSSION”, data corroborating the importance of LPMOs in a systemic infection model has been published for *Listeria monocytogenes* (the text in the Discussion reads: “Aligned with our results, deletion of *lpmO* (*Imo2467*) in *Listeria monocytogenes* attenuated bacterial loads in the spleen and liver of mice” (Chaudhuri, S. *et al.* Appl. Environ. Microbiol. 2010, 76: 7302-7305, doi:10.1128/Aem.01338-10).

We agreed with the Reviewer’s suggestion to also explore the role of CbpD in respiratory infections. Thus, we evaluated the impact of CbpD virulence in a murine intratracheal challenge model using CD1 mice and screened mortality over a 4-day period. The experiment showed higher mortality in wild-type (WT) PA14 -infected mice (60%) compared to Δ CbpD-infected mice (10%), supporting the role of CbpD during infection respiratory tract as well as bloodstream infection. The new data have been integrated in Fig. 4a (right panel) within the revised manuscript.

The updated text in the “RESULTS” section (lines 294-298) reads:

“Considering the significance of PA in respiratory infections, the effect of CbpD virulence was also studied in a murine intratracheal challenge model. WT PA infection of CD1 mice with 5×10^6 CFU/mouse experienced approximately 60% mortality within 4 days post-infection while the Δ CbpD-infected mice had 10% mortality at the same time point (Fig. 4a, right panel). These data reveal an explicit requirement of CbpD for full PA virulence in these two *in vivo* models.”

The updated “DISCUSSION” (lines 488-493) reads:

“Based on functional studies, e.g., with the use of knock out strains in infection models, the somewhat enigmatic four-domain LPMO called GbpA, is considered a bacterial colonization factor related to mucosal colonization ^{11,13,83}. In light of our current work and considering the higher survival of Δ CbpD-infected mice in an intratracheal infection model in vivo, it could be that GbpA has dual virulence properties and contributes to immune evasion in a different context of infection, e.g., systemic disease, similar to that of CbpD.”

We have also updated “SUPPLEMENTARY MATERIALS AND METHODS”, section “**Murine model of lung infection**” (698-705) and the **Figure 4** (panel a, line 1018-1021) legend accordingly.

2. *P. aeruginosa* induces LPS-mediated inflammation to cause disease. LptE exposes LPS in the bacterial surface. LptE is reduced in the cbpD PA14 mutant. Is lack of surface-exposed LPS the reason why there is less inflammation and less recovered CFUs by the mutant?

Response: The reviewer’s comment is very interesting and relevant as several proteins associated with the homeostasis of the outer membrane or LPS biogenesis and transport in PA were significantly downregulated in Δ CbpD or only detected in the wild-type parent strain regardless of the growth conditions (LB, RPMI, RPMI/NHS). To test the hypothesis put forward by the referee, LPS was extracted from the WT PA and the Δ CbpD strains grown to mid-logarithmic phase in LB medium and further used for compositional analysis and lipid A profiling. All characterized LPS properties were essentially identical when comparing the WT and Δ CbpD strains, except the latter showing a 1.5-fold increase in Kdo (3-deoxy-d-manno-oct-2-ulosonic acid). The inner core of PA LPS contains two Kdo moieties (Reviewed by Pier. Int. J. Med. Microbiol. 2007, 297 (5): 277-295. doi: [10.1016/j.ijmm.2007.03.012](https://doi.org/10.1016/j.ijmm.2007.03.012)) that, together with two L-glycero-D-manno-heptose moieties, connect the Lipid A structure to the outer core structure of the LPS molecule. Thus, it appears as the Δ CbpD strain contains, on average, one additional Kdo moiety per LPS molecule. Comparing the expression levels of proteins involved in Kdo biosynthesis, e.g., KdsB, which generates the activated form of Kdo, Kdo-CMP, and KdsA, the enzyme responsible for attaching Kdo to Lipid A, did not show trends pointing towards an altered Kdo biosynthesis. Thus, the change in LPS inner core glycan structure is not straight forward to explain from the proteomic analysis. All in all, more and deeper research is needed to determine the functional reason for the increase of Kdo in the Δ CbpD LPS. Having a modified LPS inner core structure may influence the ability of the bacterium to resist complement, but fully understanding the consequences of such alterations requires a dedicated separate study. Nevertheless, we made an effort to investigate if the LPS modification changed the integrity of the membrane using a detergent sensitivity analysis where WT and Δ CbpD grown in LB medium and exposed to variable concentrations of detergent were compared with respect to viability. The assay showed comparable resistance of WT and Δ CbpD, indicating that deletion of CbpD (and thus modification of the LPS) did not influence the cell envelope integrity itself in the absence of environmental stress, e.g. lack of host innate immune components. The latter results corroborated our outer membrane integrity assay (Fig. S10h) that was performed in the absence of serum. The new findings and experiments described above have been included in the “Results” and are discussed in the “Discussion” section. New references and a figure (Supplementary Figure 9, line 1243-1260) have also been added in the revised manuscript.

The revised text in the “RESULTS” (line 171-186) section now reads:

“Several proteins associated with outer membrane homeostasis in PA ⁴⁴⁻⁴⁸ were significantly downregulated (**Fig. 2g and Dataset 3, 4 and 6**) upon loss of CpbD or only detected in WT parent strain under all examined conditions. Channel proteins (OprM, OprD, OprQ, OprE, OprB, and OprF), LPS-modifying/assembly proteins (PagL, LptE, LptD), outer membrane protein assembly Bam complex (BamB, BamE, BamD) and VacJ (**Fig. 2g and Dataset 6**) were among those proteins. In addition, several proteins associated with the type II secretion system (SecD, SecY and XcpQ)⁴⁹ were commonly downregulated or not detected upon loss of CpbD (**Dataset 6**). Despite roles of these proteins in structural stability of the PA outer membrane, both strains displayed comparable detergent resistance upon growth in LB (**Supplementary Fig. 9a**). Given the function of LptE, LptD and PagL, in lipopolysaccharide (LPS) biogenesis or cell surface localization, LPS was extracted from mid-log phase WT and the Δ CbpD mutant. Compositional and structural analysis revealed no difference between the lipid-A structures (**Supplementary Fig. 9d**) nor in the O-antigen and outer core monosaccharide composition (**Supplementary Fig. 9c**). However, the inner core monosaccharide composition had a ~50% higher amount of 3-deoxy-d-manno-oct-2-ulosonic acid (Kdo) in the extracted LPS from Δ CbpD

compared to WT (**Supplementary Fig. 9b**), indicating that CbpD has a direct or indirect influence on LPS composition.”

The updated “DISCUSSION” (line 447-457) now reads:

“While the association between CbpD and complement-mediated resistance is evident, we cannot exclude effects of CbpD on outer membrane homeostasis in PA under environmental stress. Several proteins are involved in outer membrane homeostasis and LPS biogenesis/transport⁴⁴⁻⁴⁸ in PA, and our proteomics studies showed that the deletion of *cbpD* led to decreased expression of several of these proteins (**Fig. 2**). Interestingly the LPS inner glycan core composition of Δ CbpD showed a higher abundance of Kdo, possibly resulting from the altered expression of proteins involved in LPS transport and biosynthesis. Such structural changes may contribute to the higher susceptibility of the Δ CbpD outer membrane to antimicrobial host components, e.g. MAC. (as was indeed observed; **Supplementary Fig. 10**), and thus enhance MAC-mediated clearance. In support of this possible role of CbpD, it has been shown that several Gram-negative bacteria can resist MAC-mediated lysis by altering their surface properties (⁷² and references within).”

Detailed methodology sections on “**Detergent sensitivity assay**” and “**Isolation, purification and analysis of LPS**” have been added to the “SUPPLEMENTARY MATERIALS AND METHODS” (line 642-669).

3. As briefly noted in the discussion, CbpD production is controlled by Crc, which is the catabolite repressor control that controls carbohydrate consumption by *P. aeruginosa*, and is activated by succinate, which in turn. Succinate is a major LPS-controlled immunometabolite secreted by macrophages. Is succinate facilitating CbpD-mediated complement attack resistance in PA14? What in the host induces CbpD? Is it succinate?

Response: To explore this interesting point raised by the Reviewer and obtain a better insight into the influence of succinate on CbpD expression, we evaluated the growth rate of WT and the Δ CbpD strain using succinate as a sole carbon source. In addition, we explored whether an increasing concentration of succinate could influence the expression of CbpD when the wild-type bacterium was growing in minimal medium (PA_{M9}) supplemented with glucose as a carbon source. The obtained results demonstrated comparable growth of the WT and Δ CbpD strains using 40 mM succinate as sole carbon source (no growth was observed with a succinate concentration of 5 μ M). In addition, increasing concentrations of succinate did not change expression of CbpD (analyzed by label-free quantitative proteomics). The examined succinate range included physiologically relevant concentrations of this compound in the blood plasma, e.g. 5 μ M (Kushnir *et al.* Clinical Chemistry 2001, 47: 1993-2002. <https://doi.org/10.1093/clinchem/47.11.1993>).

The updated “RESULTS” section (line 216-218) now reads:

“CbpD expression did neither change in the presence of increasing concentrations of succinate (**Supplementary Fig. 8f**), an important LPS-controlled signaling metabolite produced by phagocytes (reviewed in⁵³).”

The “SUPPLEMENTARY MATERIALS AND METHODS”, sections “**Analysis of the bacterial proteome**” (line 479-487) and “**Bacterial growth curves**” (line 378-387), and the Figure Legend for Supplementary **Figs. 8** (panel f, line 1236-1241) and **10** have been updated correspondingly.

4. Page 2, line 111, it makes references to Fig S6D. 6D panel is not shown in Figure S6.

Response: We thank the reviewer for alerting us to the mistake. The text has been updated to Supplementary Fig. 6c.

Reviewer #2 (Remarks to the Author):

Reviewer Summary

The manuscript entitled “The lytic polysaccharide monoxygenase CbpD promotes *Pseudomonas aeruginosa* virulence in systemic infection” by Askarian *et al.* describes the structure and the role of the lytic polysaccharide monoxygenase, CbpD, during *P. aeruginosa* (PA) infection. The authors describe

CbpD as a copper-dependent, tri-modular enzyme whose AA10 catalytic module facilitates the degradation of the model substrate, β -chitin and whose function is augmented in the presence of pyocyanin and azurin.

The authors then evaluated the expression of cbpD and the proteome of PA14 and Δ CbpD under different infection-mimicking conditions. The authors found that distinct proteomic profiles exist for PA14 and Δ CbpD under host physiological conditions, with an observed decrease in metabolic activity and increased demand for protein expression.

The authors then explored the effect of CbpD on virulence under physiological conditions. The authors attribute previously observed effects of CbpD on virulence to immune evasion via resistance to complement-mediated lysis. The authors, however, did not discount the possible effects of CbpD on membrane homeostasis. The authors also showed a differential ex vivo production of a small subset of cytokines in response to infection with either PA14 or Δ CbpD in whole blood samples.

Finally, the authors examine the requirement of CbpD during systemic infection in mice, finding that mice intravenously injected Δ CbpD survival rate after 48 hours is significantly higher than those infected with PA14 (80% vs 0%). The authors found that under these infection conditions, Δ CbpD is attenuated, provoking reduced neutrophil infiltration into infected organs and a reduction in serum cytokine levels compared to PA14-infected mice. The authors further probed the effect of CbpD through analysis of the proteome during systemic PA infection in mice infected with either PA14 or Δ CbpD. Finally, the authors found that the control, PA14-infected, and Δ CbpD-infected have distinct proteomic signatures, further suggesting the role of CbpD in the pathogenesis of PA.

Reviewer Comments

Overall, the reviewer finds that the research presented in this manuscript is sound, reproducible, and is relevant to the field as the authors are effective in demonstrating the effects and role of CbpD on the pathogenesis of *P. aeruginosa* during systemic infection. However, the reviewer does recommend that the authors revise, clarify, and elaborate on key aspects of their work discussed below prior to publication in Nature Communications.

Response: We thank the Reviewer for the supportive comments on the quality and depth of our data and its interesting nature. We appreciate the valuable specific feedback as well.

Major concerns/Questions to Discuss or Address (to be edited/refined by us)

Line 133 page 3, LB media is called bacteriological media, please refer to it as Luria-Bertani media as this is a commonly used name, unless the authors refer to a different LB media.

Response: We agree with the reviewer and to avoid any confusion have updated not only line 133 (line 133 in the original manuscript and 137 in the revised manuscript) but also the rest of the text, e.g. lines 140 and 151, if needed accordingly. All changes are highlighted as yellow within the text.

Line 194 page 4. Did the authors mean a host for heterologous CbpD expression? Can the authors please clarify this wording in the text?

Response: We apologize for not being clear and have clarified it within the revised “RESULTS” section (line 222-225) by replacing “heterologous bacteria” with “other bacterial strains”. The revised text reads:

*“To pursue this point further, we hypothesized that CbpD might also promote blood survival of other bacterial strains that lack LPMOs in their genome. According to the CAZy database⁵⁴, neither Gram-negative *E. coli* nor Gram-positive *Staphylococcus aureus* harbor LPMO-encoding genes in their genomes.”*

Figure 1

Can the authors please explain why panels E and F are not in the same format? Additionally, the figure legend for 1E says the conditions were in the presence or absence of 100 μ M pyocyanin, but the left axis shows a serial dilution of PCN.

Response: These are good points made by the referee. We originally showed the chromatograms associated with Fig. 1e as Supplementary Fig. 7c (left panel), and we had pointed this out in the figure legend. However, we agree that the data representation in Fig. 1e (heatmap format) can be a bit confusing. Thus, we have replaced the panel with the chromatograms representing 2h incubation of rCbpD and ascorbate in the presence of 0,10,100 and 1000 uM pyocyanin (PCN). The chromatograms have been offset on the Y-axis to facilitate visual interpretation. We still keep Fig. S7C in its original form in order to provide information about different timepoints and the presence/absence of different reductants (e.g., ascorbate/NADH) when CbpD is exposed to a serial dilution of PCN. Regarding the reviewer's second concern, we thank the reviewer for alerting us to the mistake in the Figure legend (Fig. 1e), which now has been corrected. It has been updated accordingly in the revised manuscript:

The revised figure legend (**Fig. 1e**), lines 850-855, now reads:

*“e HILIC analysis of reaction products emerging from a reaction of 1 μ M rCbpD_{EC} with 10 mg/ml β -chitin, 250 μ M ascorbate in 20 mM Tris-HCl pH 7.0, and serial dilutions of pyocyanin (PCN) for 2h at 37 °C. The chromatograms have been offset on the Y-axis to enable visual interpretation of their quantitative magnitude. Chromatograms for reactions containing serial dilutions of PCN and with or without various reductants that were sampled at different time points are shown in **Supplementary Fig. 7c.**”*

The authors propose that azurin may donate copper to CbpD during infection; however for the sake of the readership, this reviewer suggests that the authors elaborate on the relevance of copper during pathogenesis, especially the need for azurin in copper scavenging. The authors propose that pyocyanin may donate H₂O₂ as a co-substrate but H₂O₂ could also be provided by neutrophil or macrophage respiratory burst. Why isn't this addressed?

Response: Thank you for these good comments. Discussing the role of copper and the importance of azurin in nutritional immunity at the host-pathogen interface is indeed highly relevant for the manuscript. Additionally, we agree that the possibility of H₂O₂ availability, as co-substrate, due to respiratory burst should not be excluded from the discussion. Thus, we have elaborated these points further in the “Discussion” in the revised manuscript, which also includes addition of new references.

The revised text in the “DISCUSSION” section (lines 458-477) reads:

*“The effect of CbpD on survival in blood ex vivo depends on its catalytic activity, as the catalytically inactive H129A variant could not complement cbpD deficiency (**Fig. 3**). Similar to all other biochemically-characterized LPMOs from pathogenic bacterial species, CbpD degrades chitin by oxidation. Using chitin as a model substrate we noted that CbpD enzymatic performance was affected by other redox-related virulence factors that are secreted by PA, namely pyocyanin, possibly generating the LPMO co-substrate H₂O₂, and azurin, which could supply the LPMO with copper (**Fig. 1**). Considering the significance of copper in host nutritional immunity and bacterial physiology⁷³, pathogens have developed multiple strategies to overcome this nutritional challenge⁷³. Azurin is an important component of a Cu²⁺-scavenging pathway in PA, more specifically a type VI secretion system (T6SS)-mediated metal transport pathway, which is involved in sequestering of Cu²⁺ from the environment⁷⁴. Intriguingly, several existing observations connect CbpD with pyocyanin and azurin. First, secretion of azurin and CbpD are regulated through the same post-transcriptional regulator, called Crc⁷⁵. Second, pyocyanin, azurin, and CbpD are all regulated through quorum sensing⁷⁶. Thus, given the CbpD-compatible concentrations of PCN measured in human blood (up to 130 μ M)⁷⁷, and the presence of ascorbate in the blood⁷⁸, it is conceivable that these secreted compounds act in concert during PA systemic infection. Alternatively, H₂O₂ can also be provided through the oxidative respiratory burst produced by phagocytes during infection, an important mechanism of antimicrobial host defense⁷⁹. Copper could also be acquired from the host by CbpD itself, as shown in a recent study of the fungal pathogen *Cryptococcus neoformans*, which utilizes an LPMO-like protein for copper acquisition during infection¹⁹.”*

Figure 2

In this reviewer's opinion the first panel of Figure 2A is not providing much information (RNA expression of CbpD in LB media), this figure is just presenting expression in different growth phases. Which seems more an experiment to set up conditions than an actual finding.

Response: After re-evaluating this panel, we agree with the reviewer and have removed it from figure 2 (**panel a**). The associated figure legend (**Fig. 2a**, line 862-865) and the “SUPPLEMENTARY MATERIALS AND METHODS”, section “**cbpD expression profile**” (lines 406-432) have been updated accordingly.

The authors emphasize that CbpD is secreted and performed proteomic analyses, but when *cbpD* expression is evaluated (by RNA), the authors do not examine protein-level production of CbpD or if it is actually secreted under “host-mimicking” conditions by PA14. Please clarify this point.

Response: We agree with the reviewer’s comment. In order to amend this issue, we have conducted proteomics experiments to examine the secretion of CbpD by growing WT and Δ CbpD to mid-logarithmic phase (OD 600=0.6) in LB or tissue culture medium (RPMI supplemented with 10% LB to support the bacterial growth, i.e. “host-mimicking” conditions). The resulting data showed secretion of CbpD under both examined conditions. It should be noted that we did not include RPMI/NHS in our experimental setting. We made this decision based on our experience that it is very challenging to profile the expression of a particular bacterial protein in the secretome in the presence of normal human serum as the latter contains an abundance of proteins that can mask the signals of many bacterial proteins upon analysis. The new data has been added to the revised manuscript as Supplementary Fig. S8c. In addition, based on a suggestion provided by Reviewer #1 (please look at response to Reviewer 1, Q3 above), we evaluated the expression of CbpD in response to succinate (the bacterial growth in M9_{PA} [in the absence of succinate] and LB were included as controls). In accordance with our transcriptional data (Fig. 2a), the expression of CbpD in this experiment was higher in the minimal medium, M9_{PA} (in the absence of succinate), compared to LB medium (Supplementary Fig. S8f). Together, all these results verifying translation and secretion of CbpD in our experimental conditions. The “SUPPLEMENTARY MATERIALS AND METHODS” section “**Analysis of the bacterial proteome**” (line: 441) and the “RESULTS” have been updated accordingly and all amendments are highlighted.

The revised text in the RESULTS section (line 147-158) reads:

*“Following verification of *cbpD* transcription, the functional impact of CbpD on PA pathophysiology was scrutinized. This was achieved by comparing the proteomic profiles of the WT parent strain PA and its isogenic mutant PA14 Δ *cbpD* (hereinafter referred to as “ Δ CbpD”) grown to mid-logarithmic phase (OD 600=0.6) in LB medium or in a tissue culture medium (RPMI) that better mimics in vivo conditions, with or without supplemental normal human serum (NHS). Secretion of CbpD in LB and RPMI was verified by quantitative proteomic analysis of the secretome (**Supplementary Fig. 8c**). Our analysis identified 2128 proteins, of which 1202 were shared for both strains under all three conditions (**Fig. 2b; Dataset 2**). Principal component analysis (PCA) (**Fig. 2c**) and hierarchical clustering (**Fig. 2e; and Dataset 2**) showed strong coherence between biological replicates and a different protein expression response of WT compared to Δ CbpD, particularly in the presence of human serum (RPMI/NHS).”*

The figure legend (Supplementary Fig. 8, panel c, line 1227-1230) has also been modified accordingly.

Line 163 has typo - processed should be processes.

Response: This has been corrected.

Can the authors further contextualize and interpret their proteomics results pertaining to CbpD’s role in virulence or systemic infection? For instance, why might the loss of CbpD lead to transcriptional and metabolic changes during infection?

Response: This is a very good observation and we agree that the manuscript would benefit from a deeper analysis and interpretation of the proteomics data. Proteome modulation can be a protective strategy that aids the pathogen to survive within the host’s hostile environment. In our expanded analysis, the modulation strategies that were employed by Δ CbpD compared to WT PA clearly show the struggle of the isogenic mutant in achieving physiological adaptation to the host environment and overcoming the complement-mediated stress due to presence of serum. To improve the discussion of these issues, the “DISCUSSION” and the “RESULTS” sections have been updated with additional text accordingly and new references have been added to the manuscript.

The revised text in the “RESULTS” (line 191-203) now reads:

“Finally, upon exposure to NHS, a wider set of functionally inter-connected regulators (e.g., Vfr, AlgU, MvaT, AlgR, AlgQ, Fur, FliA) and proteins associated with virulence or environmental versatility of PA (**Dataset 3**; marked in bold), were differentially up- or down-regulated in the mutant lacking cbpD (**Fig. 2i and 2j**). Several super-regulators of quorum sensing (QS) in PA (AlgR, DksA, MvaT, and Vfr)⁵⁰, or QS by-products such as phenazine synthesis (PhzM, PhzB1, and PhnB), T3SS regulator (Vfr), or virulence-associated transcriptional factors (AlgU, AlgR)⁵¹ were upregulated (**Fig. 2j and 2l**) in Δ CbpD compared to WT in the presence of NHS. In addition, several proteins involved in biosynthesis of antibiotics (BkdB, AcsA, Zwf, PurH, TpiA, GcvH, GlyA) or flagellar-associated proteins (FliD, FliC, FliA) were among the differentially (up/down) regulated proteins in Δ CbpD vs. WT (**Fig. 2i and 2j**). Proteome modulation could be a protective strategy to aid the pathogen survival within a hostile host environment. The different modulation strategies employed by WT and Δ CbpD PA strain suggest that CbpD action directly or indirectly affects several cellular processes associated with metabolism and pathogenicity upon exposure to host environmental conditions.”

The revised text in the “DISCUSSION” (line 394-406) now reads:

“The major impact of CbpD was reflected in our proteome studies that showed significant effects caused by cbpD deletion on both the bacterial and the host proteome (**Figs. 2 & 4**). The specific changes in the Δ CbpD proteome in response to a low concentration of NHS (which contains complement components), included a down-regulation of several metabolic processes and differential regulation of virulence factors, including an up-regulation in the expression of several key regulators or transcriptional factors (e.g., AlgR, DksA, MvaT, AlgU, and Vfr). These changes reflect the struggle of the isogenic mutant to achieve physiological adaptation to the host environment and to overcome the complement-mediated stress (**Fig. 2**). Alteration in carbon metabolism and virulence has been suggested as an adaptive mechanism for PA to promote viability in human blood⁶⁰. Despite the presence of strong crosstalk among virulence-associated regulators in PA⁵¹, the proteome modulation strategy employed by the mutant was not sufficient to produce a functional impact in Δ CbpD defense during sepsis. This underscores the importance of CbpD in PA pathogenesis and adaptability during infection ranging from acute bacteremia to respiratory tract infection (**Fig. 3 & 4**).”

Figure 3

Did the rCbpDA that was added exogenously already contain copper? This would speak to the role of azurin (at least) or if it is acquiring copper from the blood.

Response: This is a good question with a simple answer, but a complicated explanation. First, the answer to the question: Exogenous copper was not added to rCbpD_{PA} before being used in the *ex vivo* experiments (whole human blood analysis). There are three reasons for this:

1. When analyzed for activity towards the model substrate, rCbpD_{PA} (used as is) showed LPMO activity similar to that observed for other chitin-active LPMOs and copper saturated CbpD (see figure below; all *in vitro* activity assays shown in the original manuscript utilized copper-saturated CbpD). This verifies that copper was present in the active site of at least a portion of the CbpD molecules present in the reaction. The result showing inactivity of apo-CbpD (e.g., Fig. S7c) verifies that there was no putative source of copper present in the reaction mixture (e.g., the model substrate) that could provide copper to CbpD. This provides an additional indication that the enzyme, “as is”, contained copper.
2. The copper saturation procedure can leave traces of copper in the enzyme solution, which may give unwanted effects in the *ex vivo* assay.
3. We observed that additional copper saturation of CbpD resulted in protein stability issues upon long term storage (that we do not fully understand). This unwanted effect could have influenced the *ex vivo* assays.

Unfortunately, we do not know the ratio of apo- vs. holo-CbpD, which could have partly addressed the referee question. However, we are working on methods to determine this and on other copper-related aspects of LPMOs, in ongoing studies. To avoid any confusion, we have described this more specifically in the experimental procedure (SUPPLEMENTARY MATERIALS AND METHODS, section “**CbpD activity assay**”, line 349-340):

“For *ex vivo* analysis, rCbpD_{PA} was used “as is” without copper saturation.”

Product formation by 1 μM of rCbpD_{PA} (used as is) after a 30-, 60-, 180-, 240- and 360-minutes reaction at 37 °C with 10 $\text{mg}\cdot\text{ml}^{-1}$ β -chitin in 20 mM Tris-HCL pH 7.0, with and without 1 mM ascorbate as reducing agent, analyzed by HILIC. The degrees of polymerization (DP) of oxidized chitooligosaccharide aldonic acids in a standard sample are indicated. Control reactions without ascorbate did not show product formation.

Can the authors provide a reference that demonstrates complement activation and killing in the whole blood experiments is sufficient time for bacterial killing? For example, why might host cell apoptosis be affected by the loss of CbpD?

Response: We thank the referee for taking up an important point. Complement can indeed kill bacteria within a relatively short time frame. Upon activation of the complement cascade, the immune system triggers a diverse range of actions for the rapid elimination of pathogens, which ranges in reaction time from minutes to hours. The activation of the complement system and the subsequent engagement of MAC in lysis (killing) of Gram-negative bacteria is one of the immune system’s early responses and can be observed within minutes of activation, a trait that has been acknowledged/discussed in several studies (e.g., Reviewed by Heesterbeek *et al.* J Innate Immun 2018; 10:455–464). The rapid activation of the complement system (within 30 min) upon infecting human hirudin-anticoagulated blood with WT and ΔCbpD *ex vivo* was also observed in our study (Fig. 3f). Although the bacterial strain, amount of inoculum, percentage of blood, and the choice of anticoagulant can influence the results of whole blood infection assay (an *ex vivo* model for bacteremia), hirudin-anticoagulated blood has been shown to efficiently clear several Gram-negative strains even after only 1h incubation in several studies (e.g. Mollnes *et al.*, Blood 2002; 100(5):1869-77; van der Maten *et al.*, Sci Rep. 2017; 7:42137; Strobel & Johswich, Sci Rep. 2018; 8: 10225). In accordance with these studies, the overall efficiency of this clearance is also obvious in our *ex vivo* blood assay system (e.g., Fig. 3a.). References to these studies have been included in the revised manuscript (see changes indicated below).

When it comes to the second question, we are not 100% sure if we fully understand how it relates to the first question. However, we interpret it as relating other mechanisms of bacterial clearance/resistance by the host to our results, especially for the ΔCbpD strain that shows activation of additional host immune pathways aiding clearance of pathogens. To give a good answer to the question, it is important to acknowledge that bacteria have evolved several mechanisms for complement resistance (Reviewed by Vogel and Frosch, Mol Microbiol 1999; 32: 1133.; Doorduyn *et al.*, Bio Essays 2019; 41: 1900074). However, while the complement system is an efficient arm of the innate immune system in clearance of Gram-negative pathogens during sepsis, it is not sufficient on its own. To cope with sepsis, the systemic activation of the innate immune system triggers not only the complement but also several other interconnected immune pathways for a successful host response. This was clearly reflected in the observed splenic proteome response, in which we observed enrichment of several distinct pathways, including apoptosis in ΔCbpD infected mice. Employment of apoptosis by the host to combat infection is well acknowledged (e.g., reviewed by Jorgensen *et al.* Nat. Rev. Immunol. 2017, 17: 151-164, doi:10.1038/nri.2016.147). Based on the two questions raised by the reviewer, we have re-written the relevant part of the discussion. In retrospect, we also realize that the original manuscript had too much emphasis put on the enrichment of apoptosis as the dominating host response in the ΔCbpD -infected mice compared to mock-infected mice. In the revised manuscript, we describe a deeper analysis of the host proteomic data and provide a more balanced interpretation of the splenic proteome analysis.

The text in the revised “DISCUSSION” has been thoroughly revised (line 407-440) and reads:

“To cope with sepsis, the systemic activation of the innate immune system triggers a variety of interconnected signaling pathways for a successful host response. Although WT- and ΔCbpD -infected mice showed similar trends in the enrichment of several pro-inflammatory pathways in their shared splenic

proteomes (e.g., Toll-like, Rig-1 like, IFN β), comparison of the unique proteomes revealed that Δ CbpD and WT elicited different host immune responses in the infected mice. The differences in the splenic unique proteomes were reflected by the Δ CbpD-infected mice showing modulation of a larger number of proteins and enrichment of distinct cellular processes or pathways such as apoptosis, cell cycle, and ERK 1/2 pathways (**Fig. 4**). Indeed, the role of apoptosis, one of the top enriched pathways, as a means to aid the host during infection is well established (reviewed in ⁶¹) and is suggested to operate by removing the intracellular niches for selected pathogens ⁶¹. Moreover, the cellular debris resulting from apoptosis can trigger activation of innate immune responses (reviewed in ⁶¹) such as the complement system, which subsequently enhances clearance of the infection (reviewed in ⁶²). The different host immune response elicited by the Δ CbpD strain compared to the WT may reflect the hypovirulent nature of the Δ CbpD strain and underpins the contribution of CbpD significantly to the extent and consequence of PA-associated infection. From the perspective of understanding PA systemic infections in general, one important host marker unique to the WT-infected mice was the complement negative regulator factor H (**Dataset 11**). This observation indicates that the enrichment of cellular processes such as “negative regulation of immune responses” in the WT-infected mice may be partly responsible for the development of intense bacteremia observed for this strain. Importantly, increased expression of factor H during bacterial infection is associated with prolonged hospitalization of the patients ⁶³.

Although PA strains show different levels of resistance to complement-mediated lysis ⁶⁴, the importance of the complement system, particularly the membrane attack complex (MAC; C5b-C9), in eradication of PA is well documented ^{65,66}. In addition, complement-deficient mice show an exacerbated inflammatory response ⁵⁶ and high susceptibility to PA infection ^{56,57}. The role of MAC in killing/lysis of Gram-negative bacteria has been acknowledged/discussed in several ex vivo or in vivo studies, which have shown that this lytic macromolecular complex can clear the majority of the bacteria within minutes to hours, depending on the experimental setting and the bacterial strain (e.g., ⁶⁷⁻⁶⁹ and reviewed in ^{55,70}). Recently, it has been shown that assembly of C5 convertase on the bacterial surface ⁵⁸ and direct attachment of C5b-7 ⁷¹, a MAC precursor, to the bacterial outer membrane, is crucial for MAC-mediated bacterial lysis. While rCbpD did not entirely block complement activation, our data show that CbpD decreases C5a generation, assembly of the C5 convertase, and deposition of C9 and C5b-9 (MAC), which could explain the observed protective effect of the LPMO against complement-mediated killing in blood.”

Figure 4

Does CbpD have an impact on virulence in canonical respiratory infections (not systemic)? In a normal route of infection does CbpD play a role in protecting against complement? Intranasal infections would also potentially address and elucidate the actual functional role of CbpD, perhaps on degrading mucins like other LPMOs. The immune context of the blood is in stark contrast to the mucosal immune context. This reviewer is curious about the choice of intravenous versus intranasal infection models. Can the authors please elaborate on this?

Response: As pointed by Reviewers #1 and #2, evaluating the role of CbpD in respiratory infections is highly relevant and interesting. There are indeed some indications of CbpD being important for respiratory infection, e.g. the induction of *cbpD* transcription upon exposure to human respiratory mucus in mucoidal PA (Cattoir *et al.* Mbio 3, doi:ARTN e00410-12) or its high abundance in the secretome of an acute transmissible cystic fibrosis-associated strain (Scott, *et al.* J. Proteome Res. 2013, 12: 5357-5369, doi:10.1021/pr4007365). In the original study we decided to pursue our research in the direction of blood stream infection since:

1. We obtained no binding of rCbpD_{EC} and rCbpD_{PA} in the mammalian glycan array screening, which covers a variety of structures found in mucins (for details, please see our reply to the question raised by Reviewer 2 on “Nomenclature of glycans in supplemental dataset was not clear and mucins were not evaluated (similar to function of GbpA in *V. cholera*)”.
2. Bloodstream infection causing by PA has a high clinical significance/relevance and we obtained a phenotype in a bacteremia model using ex vivo and in vivo analysis.
3. As stated in the original “DISCUSSION”, data corroborating the importance of LPMOs in a systemic infection model has been published for *Listeria monocytogenes* (the text in the Discussion reads: “Aligned with our results, deletion of *lpmO* (*lmo2467*) in *Listeria monocytogenes* attenuated bacterial loads in the spleen and liver of mice” (Chaudhuri, S. *et al.* Appl. Environ. Microbiol. 2010, 76: 7302-7305, doi:10.1128/Aem.01338-10).

We agreed with the Reviewer's suggestion to also explore the role of CbpD in respiratory infections. Thus, we evaluated the impact of CbpD virulence in a murine intratracheal challenge model using CD1 mice and screened mortality over a 4-day period. The experiment showed higher mortality in wild-type (WT) PA14-infected mice (60%) compared to Δ CbpD-infected mice (10%), supporting the role of CbpD during infection respiratory tract as well as bloodstream infection. The new data have been integrated in Fig. 4a (right panel) within the revised manuscript.

The updated text in the "RESULTS" section (lines 294-298) reads:

"Considering the significance of PA in respiratory infections, the effect of CbpD virulence was also studied in a murine intratracheal challenge model. WT PA infection of CD1 mice with 5×10^6 CFU/mouse experienced approximately 60% mortality within 4 days post-infection while the Δ CbpD-infected mice had 10% mortality at the same time point (Fig. 4a, right panel). These data reveal an explicit requirement of CbpD for full PA virulence in these two in vivo models."

The updated "DISCUSSION" (lines 488-493) reads:

"Based on functional studies, e.g., with the use of knock out strains in infection models, the somewhat enigmatic four-domain LPMO called GbpA, is considered a bacterial colonization factor related to mucosal colonization^{11,13,83}. In light of our current work and considering the higher survival of Δ CbpD-infected mice in an intratracheal infection model in vivo, it could be that GbpA has dual virulence properties and contributes to immune evasion in a different context of infection, e.g., systemic disease, similar to that of CbpD."

We have also updated "SUPPLEMENTARY MATERIALS AND METHODS", section "**Murine model of lung infection**" (698-705) and the **Figure 4** (panel a, line 1018-1021) legend accordingly.

Is CbpD acting to evade complement *in vivo*? This question is not addressed.

Response: We agree with the Reviewer that ultimately future studies to expand on the role of CbpD in evasion of the complement during systemic infection are of interest, but we were faced with some experimental limitations, and feel this is beyond the scope of the current (already expansive) manuscript. The volume of serum/plasma obtained from the individual mice is only 0.1 to 0.15 ml, and thus we decided to only harvest serum at the end of the challenge for cytokine analysis. Although we do not have any direct evidence on whether CbpD contributes to evading of the complement system *in vivo*, we were inspired by this question to look further into the splenic proteome data, specifically searching for proteins associated with the complement cascade. Our data revealed the upregulation of complement factor C4 molecule (crucial in forming C3 and C5 convertase in classical and lectin pathways) and complement regulator factor H in Δ CbpD and WT infected mice compared to control, respectively. Factor H is a crucial negative complement regulator that aids in preventing complement-mediated detrimental damage on host cells while preserving the protective role against the pathogen. The upregulation of factor H in the splenic proteome of WT-infected mice may indicate the severity of infection. Importantly, increased expression of factor H during bacterial infection was also found to be associated with prolonged hospitalization (Willems et al., EBioMedicine 2019, 45: 303-313). We thus connect these findings with increased bacterial burden and mortality in WT-infected compared to Δ CbpD infected mice. To accommodate this new insight into the revised manuscript, the "RESULTS" and "DISCUSSION" sections have been updated accordingly and new references have been added within the text.

The updated "RESULTS" in the revised manuscript (line 374-380) reads:

"Ultimately, analysis of the proteins involved in the complement cascade in the splenic proteome revealed identification of several proteins associated with this system (e.g., Cr2, Cfh, C4b (Slp), Cfb, C3, Vtn, and C1qbp) (Dataset 7). However, only the C4b molecule (representative of complement factor C4) (Dataset 10) and complement factor H (Cfh) (Dataset 11) were identified among the upregulated proteins unique to the Δ CbpD and WT-infected mice, respectively. Vitronectin (Vtn), which is involved in the regulation of the terminal complement cascade, was upregulated (Dataset 11) in the shared proteome of infected vs. mock-infected mice."

The updated "DISCUSSION" in the revised manuscript (line 421-427) reads:

“From the perspective of understanding PA systemic infections in general, one important host marker unique to the WT-infected mice was the complement negative regulator factor H (**Dataset 11**). This observation indicates that the enrichment of cellular processes such as “negative regulation of immune responses” in the WT-infected mice may be partly responsible for the development of intense bacteremia observed for this strain. Importantly, increased expression of factor H during bacterial infection is associated with prolonged hospitalization of the patients⁶³.”

I'm a bit surprised you don't see TNF α /IL-1/IL-6 in PA14, those are pretty stereotypical sepsis cytokine. Please explain.

Response: We do indeed observe increased expression of several stereotypical sepsis cytokines in infected compared to mock-infected mice, but the information was unfortunately hidden in the graphical representation of the data. The reason was we used a standard heatmap for visualization of all profiled cytokines, in which the highly expressed cytokines (e.g., G-CSF) dominated (since they are up to 1000-fold higher in numbers than e.g., IL-6). As the scale is linear, the data with low values will be visualized as comparable/similar level in both infected and control mice, even though the difference may be more than 10-fold. In addition, our focus was/is on the cytokines that showed a statistically significant difference when comparing WT and Δ CbpD infected mice, which was reflected in the version of the heatmap presented in the original manuscript. To improve the graphical representation of the data, we have replaced the original heatmap with a categorical heatmap and provided the geometric means of the cytokine values (to allow direct comparison of the numbers) within each cell (**Fig. 4c** in the revised manuscript). The improved heatmap gives the possibility of better visualization of the cytokine changes. In addition, we generated new graphs, where individual values are plotted (**Supplementary Fig. 12a**). Finally, for consistency, we have also updated the heatmap associated with the cytokine profile in whole human blood as a categorical heatmap (**Fig. 3g** and associated figure legend: line 948-953) and plotted the individual values (**Supplementary Figures 11j-k** and associated figure legend: line 1328-1330) to support the heatmap accordingly.

The revised figure legend (**Fig. 4c**, line 1025-1030) reads:

“**c** The categorical Heatmap shows the concentration of cytokines, chemokines or growth factors in the serum of CD-1 mice 4h post infection (as described in b). The data are depicted as the geometric mean of the cytokine values in each cell, representing one experiment performed with 8 mice/group and analyzed by two-way ANOVA. Mock-infected mice (n=3) were included as control. The significant difference between WT and Δ CbpD is indicated by asterisks (*). Individual values are plotted in Supplementary Fig. 12a.” Additional Information associated with the statistical analysis in figure 4 is provided at the end of the legend as follows: “When applicable, the significance is indicated by asterisks (*): *p \leq 0.05; **p \leq 0.01; ***p \leq 0.001; ****p \leq 0.0001.”

The updated supplementary Fig. 12 (line 1360-1367) reads:

“**Supplementary Fig. 12. In vivo analysis of CbpD virulence. a** Concentration of cytokines, chemokines or growth factors in the serum of CD-1 mice 4h post infection with WT and Δ CbpD. The data are plotted as the mean \pm SEM, representing one experiment performed with 8 mice/group (n=8) and analyzed by two-way ANOVA. Mock-infected mice (n=3) were included as control. The significant difference between WT and Δ CbpD is indicated by asterisks (*): *p \leq 0.05; **p \leq 0.01; ***p \leq 0.001; ****p \leq 0.0001.”

Can the authors further contextualize and interpret their host proteomics results pertaining to CbpD's role in virulence or systemic infection?

Response: We agree that a deeper analysis and interpretation of the host proteomics data would benefit the understanding of the role of CbpD in virulence. To improve the discussion, we have included substantially more analytics of the spleen proteome data, which will help to better describe the difference between infection by the WT compared to the Δ CbpD strain, since this can give us a better starting point for contextualizing and interpreting the role of CbpD in infection. Based on the new analysis, we have added some new text in the “DISCUSSION” section, but have tried to limit ourselves, due to the unavoidable speculative nature of such discussion (there are many variables that may influence the progression of an infection). To improve the analysis of how CbpD (or the lack of CbpD)

contributes to PA pathogenesis and influences the host response and cellular processes, we generated new datasets that were subjected to analysis. These datasets contain significantly regulated splenic proteins unique to the WT- or Δ CbpD-infected mice (newly generated Dataset 8 and 10) and proteins that we classify as being uniquely regulated on the basis of being either detected in Δ CbpD- or WT, but not in the mock-infected control (i.e. up-regulated) or not detected in Δ CbpD- or WT, but detected in the mock-infected control (i.e. down-regulated) (newly generated Datasets 9 and 11, respectively). Additionally, we have employed ingenuity pathway analysis (IPA) software for systemic analysis of unique proteomes in a more specific and relevant biological context (Dataset 10). These new analyses are included in the revised manuscript (see below for textual changes and additions), and several figures (**Fig. 4**, **Supplementary Fig. 13** and **Supplementary Fig. 14**) have been updated/newly generated accordingly. The additional analyses show that distinct cellular processes are elicited in the spleens of Δ CbpD-infected vs mock-infected mice compared to WT- infected vs mock-infected mice, most likely being a consequence of the change in PA pathogenicity during systemic infection *in vivo* (the Δ CbpD strain having become hypovirulent), which thus triggers a different immune response. These analyses expand our understanding of the host proteomics data, and all new findings align well with what was already described in the original manuscript. We have expanded relevant parts of the “RESULTS” and “DISCUSSION” section in accordance with the above, including addition of new references.

The extensively revised text in the “RESULTS” section (line 313-382) now reads:

“The volcano plots of splenic proteomes showed 42 significantly regulated proteins in WT infected vs. mock-infected mice (28 up- and 14 down-regulated), whereas 160 differentially regulated proteins were detected in Δ CbpD-infected vs. mock-infected mice (77 up- and 83 down-regulated; $FC \geq 1.5$ and $q \leq 0.05$; **Fig. 4g** and **Datasets 8 and 10**). Infected spleens shared 200 regulated proteins that had comparable relative expression irrespective of the infecting strain (**Datasets 8-11**). The shared proteome is consisted of 32 significantly regulated proteins ($FC \geq 1.5$ and $q \leq 0.05$) (**Fig. 4h**, **Datasets 8 and 10**) and 168 proteins that were only detected in the infected compared to control mice (No FC value, **Datasets 9 and 11**). Functional analysis of this shared proteome (**Dataset 10 and 11**) revealed high enrichment of several immune responses, including cellular response to IFN β , RIG-I-like receptor signaling pathway and Toll-like receptor signaling as a general response to PA infection (**Supplementary Fig. 13a**). Evaluation of the unique proteome signature associated with the deletion of CbpD revealed that 10 and 128 proteins were regulated (**Datasets 8 and 10**) in response to WT PA or Δ CbpD infection, respectively ($FC \geq 1.5$ and $q \leq 0.05$). In addition, 44 and 65 proteins were only detected in the WT and Δ CbpD-infected compared to control mice, respectively (**Datasets 9 and 11**). The Δ CbpD associated unique proteome had a scattered expression with relative changes from +3.2 (Serpina3g) to -9.7 (Amy2) compared to the control mice (**Datasets 8 and 10**). String analysis revealed these proteins were highly interconnected (**Supplementary Fig. 13b**).

Functional exploration of the distinct markers (193 proteins, **Datasets 10 and 11**) unique to the Δ CbpD-infected spleens showed enrichment of several cellular responses including apoptosis induced DNA fragmentation and autoimmune thyroid disease by the up-regulated proteins, and synthesis of leukotrienes (LT) and eoxins (EX) and engulfment of apoptotic cells by the down-regulated proteins (**Fig. 4i**). Inflammatory responses are well-known to alter metabolism in host cells, which was corroborated by the observed enrichment of pathways associated with several terms coupled to metabolic process such as “Glycine, serine and threonine metabolism”, “Branched-chain amino acid catabolism” and “alpha-amino acid biosynthetic processes” in the down-regulated proteins (**Fig. 4i**). Some of the proteins associated with the enriched cellular processes (e.g., isoforms of 14-3-3 proteins: Ywhae, Ywhag, Ywhaz) were localized in the central part of the STRING network analysis or clustered together (e.g. Hist1h1a, Hist1h1e, Hmgb2, Hmgb2) (**Supplementary Fig. 13b**). To further explore the interactions among regulated proteins (128) unique to the Δ CbpD-infected mice (**Dataset 10**), an ingenuity pathway analysis (IPA)-based protein network was performed, and nine different disease-based and molecular networks were algorithmically generated (**Supplementary Fig. 13c and 14**). The “Inflammatory response, lipid metabolism, small molecule biochemistry” was ranked (score 42) as the top enriched function- and disease-based protein network (**Supplementary Fig. 13c**). Several of the proteins associated with this network showed high connectivity to the extracellular signal-regulated protein kinases 1 and 2 (ERK 1/2) hub (**Supplementary Fig. 14, network 1**), which was also among the enriched pathways associated with upregulated proteins in Δ CbpD-infected mice (**Supplementary Fig. 4i**). On the single protein level, some of the upregulated proteins of this network are involved in immune responses e.g., C4b (representative of complement factor C4), Serpina3g and H2-Ab1 (H-2 class II histocompatibility antigen). The other enriched networks with a score over 20 were “Cancer,

Cell-To-Cell Signaling and Interaction, Cellular Movement”, “Cancer, Cellular Assembly and Organization, Neurological Disease”, “Cancer, Endocrine System Disorders, Protein Trafficking” and “Cancer, Molecular Transport, Organismal Functions” (**Supplementary Fig. 13c and 14**). Canonical pathway analysis of the unique proteome of the Δ CbpD-infected mice revealed significant enrichment of 15 different canonical pathways (**Supplementary Fig. 13d**), in which cell cycle: G2/M DNA damage checkpoint regulation (e.g., Top2a, Ywhae, Ywhag, Ywhaz), valine degradation I (e.g., Aldh111, Bcat2, Dbt), and oxidative phosphorylation (e.g., Atp5e, Cox72a, Ndufa11, Ndufa9, Ndufb10) were among the top three (**Supplementary Fig. 13d and Fig. 4i**). Several of these pathways were highly interconnected and formed two main canonical pathway webs that were associated with either metabolic processes or host immune responses (**Fig. 4i**).

Functional analysis of the regulated or detected markers (in total 54 proteins) (**Datasets 10 and 11**) unique to the WT-infected spleens showed that these proteins were associated with negative regulation of immune responses (e.g., Arg1, Arg2, Ppp3cb, Clec2d, Cfh), negative regulation of cell proliferation (e.g., Cnn1, Hdac2, Pdcd4, Ddah1, Cers2), and several catabolic processes (**Fig. 4i**). IPA analysis of the regulated proteins unique to the WT-infected mice (**Dataset 10**) revealed that 90% of the proteins were categorized under a network associated with “energy production, lipid metabolism, small molecule biochemistry” (score 27) (**Supplementary Fig. 13e**). Several of these proteins (Cstb, Slc25a3, Clec2d, Pdcd4, Ppp3cb) were directly or indirectly associated with the Myc hub, a master transcription factor of multiple proliferative genes (**Supplementary Fig. 13e**), which aligned with our enrichment analysis (**Fig. 4i**). Canonical pathway analysis of the unique proteome (**Dataset 10**) showed necroptosis (e.g., Ppp3cb, SLC25A3) and myo-inositol biosynthesis (e.g., Impa1) as the profoundly enriched pathways in WT infected mice (**Fig. 4i**).

Ultimately, analysis of the proteins involved in the complement cascade in the splenic proteome revealed identification of several proteins associated with this system (e.g., Cr2, Cfh, C4b (Slp), Cfb, C3, Vtn, and C1qbp) (**Dataset 7**). However, only the C4b molecule (representative of complement factor C4) (**Dataset 10**) and complement factor H (Cfh) (**Dataset 11**) were identified among the upregulated proteins unique to the Δ CbpD and WT-infected mice, respectively. Vitronectin (Vtn), which is involved in the regulation of the terminal complement cascade, was upregulated (**Dataset 11**) in the shared proteome of infected vs. mock-infected mice.

Collectively, these data suggest that CbpD contributes to PA virulence in vivo and its loss leads to changes in the host proteome during systemic infection in vivo.”

The extensively revised text in the “DISCUSSION” (line 394-427) now reads:

“The major impact of CbpD was reflected in our proteome studies that showed significant effects caused by cbpD deletion on both the bacterial and the host proteome (**Figs. 2 & 4**). The specific changes in the Δ CbpD proteome in response to a low concentration of NHS (which contains complement components), included a down-regulation of several metabolic processes and differential regulation of virulence factors, including an up-regulation in the expression of several key regulators or transcriptional factors (e.g., AlgR, DksA, MvaT, AlgU, and Vfr). These changes reflect the struggle of the isogenic mutant to achieve physiological adaptation to the host environment and to overcome the complement-mediated stress (**Fig. 2**). Alteration in carbon metabolism and virulence has been suggested as an adaptive mechanism for PA to promote viability in human blood⁶⁰. Despite the presence of strong crosstalk among virulence-associated regulators in PA⁵¹, the proteome modulation strategy employed by the mutant was not sufficient to produce a functional impact in Δ CbpD defense during sepsis. This underscores the importance of CbpD in PA pathogenesis and adaptability during infection ranging from acute bacteremia to respiratory tract infection (**Fig. 3 & 4**).

To cope with sepsis, the systemic activation of the innate immune system triggers a variety of interconnected signaling pathways for a successful host response. Although WT- and Δ CbpD-infected mice showed similar trends in the enrichment of several pro-inflammatory pathways in their shared splenic proteomes (e.g., Toll-like, Rig-1 like, IFN β), comparison of the unique proteomes revealed that Δ CbpD and WT elicited different host immune responses in the infected mice. The differences in the splenic unique proteomes were reflected by the Δ CbpD-infected mice showing modulation of a larger number of proteins and enrichment of distinct cellular processes or pathways such as apoptosis, cell cycle, and ERK 1/2 pathways (**Fig. 4**). Indeed, the role of apoptosis, one of the top enriched pathways, as a means to aid the host during infection is well established (reviewed in⁶¹) and is suggested to operate by removing the intracellular niches for selected pathogens⁶¹. Moreover, the cellular debris resulting from apoptosis can trigger activation of innate immune responses (reviewed in⁶¹) such as the complement system, which subsequently enhances clearance of the infection (reviewed in⁶²). The different host

immune response elicited by the Δ CbpD strain compared to the WT may reflect the hypovirulent nature of the Δ CbpD strain and underpins the contribution of CbpD significantly to the extent and consequence of PA-associated infection. From the perspective of understanding PA systemic infections in general, one important host marker unique to the WT-infected mice was the complement negative regulator factor H (**Dataset 11**). This observation indicates that the enrichment of cellular processes such as “negative regulation of immune responses” in the WT-infected mice may be partly responsible for the development of intense bacteremia observed for this strain. Importantly, increased expression of factor H during bacterial infection is associated with prolonged hospitalization of the patients⁶³.”

The newly generated figure legends to Supplementary Figures 13 (line 1396-1417) and 14 (line 1438-1450) read:

Supplementary Fig. 13. The effect of CbpD on the splenic proteome during systemic infection in vivo. **a** Pathways enriched in the shared splenic proteome of the WT-/ Δ CbpD-infected vs uninfected mice. The enrichment analysis was performed using Metascape (p value cut off=0.01). **b** STRING network analysis showing the connection of regulated proteins in the unique spleen proteome of Δ CbpD-infected mice vs control (dataset 10). The average fold change values of up- or down-regulated proteins are mapped to the nodes and is visualized using a blue-white-red gradient. Proteins without any interaction partners within the network (singletons) are omitted from the visualization. **c** The score of the molecular networks associated with regulated proteins (Dataset 10) in the unique spleen proteome of Δ CbpD-infected vs control mice. In total, 119 out of 128 uniquely regulated proteins (Dataset 10) were mapped to the database, and nine networks were identified by Ingenuity Pathway Analysis (IPA) (see Supplementary Fig. 14 for details). The number of the focus molecules (differentially regulated) associated with each network is presented on top of the chart. **d** IPA canonical pathways significantly altered in Δ CbpD-infected mice. The bottom X-axis shows the percentage of upregulated (red), downregulated (blue), and proteins not overlapping with other data set (white) in each pathway. The top X-axis shows the enrichment score ($-\log p$ -value) for each pathway (cut off=0.01) as indicated by the red dots. The numbers to the right of each bar show the total number of proteins in that particular canonical pathway. **e** Molecular and disease-based protein network detected by IPA analysis. The network is associated with the regulated proteins in the unique spleen proteome of WT-infected (Dataset 10) vs control mice. The average fold change value of up- or down-regulated proteins are mapped to the nodes and are visualized using a green-white-red gradient. The shapes represent the molecular classes of the proteins (see legend to supplementary Fig. 14). Direct and indirect connections are presented with solid and dashed arrows, respectively.

Supplementary Fig. 14. Molecular and disease-based protein networks identified by IPA. Regulated proteins in the unique spleen proteome of Δ CbpD-infected mice vs control were used for IPA analysis (dataset 10). The score and rank of the networks are presented in supplementary Fig. 13c. Direct and indirect connections are presented with solid and dashed arrows, respectively. The average fold change value of up- or down-regulated proteins are mapped to the nodes and are visualized using a green-white-red gradient. The shapes represent the molecular classes of the proteins. Network 1: Inflammatory response, lipid metabolism, small molecule biochemistry; Network 2: cancer, cell-to-cell signaling and interaction, cellular movement; Network 3: cancer, cellular assembly and organization, neurological disease; Network 4: cancer, endocrine system disorders, protein trafficking, Network 5: cancer, molecular transport, organismal functions; Network 6: cell cycle, gene expression, RNA damage, and repair; Network 7: cell signaling, post-translational modification, protein synthesis; Network 8: cell signaling, molecular transport, small molecule biochemistry; Network 9: cell death and survival, cell signaling, vitamin, and mineral metabolism.”

The “SUPPLEMENTARY MATERIALS AND METHODS” on “**Defining the murine splenic proteome in response to PA14 and Δ CbpD infection**” (line 747-755) has been updated accordingly.

Supplemental

Line 111 refers to Fig. S6D which does not exist.

Response: This has been corrected to Fig. S6C.

Nomenclature of glycans in supplemental dataset was not clear and mucins were not evaluated (similar to function of GbpA in *V. cholera*)

Response: We apologize for any lack of clarity in the nomenclature of the glycans in the glycan array supplemental dataset (**Dataset 1**). The file was obtained from the National Center of Functional Glycomics (NCFG) that performed the analysis, and the glycan monosaccharides are named according to the established IUPAC nomenclature. However, we agree that the way the glycans are named is not optimal since the stereochemistry of the anomeric carbon of the monosaccharides are indicated by “a” or “b” instead of their Greek letter. Also, the spacer type used to anchor the glycans to the chip is also indicated for each glycan, which may confuse the reader. In order to clarify the glycan nomenclature, we have therefore added a detailed explanation in the dataset legend.

The revised data file legend (Supplemental File, line 1722-1730) reads:

“Dataset 1. *The data provided show the binding of fluorescently labeled CbpD purified from either Pseudomonas aeruginosa (rCbpD_{PA}) or E. coli (rCbpD_{EC}) to version 5.3 of the mammalian glycan array provided by the National Consortium for Functional Glycomics (NCFG). Monosaccharides are named according to the established IUPAC nomenclature. The “Spn” ending of all glycan names indicates the spacer type used for attaching the glycan to the glycan chip surface. The “a” or “b” following the monosaccharide abbreviation indicates α - or β -configuration, of the anomeric carbon, respectively. Detailed information can be found at the NCFG website (www.functionalglycomics.org). The screening was performed in 15 mM Tris-HCl pH 7.5, 150 mM NaCl at final rCbpD concentrations of 5 or 50 $\mu\text{g}\cdot\text{ml}^{-1}$. Data are presented as mean \pm standard deviation (SD) of one experiment performed in triplicate.”*

The referee also comments on the evaluation of CbpD binding to mucins, which appeared not to be analyzed in the original manuscript. This is a very good point as mucin binding was observed for GbpA from *V. cholerae*, which is a CbpD ortholog. However, the mammalian glycan array is relevant for the evaluation of mucin binding as it contains core 1, type 1, blood group, and Lewis type glycans that are found in mucins (see e.g., Table 10.1 in the book Essentials of Glycobiology [Internet]. 3rd edition., Varki A., Cummings RD, Esko JD *et al* (editors) (<https://www.ncbi.nlm.nih.gov/books/NBK453030/> for details). We realize that this should have been indicated in the manuscript and have updated the revised version accordingly. We have also included a reference to the relevant chapter in the book referred to in the former sentence. In the revised manuscript, we have added the following sentence to the “RESULTS” under the title “CbpD enzymatic performance is influenced by other redox-active virulence factors” to amend this lack of information.

The revised text (line 116-121) now reads:

*“To explore other potential CbpD substrates, the binding properties of rCbpD_{EC} and rCbpD_{PA} were screened using a mammalian glycan array that contains 585 glycan structures present in mammals, including a variety of structures found in mucins such as core 1, type 1, blood group and Lewis type glycans³⁹. Binding was not detected to any glycan for either CbpD variant at 5 and 50 $\mu\text{g}\cdot\text{ml}^{-1}$ (**Dataset 1**), nor was binding to GlcNAc₆ (water-soluble chitooligosaccharide) observed (**Supplementary Fig. 7b and Dataset 1**).”*

The Phylogenetic tree in figure S2 has no branch support, please add this information.

Response: Thank you for pointing this out. We have now added branch support values based on 100 bootstrapped trees. Please note that the original online software used to generate the phylogenetic tree (phylogeny.fr) had been updated after we used it to generate the original tree, and we did find a possibility in the new version to show branch support values. Thus, the revised phylogenetic tree was constructed using the ETE3 v3.1.1. toolkit for analysis and visualization of phylogenetic trees (online version provided by ww.genome.jp). The resulting tree is essentially identical to the original but is displayed differently to accommodate optimal placement of the branch support values.

The revised figure legend reads:

“Supplementary Fig. 2. Phylogenetic tree of selected family AA10 LPMOs (AA10 modules only). *The phylogenetic tree is based on a multiple sequence alignment of selected sequences made using MUSCLE (with default parameters) and was constructed using the “build” function of ETE3 v3.1.1⁶³ that utilizes PhyML v20160115. Branch support values were computed from 100 bootstrapped trees.*

UniProt identifiers for the proteins are indicated next to the name of the bacterial species. CbpD is indicated in red-colored bold formatting.”

Reviewer #3 (Remarks to the Author):

I am largely commenting on the structural part of the Ms. The authors constructed a homology model of CbpD, performed MD simulations and validated/refined the model against the SAXS data. The quality of the experimental SAXS data collected on a lab source is somewhat limited. The analysis is based on the measurements of a single concentration only (in a normal practice, a concentration dependence is measured and extrapolation to zero solute content is made and the authors should explain why was this not done).

Response: Although we are confident that the original experiment was of high quality, we agree that obtaining datasets from different CbpD concentrations further increases the quality of the model. Such an experiment could not be done using our home source, but fortunately, we were able to obtain beam time at the ESRF recently. We collected synchrotron SAXS data at ESRF beamline BM29 for four different concentrations of rCbpD_{EC} at 20 °C and extrapolated the data to zero solute concentration. The new data are of higher quality than the home source data, and resulted in minor changes of the SAXS model, however, the conclusions from the SAXS studies are unchanged. The new data (**Supplementary Figure 4**) and associated table (**Supplementary Table 2**) prepared according to the guidelines presented in Trewhella *et al.* (Acta Cryst. (2017), D73, 710), have been included in the revised version of the manuscript. We further prepared tables for SAXS data collected at 24 and 37 °C, respectively, using our home source, which are attached at the end of the response letter for the reviewer’s perusal (they are not included in the manuscript). All three data sets are deposited in the Small Angle Scattering Biological Data Bank SASBDB (entries SASDK42, SASDJQ5 and SASDJR5). In addition, the “SUPPLEMENTARY RESULTS” and “SUPPLEMENTARY MATERIALS AND METHODS” have been updated.

The updated “SUPPLEMENTARY RESULTS” (line 102-124) now reads:

“To obtain experimental data on the potential flexibility of CbpD, the solution structure of full-length CbpD was analyzed using Small Angle X-ray Scattering (SAXS) (**Figs. 1c and Supplementary Fig. 4**; SASBDB ID: SASDK42). Recombinantly produced full-length rCbpD_{EC} was measured at 20 °C (**Supplementary Fig. 4a and 4b**). SAXS data and analysis are summarized in **Supplementary Table 2**. Both the forward scattering, obtained from the Guinier analysis (**Supplementary Fig. 4c**), and the Porod volume (**Supplementary Fig. 4e**) are consistent with a monomeric protein. The large radius of gyration of 32.16 +/- 0.11 Å, which approximately matches the predicted value (**Supplementary Fig. 3b**), and the shape of the pair-distance distribution curve (**Supplementary Fig. 4d**), both indicate an elongated protein shape (**Fig. 1c**). The maximum diameter of the protein was estimated to be approximately 120 Å. We plotted the data in a dimensionless Kratky plot (**Supplementary Fig. 4f**), giving a bell-shaped curve, consistent with a folded protein, but with a maximum at $qR_g = 3$, which is higher than the maximum of $qR_g = \sqrt{3}$ for completely globular proteins⁵. This indicated that the protein is folded but has flexible regions. We generated 20 low-resolution models, which were averaged and refined to obtain one representative low-resolution model (**Fig. 1c**). CbpD has highly flexible regions, especially the L2 linker, which significantly limits the level of detail provided by the low-resolution model. It was therefore at first challenging to estimate which end of the molecular envelope belonged to the larger AA10 domain, and which to the CBM73. However, superimposing the atomistic CbpD Peps-SAXS model ($\chi^2 = 2.35$) onto the low-resolution *ab initio* model envelope resulted in a good fit (**Fig. 1c**), giving high confidence in the model. Comparison of the SAXS data collected at the synchrotron (ESRF) to the data collected on our home source at 24 °C (SASDJQ5) and 37 °C (SASDJR5) showed that the CbpD solution structure is consistent over this temperature range. The elongated shape of the CbpD model resembles the solution structure of the homologous GbpA, which was determined with SAXS by Wong *et al.*⁶.”

The updated “SUPPLEMENTARY MATERIALS AND METHODS” on “Small-angle X-ray scattering” (line 293-318) along with updated **Supplementary Figure 4** and the associated Figure legend has been provided accordingly.

Overall, though, the structural model looks feasible and corroborates the simulations. What appears missing is the relation of the structural results to the subsequent studies (functional, molecular

biological, ex vivo etc). It would have been useful to explain how the structural model and its observed flexibility help in the understanding of the functional features of the protein.

Response: We agree that better relating the structural data to the protein's functional features could further improve our manuscript. As described in our response to Reviewers#1 and 2 (e.g., see comment no. 1 of Reviewer#1), one of the experiments we performed for this revision was a respiratory infection model study, which indeed showed that CbpD is important for a form of infection related to mucosal surfaces, thereby creating a closer link to the proposed function of the *Vibrio cholerae* LPMO and CbpD ortholog, GbpA. The latter protein has been shown to be important for the colonization (and most likely survival) of *V. cholerae* of the host gut mucosa. GbpA is the only multi-domain LPMO with its structure solved. Since multiple domains (both carbohydrate binding domains and domains of unknown function) are a hallmark for virulence-related LPMOs, we decided to perform a deeper structural investigation of GbpA. As a starting point for this analysis, we performed a search at the DALI server using the GbpA crystal structure intending to identify proteins structurally similar to GbpA (and thus indirectly to CbpD), which could add novel information to the comparison. Excitingly, some of the top hits from the DALI search, that were not other LPMOs, were complement factors C3, C4, and C5, the C5b-C6 complex giving the best match for the full-length protein (these three complement factors are paralogs, and although sequence similarity is relatively low between the three proteins, their structures are similar). The similarity can of course be by chance, but in light of the functional data we have found for CbpD, it does bring support to the hypothesis that the LPMO domain interacts with parts of the complement system and thereby attenuates its impact on bacterial killing, in line with a classical “molecular mimicry” model, where an inhibitory protein adopts the shape and properties of parts of a host protein complex, acting as a substitute that prevents the host protein complex in performing its activity. Following up on these observations for GbpA, we performed a similar DALI search using the CbpD model (structurally fitted to the SAXS data), but no similar matches could be found. This may be due to the elongated conformation of CbpD in solution compared to the more U-shaped form of GbpA in its crystal structure form (which we used in the DALI search). Importantly, the SAXS data for GbpA published by Wong *et al.* (Plos Pathogens 8 (1): e1002373. <https://doi.org/10.1371/journal.ppat.1002373>) show an elongated conformation, similar to what we observe for CbpD. Therefore, it is likely that the flexible linkers joining the individual modules of these LPMOs allow accommodation of the protein shape to that of the putative interaction partner, in this case mimicking the C3, C4 or C5 (or their respective sub-parts -a or -b) in the complement cascade. Our SAXS and MD data indicate that CbpD is a flexible protein, supporting the hypothesis that the protein can adopt a conformation similar to the GbpA crystal structure. We acknowledge that the above is somewhat speculative, but we find the observations intriguing in the context of our functional data and have therefore concluded that these novel findings should be included in the revised manuscript. Due to the speculative nature of these results, we have added a cautious short piece of text to the revised manuscript to couple the structural data to the functional data. We have also added a supplemental figure (**Supplementary Figure 5, panels: d and e**) showing the superposition of GbpA on the C5 molecule.

The updated “DISCUSSION” (line 496-503) in the revised manuscript reads:

“In this context, it is noteworthy that the crystal structure of GbpA shows structural similarity to macroglobulin-like (MG) domains 2,3 and 4 of the C5b molecule present in the C5b-C6 complex (Supplementary Fig. 5d). Such a similarity is not apparent for the CbpD model, which exhibits an elongated shape in solution, as has also been observed for GbpA by SAXS experiments¹³. The flexibility of these proteins, as e.g. observed in our SAXS, MD and modelling data (e.g., Fig. 1) may allow adoption of a U-shaped conformation like the GbpA crystal structure (Supplementary Fig. 5e). The similarity of virulence-related LPMOs to complement factors may indicate a role of these proteins is related to molecular mimicry.”

The updated figure legend reads:

“Supplementary Fig. 5. Homology modeling of CbpD domain 3 and structural similarity between GbpA and the complement factor C5-C6 complex. a Ribbon representation of superimposed CBM5 (PDB ID: 2D49) and CMB12 (PDB IDs: 4TX8 and 1ED7) domains that were used to predict the structure of the CBM73 domain of CbpD in RaptorX. **b** Snapshot of the MD-equilibrated third domain of CbpD (CBM73; violet), superimposed on the CBM12 domain of PDB ID: 4TX8 (beige). **c** Cartoon representation of CbpD domain 3 (CBM73). The predicted disulfide bridge is shown in ball and stick representation. **d** Structural similarity between GbpA and the C5-C6 complex identified by searching

the DALI database ⁶⁴ (Z-score = 10.2, RMSD = 13 Å from 1528 aligned atoms) The C5b-C6 complex (PDB ID: 4A5W) and GbpA (PDB ID: 2XWX) are shown in cartoon representation. The C5b and C6 molecules are colored orange and blue, respectively, and the GbpA molecule is colored purple. The right panel shows an enlarged image of GbpA superimposed on the MG2, -3 and -4 domains of C5b (see ⁶⁵ for a detailed description of the C5b-C6 complex). **e** Superposition of GbpA (PDB ID: 2XWX; green colored ribbon) on the CbpD model fitted to the SAXS-data (purple colored ribbon), yielding a RMSD = 17 Å from 1364 aligned atoms. The right panel shows superpositioning of the individual CbpD modules on the GbpA structure, dashed arrows indicating the putative structural movement of the CbpD modules needed to obtain the GbpA 3D modular arrangement. The individual superpositions yielded RMSD values of 1.8 Å (705 atoms aligned), 1.9 Å (540 atoms aligned) and 8.0 Å (141 atoms aligned) for the AA10 modules M2 modules and M3/CBM73 modules, respectively. It should be noted that the GbpA M3 module has low structural similarity to the CbpD CMB73 model, but the overall shape is relatively similar, the latter being slightly smaller.”

According to the publication guidelines of SAXS data (Trehwella et al Acta Cryst. (2017), D73, 710), a Table summarizing the results should be presented (see Table 5 in the guidelines).

Response: We apologize for overlooking this requirement in the previous version. As mentioned above, a table (**Supplementary Table 2**) generated according to the guidelines presented in Trehwella *et al.* (Acta Cryst. (2017), D73, 710) has now been added in the new version of the manuscript. We also provided the table (as well as those for the in-house SAXS data) at the end of the response letter for your convenience.

Further comments:

The references to the programs used for Fig. 1B, C (Raptor-X, DAMMIF, Pepsi-SAXS) must be given in the main text. Similar holds for the deposition codes to SASBDB.

Response: The deposition codes for the SASBDB have now been included in the section “DATA AVAILABILITY” and in Supplementary Table 2 (SASBDB IDs: SASDK42 (20 °C). The SASBDB entries will become public upon publication of the manuscript. The reviewer can access the entries with the following URLs: 20 °C: <https://www.sasbdb.org/data/SASDK42/ycgo4ivmjpl/>

Moreover, the reviewer can also access to the data associated with the home source facility, SASDJQ5 (24 °C) and SASDJR5 (37 °C), through the following links:

24 °C: <https://www.sasbdb.org/data/SASDJQ5/243l3pg5k4/>

37 °C: <https://www.sasbdb.org/data/SASDJR5/ttd5lwwme5/>

The respective parts of the legend of figure 1 now read: “**b** Homology model of CbpD generated with Raptor-X⁸⁶, showing flexible linkers in an extended conformation. The active site is indicated. **c** SAXS model of monomeric CbpD ($\chi^2 = 2.35$; produced with Pepsi-SAXS⁸⁷; SASBDB ID: SASDK42), superimposed onto the *ab initio* SAXS model ‘envelope’ (produced with DAMMIF^{88,89} based on an averaging of 20 calculated models). Panels b and c were generated using PyMol.”

References in the main text have been updated as follows:

86. Kallberg, M. *et al.* Template-based protein structure modeling using the RaptorX web server. *Nat Protoc* **7**, 1511-1522, doi:10.1038/nprot.2012.085 (2012).

87. Grudin, S., Garkavenko, M. & Kazennov, A. Pepsi-SAXS: an adaptive method for rapid and accurate computation of small-angle X-ray scattering profiles. *Acta Crystallogr D Struct Biol* **73**, 449-464, doi:10.1107/S2059798317005745 (2017).

88. Franke, D. & Svergun, D. I. DAMMIF, a program for rapid *ab-initio* shape determination in small-angle scattering. *J Appl Crystallogr* **42**, 342-346, doi:10.1107/S0021889809000338 (2009).

89. Volkov, V. V. & Svergun, D. I. Uniqueness of *ab initio* shape determination in small-angle scattering. *J Appl Crystallogr* **36**, 860-864, doi:10.1107/S0021889803000268 (2003).

Chi2: give a proper greek letter

Response: We thank the reviewer for alerting us to the mistake in Fig. 4 panel “a”. The updated figure contains the synchrotron data. All panels have been carefully reviewed to avoid new mistakes.

The reported value of the maximum size ($D_{max}=121$ Angstrom) is meaningless. The accuracy of D_{max} is at best 5 Angstrom, for such a long particle rather 10 Angstrom.

Response: We agree with the reviewer that the precision of D_{max} is lower than suggested by the 121 Å value and have now corrected it to 120 Å to properly reflect the precision. We still believe it has relevance to report the maximum size, since it is needed in the shape modeling.

Fig S4: range of q in plate A is incorrect.

Response: We thank the reviewer for alerting us to the mistake in Fig. S4A of the original manuscript. The original Supplementary Figure 4 has now been replaced by a figure that shows the new SAXS data (20 °C; ESRF BM29) as described above.

Supplementary Ref 32 appears incomplete

Response: We thank the reviewers for bringing our attention to this point. Ref 32 has now been corrected as follows in both the supplementary (Ref 35) and the main text (Ref 89):

- Volkov, V. V. & Svergun, D. I. Uniqueness of *ab initio* shape determination in small-angle scattering. *J Appl Crystallogr* **36**, 860-864, doi:10.1107/S0021889803000268 (2003)

Supplementary Table 2. SAXS data collection and analysis (20 °C; ESRF BM29).

(a) Sample details	
	CbpD
Organism	Pseudomonas aeruginosa
Source (Invitrogen)	E. coli expression host (BL21 star)
Uniprot sequence ID (Residues in construct)	Q9I589 (26-389)
Extinction coefficient [A_{280} , 0.1% (=1 g/l)]	2.098
\bar{v} from chemical composition ($\text{cm}^3 \text{g}^{-1}$)	0.73
Particle contrast, $\Delta\rho$ (10^{10}cm^{-2})	3.01
MM from chemical constituents (kDa)	39.2
Protein concentration (mg/mL)	2.50, 1.25, 0.63 and 0.31 (data extrapolated to zero solute concentration)
Solvent	150 mM NaCl, 15 mM Tris-HCl, pH 7.5

(b) SAXS data collection parameters	
	CbpD
Instrument	BM29 with Pilatus2M detector at ESRF ²⁵
Wavelength (Å)	0.992
Beam size (µm)	700 X 700
Sample to detector distance (cm)	282.7
q measurement range (Å^{-1})	0.00449 - 0.51787
Absolute scaling method	Milli-Q water standard measurement
Normalization	Transmitted intensities through semi-transparent beam-stop
Exposure time (h)	10 frames of 1 s
Radiation damage monitorization	Frame to frame evaluation and comparison with home-source data
Capillary size (mm)	1.8
Sample temperature	20 °C

(c) Software employed for SAXS data reduction, analysis and interpretation	
	CbpD
SAXS data reduction	BsxCuBE beamline software ²⁵
Extinction coefficient estimate	ProtParam ⁵⁹
Calculation of contrast and specific volume	MULCh1.1 ⁶⁰
Basic analysis	PRIMUS (ATSAS) ^{32, 33}
Shape reconstruction	DAMMIF ³⁴ /DAMAVER ³⁵ /DAMMIN ³⁶
Atomic structure modelling	Pepsi-SAXS ³⁷
Representation	PyMOL

(d) Structural parameters	
	CbpD
Guinier analysis	
$I(0)$ (cm ⁻¹)	0.03086 ± 0.00007
R_g^{**} (Å)	32.16 ± 0.11
q_{min} (Å ⁻¹)	0.0137
qR_g max	1.29
R^2	0.987
MM from $I(0)$ (kDa) (ratio to predicted)	38.7 (0.99)
$P(r)$ analysis	
$I(0)$ (cm ⁻¹)	0.0314 ± 0.0001
R_g (Å)	34.51 ± 0.1411
d_{max}^{***} (Å)	120
q range (Å ⁻¹)	0.0137 - 0.2480 (0.1760)
χ^2	0.99
Total quality estimate from PRIMUS	0.78
MM from $I(0)$ (kDa) (ratio to predicted)	39.4 (1.01)
Porod analysis	
Porod volume (Å ³)	64131
MM from Porod volume (kDa) (ratio to predicted)	40.1 (1.02)

(e) Shape model-fitting results	
	CbpD
DAMMIF	
q range (Å ⁻¹)	0.0137 - 0.2480
Symmetry, anisotropy assumptions	$P1$, none
NSD (Standard deviation)	0.917 (0.192)
Constant adjustment	Skipped
Resolution (Å)	33 +/-3
MM from DAMMIF (kDa) (ratio to predicted)	30.1 (0.77)
χ^2	1.668-1.676
DAMAVER/DAMMIN	
q range	0.0137 - 0.2480
Symmetry, anisotropy assumptions	$P1$, none
χ^2	1.642
CorMap p-value	0.1403
Constant adjustment	Skipped

(f) Atomistic modelling	
	CbpD
Pepsi-SAXS	
Homology model	Raptor-X https://github.com/kiendseth/CBPD_MD_simulations/tree/master
q range (Å ⁻¹)	0.00449 - 0.51787
Flexibility optimization	Enabled
Constant adjustment, allowed	-0.000127621
Rigid bodies, residues	AA1-115, AA192-292, AA307-367
χ^2	2.35

(g) SASBDB ID	
SASDK42	

*Molecular mass; **Radius of gyration; *** d_{max} : Maximum diameter

SAXS data collection and analysis (24 and 37°C, in-house Bruker NanoStar instrument (RECX) at the University of Oslo.

(a) Sample details	
	CbpD
Organism	Pseudomonas aeruginosa
Source (Invitrogen)	E. coli expression host (BL21 star)
Uniprot sequence ID (Residues in construct)	Q9I589 (26-389)
Extinction coefficient [A_{280} , 0.1% (=1 g/l)]	2.098
\bar{v} from chemical composition (cm ³ g ⁻¹)	0.73
Particle contrast, $\Delta\rho$	3.01
MM from chemical constituents (kDa)	39.2

Protein concentration (mg/mL)	2.18
Solvent	150 mM NaCl, 15 mM Tris-HCl, pH 7.5

(b) SAXS data collection parameters	
	CbpD
Instrument	Bruker Nanostar with InCoatec Cu microsource and Vântec-2000 detector
Wavelength (Å)	1.54
Beam size (µm)	750 X 750
Sample to detector distance (cm)	109
q measurement range (Å ⁻¹)	0.00925 - 0.29866
Absolute scaling method	Milli-Q water standard measurement
Normalization	Transmitted intensities through semi-transparent beam-stop
Exposure time (h)	2
Capillary size (mm)	1.5
Sample temperature	24 °C and 37 °C (only 24 °C used for data analysis)

(c) Software employed for SAXS data reduction, analysis and interpretation	
	CbpD
SAXS data reduction	SUPERSAXS (CLP Oliveira and JS Pedersen, unpublished, based on principles explained in ¹ , implementation partially explained in ²)
Extinction coefficient estimate	ProtParam ³
Calculation of contrast and specific volume	MULCh1.1 ⁴
Basic analysis	PRIMUS (ATSAS) ^{5,6}
Shape reconstruction	DAMMIF ⁷ / DAMAVER ⁸ / DAMMIN ⁹
Atomic Structure Modelling	Pepsi-SAXS ¹⁰
Representation	PyMOL

(d) Structural parameters	
	CbpD
Guinier analysis	
$I(0)$ (cm ⁻¹)	0.0257 ± 0.0004
R_g (Å)	32.5 ± 0.9
q_{min} (Å ⁻¹)	0.011
qR_g max	1.19
R^2	0.88
MM from $I(0)$ (kDa) (ratio to predicted)	37.6
$P(r)$ analysis	
$I(0)$ (cm ⁻¹)	0.0264 ± 0.0005
R_g (Å)	35.0 ± 0.9
d_{max} (Å)	120
q range (Å ⁻¹)	0.00996-0.25067
Total quality estimate from PRIMUS	0.78
MM from $I(0)$ (kDa) (ratio to predicted)	39.7 (1.01)
Porod volume (Å ³)	73500
MM from Porod volume (kDa)	45.9 (1.17)

(e) Shape model-fitting results	
	CbpD
DAMMIF	
q range (Å ⁻¹)	0.00925 - 0.22790
Symmetry, anisotropy assumptions	$P1$, none
NSD (Standard deviation)	1.034 (0.358)
Constant adjustment	Skipped
Resolution (Å)	36 +/-3
MM from DAMMIF (kDa) (ratio to predicted)	33.0 (0.85)
χ^2	0.889-0.894
DAMAVER/DAMMIN	
q range	0.00925 - 0.22790
Symmetry, anisotropy assumptions	$P1$, none
χ^2	0.89
Constant adjustment	Skipped

(f) Atomistic modelling	
	CbpD
Pepsi-SAXS Homology model q range (\AA^{-1}) Flexibility optimization Constant adjustment, allowed Rigid blocks, residues χ^2	Raptor-X (https://github.com/kjendseth/CBPD_MD_simulations/tree/master) 0.009246 – 0.298655 Enabled -0.0001730 AA1-115, AA192-292, AA307-367 1.15

(g) SASBDB IDs	
24 °C	37 °C
SASDJQ5	SASDJR5 (Not used for modelling)

*Molecular mass; **Radius of gyration; ** d_{\max} : Maximum diameter.

References associated with SAXS data collection at 24 and 37°C:

- Lindner, P. in *Neutrons, X-rays, and light: scattering methods applied to soft condensed matter*. Vol. 1 (ed P.; Zemp Lindner, T.) 23-48 (Elsevier, 2002).
- Pedersen, J. S. A flux- and background-optimized version of the NanoSTAR small-angle X-ray scattering camera for solution scattering. *J Appl Crystallogr* **37**, 369-380, doi:10.1107/s0021889804004170 (2004).
- Gasteiger, E. *et al.* Gasteiger, E. *et al.* Protein identification and analysis tools on the ExpASY server; (In) John M. Walker (ed): *The Proteomics Protocols Handbook*, 571-607 (Humana Press, 2005).
- Whitten, A. E., Cai, S. & Trehwella, J. *MULCh*: modules for the analysis of small-angle neutron contrast variation data from biomolecular assemblies. *J Appl Crystallogr* **41**, 222-226, doi:10.1107/s0021889807055136 (2008).
- Franke, D. *et al.* *ATSAS 2.8*: a comprehensive data analysis suite for small-angle scattering from macromolecular solutions. *J Appl Crystallogr* **50**, 1212-1225, doi:10.1107/s1600576717007786 (2017).
- Konarev, P. V., Volkov, V. V., Sokolova, A. V., Koch, M. H. J. & Svergun, D. I. *PRIMUS*: a Windows PC-based system for small-angle scattering data analysis. *J Appl Crystallogr* **36**, 1277-1282, doi:10.1107/s0021889803012779 (2003).
- Franke, D. & Svergun, D. I. *DAMMIF*, a program for rapid *ab-initio* shape determination in small-angle scattering. *J Appl Crystallogr* **42**, 342-346, doi:10.1107/S0021889809000338 (2009).
- Volkov, V. V. & Svergun, D. I. Uniqueness of *ab initio* shape determination in small-angle scattering. *J Appl Crystallogr* **36**, 860-864 (2003).
- Svergun, D. I. Restoring low resolution structure of biological macromolecules from solution scattering using simulated annealing. *Biophys J* **76**, 2879-2886, doi:10.1016/s0006-3495(99)77443-6 (1999).
- Grudin, S., Garkavenko, M. & Kazennov, A. *Pepsi-SAXS*: an adaptive method for rapid and accurate computation of small-angle X-ray scattering profiles. *Acta Crystallogr D Struct Biol* **73**, 449-464, doi:10.1107/S2059798317005745 (2017).

REVIEWERS' COMMENTS

Reviewer #1 (Remarks to the Author):

A substantial amount of new data has been provided that add to the strength of the authors' findings. This is an excellent contribution.

Reviewer #2 (Remarks to the Author):

This reviewer appreciates all of the clarifications and revisions made by the authors. The additional experiments that were performed and content that has been supplemented in the discussion section have further strengthened this manuscript and provide compelling results and sound conclusions. As such, this reviewer recommends this manuscript for publication with the following very minor revisions.

Minor revisions

1. The legend of Figure 4a specifies "left panel" twice (line 1019) when specifying the routes of infection. It appears that the "(right panel)" text should follow "intratracheally (IT) with 5×10^6 CFU" and the second mention of the "(left panel)" should be deleted.
2. The methods section lists the intravenous inoculum as 2×10^8 CFU (line 688 in the supplementary materials) whereas the Figure 4 legend (line 1019) and its mention in the main text (line 288) lists the intravenous inoculum as 2×10^7 CFU. Please correct the discrepancy.

Reviewer #3 (Remarks to the Author):

The authors took into account the comments and the Ms can be recommended for publication

Second revision of “The lytic polysaccharide monoxygenase CbpD promotes *Pseudomonas aeruginosa* virulence in systemic infection” (NCOMMS-20-25008-T)

REVIEWERS' COMMENTS

Reviewer #1 (Remarks to the Author):

A substantial amount of new data has been provided that add to the strength of the authors' findings. This is an excellent contribution.

Response: We thank the Reviewer for the positive feedback.

Reviewer #2 (Remarks to the Author):

This reviewer appreciates all of the clarifications and revisions made by the authors. The additional experiments that were performed and content that has been supplemented in the discussion section have further strengthened this manuscript and provide compelling results and sound conclusions. As such, this reviewer recommends this manuscript for publication with the following very minor revisions.

Response: We thank the Reviewer for the positive feedback.

Minor revisions

1. The legend of Figure 4a specifies “left panel” twice (line 1019) when specifying the routes of infection. It appears that the “(right panel)” text should follow “intratracheally (IT) with 5×10^6 CFU” and the second mention of the “(left panel)” should be deleted.

Response: Thank you for spotting this error. It has been corrected in the revised manuscript.

The original sentence read:

“CD1 mice (n=10 per group) were inoculated intravenously (IV) (left panel) with 2×10^7 CFU (right panel) and intratracheally (IT) with 5×10^6 CFU (left panel) PA WT or Δ CbpD per mouse.”

The revised sentence now reads:

“CD1 mice were inoculated intravenously (IV) with 2×10^7 CFU (left panel) and intratracheally (IT) with 5×10^6 CFU (right panel) PA WT or Δ CbpD per mouse.”

2. The methods section lists the intravenous inoculum as 2×10^8 CFU (line 688 in the supplementary materials) whereas the Figure 4 legend (line 1019) and its mention in the main text (line 288) lists the intravenous inoculum as 2×10^7 CFU. Please correct the discrepancy.

Response: Thank you for spotting this error, the “Method” has been updated to 2×10^7 CFU.

Reviewer #3 (Remarks to the Author):

The authors took into account the comments and the Ms can be recommended for publication.

Response: We thank the Reviewer for the positive feedback.